# An analogue of the Prolactin Releasing Peptide reduces obesity and promotes adult neurogenesis

Sara KM Jörgensen[1], Alena Karnošová[2,3], Simone Mazzaferro [ID] [4,5], Oliver Rowley [ID] [1], Hsiao-Jou Cortina Chen [ID] [4,5], Sarah J Robbins[1], Sarah Christofides [ID] [1], Florian T Merkle [ID] [4,5], Lenka Maletínská [ID] [3] & David Petrik [ID] [1][✉]

## Abstract

**Hypothalamic Adult Neurogenesis (hAN) has been implicated in regulating energy homeostasis. Adult-generated neurons and adult Neural Stem Cells (aNSCs) in the hypothalamus control food intake and body weight. Conversely, diet-induced obesity (DIO) by high fat diets (HFD) exerts adverse influence on hAN. However, the effects of anti-obesity compounds on hAN are not known. To address this, we administered a lipidized analogue of an anti-obesity neuropeptide, Prolactin Releasing Peptide (PrRP), so-called LiPR, to mice. In the HFD context, LiPR rescued the survival of adult-born hypothalamic neurons and increased the number of aNSCs by reducing their activation. LiPR also rescued the reduction of immature hippocampal neurons and modulated calcium dynamics in iPSC-derived human neurons. In addition, some of these neurogenic effects were exerted by another anti-obesity compound, Liraglutide. These results show for the first time that anti-obesity neuropeptides influence adult neurogenesis and suggest that the neurogenic process can serve as a target of anti-obesity pharmacotherapy.**

**Keywords** Adult neurogenesis; Anti-obesity peptides; Hypothalamus; Neural stem cells; Prolactin Releasing Peptide
**Subject Categories** Metabolism; Neuroscience; Pharmacology & Drug Discovery

## Introduction

Energy homeostasis is regulated in the hypothalamus, a ventral part of diencephalon critical for basic physiological functions (Lechan and Toni, 2000; Timper and Bruning, 2017). In the Medial Basal Hypothalamus (MBH), anorexigenic (i.e., food-intake inhibiting) and orexigenic (i.e., food-intake promoting) neurons sense metabolites and endocrine signals from the periphery to regulate eating behaviour and appetite (Betley et al, 2013; Farooqi, 2022).

Anti-obesity compounds lower the sensation of appetite and the body weight by targeting neurons in the MBH (Tak and Lee, 2021). Unfortunately, several anti-obesity compounds originally approved for clinical use, were later rejected for severe side effects such as foetal toxicity, nausea, or depression (Aronne, 2017; Patel and Stanford, 2018; Saunders et al, 2016). The only neuroactive medicines with anti-obesity activity approved in the UK and EU are analogues of the Glucagon-like Peptide-1 Receptor A (GLP-1RA) such as Liraglutide (Saxenda) or Semaglutide (Ozempic) (Bray et al, 2016; Collins and Costello, 2022; NICE, 2020). However, their primary indication is for Type-2 diabetes mellitus (T2DM) (Ard et al, 2021; Bailey et al, 2023; Mahase, 2022). Clearly, there is a need for novel neuroactive anti-obesity compounds. Lipidized analogues of PrRP are a novel class of compounds with robust anorexigenic and anti-obesity effects (Maletinska et al, 2015; Prazienkova et al, 2017).

Despite its name, PrRP does not regulate prolactin release (Jarry et al, 2000), but is the Prlh gene-encoded anorexigenic neuropeptide (Lagerstrom et al, 2005), released by neurons in the hypothalamus (Hinuma et al, 1998), which increases neuronal survival and reduces food intake (Lawrence et al, 2000; Lawrence et al, 2002; Maletinska et al, 2019; Prazienkova et al, 2019). However, PrRP is blood brain barrier (BBB)-impermeable, and its activity is limited only to minutes (Lawrence et al, 2002). To circumvent these limitations, a lipidized analogue of PrRP was developed, which has extended pharmacokinetics and can exert its effect in the Central Nervous System after peripheral administration (Maletinska et al, 2015). This full-length, 31 amino acid long PrRP peptide palmitoylated on Lysine-11 (palm[11]-PrRP31), further referred as LiPR, activates cells in MBH to reduce food intake, fat mass and weight gain (by around −15%) in mice exposed to the obesity-inducing high fat diet (HFD) (Mikulaskova et al, 2016; Prazienkova et al, 2017). To exert its anorexigenic effects, LiPR binds to PrRP transmembrane G-protein Coupled Receptors, GPR10 and NPFF2 (Engstrom et al, 2003; Hinuma et al, 1998; Marchese et al, 1995) with higher affinity than PrRP and 2-times higher affinity for NPFF2 than GPR10 (Karnosova et al, 2021; Mikulaskova et al, 2016).

While many of the appetite-controlling neurons in the MBH originate in embryogenesis, some of them are newly generated during adulthood from adult Neural Stem Cells (aNSCs), which reside in the Hypothalamic Ventricular Zone (HVZ) in the process

[1]School of Biosciences, Cardiff University, Cardiff CF10 3AX, UK. [2]First Faculty of Medicine, Charles University, Prague 12108, Czech Republic. [3]Institute of Organic Chemistry and Biochemistry of the Czech Academy of Sciences, Prague 16610, Czech Republic. [4]Wellcome-MRC Institute of Metabolic Science, Cambridge CB2 0QQ, UK. [5]Wellcome-MRC Stem Cell Institute, Cambridge CB2 0AW, UK. [✉]E-mail: petrikd@cardiff.ac.uk

of hypothalamic adult neurogenesis (hAN) (Batailler et al, 2014; Petrik et al, 2022). Both hypothalamic aNSCs and newborn MBH neurons are critical for energy homeostasis (Gouaze et al, 2013; Kokoeva et al, 2005).

In HVZ, tanycytes, radial-glia-like cells that line the walls of 3rd ventricle (3V), serve in a dual role as the putative hypothalamic aNSCs (htNSCs) (Kokoeva et al, 2005; Lee et al, 2012) and as metabolic regulators (Ebling and Lewis, 2018) to control energy homeostasis, body weight (Li et al, 2012; McNay et al, 2012; Zhang et al, 2017) and access of anti-obesity compounds to the brain parenchyma (Imbernon et al, 2022). Tanycytes give rise to differentiated cell progeny such as glial cells or neurons (Haan et al, 2013; Kokoeva et al, 2005; Pierce and Xu, 2010; Robins et al, 2013), which are part of almost 2000 cells generated daily in the adult murine hypothalamus (Gouaze et al, 2013). New hypothalamic neurons display unique properties different from embryo-generated neurons (Yoo et al, 2021) and make up to 37% of all hypothalamic neurons (Migaud et al, 2010).

DIO caused by the HFD has been shown to reduce hAN through lower survival and higher apoptosis of newborn neurons or by decreased proliferation of tanycytes (Lee et al, 2014; Li et al, 2012; McNay et al, 2012). On the other hand, ablation of hAN leads to higher weight gains in HFD-fed animals (Gouaze et al, 2013; Kokoeva et al, 2005), whereas increasing hAN protects against adverse effects of HFD (Kokoeva et al, 2005) suggesting that newborn neurons are anorexigenic. However, the relationship between hAN and DIO is insufficiently understood and marred by conflicting results (Lee et al, 2014; Li et al, 2012; McNay et al, 2012).

Given the importance of hAN for energy homeostasis, we hypothesized that anti-obesity compounds may be neurogenic in a similar manner to anti-depressants (Eisch and Petrik, 2012; Malberg et al, 2000; Santarelli et al, 2003). To test this hypothesis, we administered LiPR and Liraglutide to mice exposed to HFD and determined the effects of this treatment on cellular and molecular processes of hAN. Our results show that HFD reduces number of htNSCs in HVZ and survival of newborn neurons in MBH. In addition, HFD downregulates adult neurogenesis in another neurogenic niche, the Subgranular Zone (SGZ) of the hippocampus (Petrik et al, 2022). Administration of LiPR, but not Liraglutide, rescues these adverse effects of HFD on the differentiation of newborn neurons in SGZ and on survival of newborn neurons and on number of hypothalamic htNSCs in MBH. LiPR also reduces proliferation and activation of hypothalamic htNSCs. This suggests that LiPR upregulates hAN by preserving the population of htNSCs and survival of new neurons. For the first time, our data show that anti-obesity treatment upregulates adult neurogenesis, which was diminished by obesity.

# Results

## LiPR lowers body weight, ameliorates metabolic parameters related to obesity and increases expression of GPR10 in MBH

To determine how HFD and LiPR affect hAN in vivo, adult male C57BL/6J mice were exposed to three different protocols lasting 7 days (7d; Fig. 1A), 21d (Fig. 1B) or 4 months (4mo; Fig. 1C). The short (7d) and intermediate (21d) protocols were expected to initiate HFD-induced inflammation and astrogliosis in the hypothalamus (Sugiyama et al, 2020; Thaler et al, 2012), whereas the long (4mo) protocol leads to DIO (Prazienkova et al, 2017). The short and intermediate HFD protocols allowed us to determine the effects of LiPR in the context of neuroinflammation and developing metabolic syndrome, whereas the long protocol provided the context of developed obesity. In the 7d and 21d protocols, LiPR was administered concurrently with HFD, and during the last 2 weeks of the 4mo protocol. As a control to HFD+LiPR, mice were administered LiPR also in the context of Control (Low Fat) Diet in the 21d and 4mo protocols. To trace adult-generated cells, mice were administered 5-bromo-2'-deoxyuridine (BrdU) with different post-BrdU (so-called "chase") periods as indicated in the protocol schematics (Fig. 1A–C). Exposure to HFD or HFD+LiPR for 7d did not significantly change the body weight (Fig. 1A). In the 21d protocol (Fig. 1B), the Two-Way ANOVA revealed that treatment, which includes either Control or HFD with or without LiPR, had a significant effect on the body weight. However, the multiple comparison Bonferroni post-hoc test did not find any statistically significant difference between any of the treatment groups, suggesting that HFD or LiPR alone cannot cause the variance found by ANOVA. In contrast, there was a statistically extremely significant effect of the treatment, its duration, and their interaction on body weight in the last 2 weeks of 4mo exposure to HFD (Fig. 1C). Mice exposed to HFD had significantly higher body weight than Controls and HFD+LiPR mice demonstrating anorexigenic effects of LiPR as shown before (Maletinska et al, 2015). In contrast, LiPR had no effects on body weight in animals kept on Control diet. To characterize the systemic effects of DIO, selected hormones and metabolites were measured in mouse plasma. Exposure to 4mo HFD significantly increased concentration of insulin, leptin and cholesterol but not triglycerides (TAG) or free fatty acids (FFA; Fig. 1D–H), which is consistent with the development of the metabolic syndrome (Kennedy et al, 2010). Administration of LiPR for 2 weeks (wk) reduced the HFD-induced plasma concentration of insulin, leptin, and cholesterol. To exert its effects, LiPR binds to PrRP receptors GPR10 and NPFFR2 (Maletinska et al, 2015), which were identified in the hypothalamus but not in specific hypothalamic nuclei or associated with neuronal structures (Feng et al, 2007; Roland et al, 1999). Using immunohistochemistry (IHC), we identified GPR10-positive (GPR10+) puncta co-localizing with structures positive for the Microtubule-associated protein 2 (Map2) that labels neuronal cytoskeleton and/or surrounding neuronal nuclei expressing the marker of mature neurons, the Human neuronal protein C and D (HuC/D) (Batailler et al, 2014), in the MBH (Fig. 1I–K). This suggests that the receptor is expressed in neurons. Interestingly, exposure to 21d HFD reduced density of GPR10, which was rescued by LiPR administration (Fig. 1L). In addition, over 90% of Map2+ structures associated with BrdU+ nuclei in the MBH (16d post BrdU) displayed co-localization with GPR10 puncta (Fig. 1M,N) suggesting that majority of adult-generated hypothalamic neurons express this PrRP receptor. Besides GPR10, we co-localized neuronal cytoskeleton structures with NPFFR2 in the MBH (Fig. 1O,P), which corresponds to its previously described expression in the hypothalamus (Zhang et al, 2021). Finally, we detected mRNA for PrRP and GPR10 in MBH and observed that HFD + LiPR treatment had a statistically significantly effect on their expression, increasing expression of Prlh compared to both Control and HFD (Fig. 1Q). Taken together, these results suggest

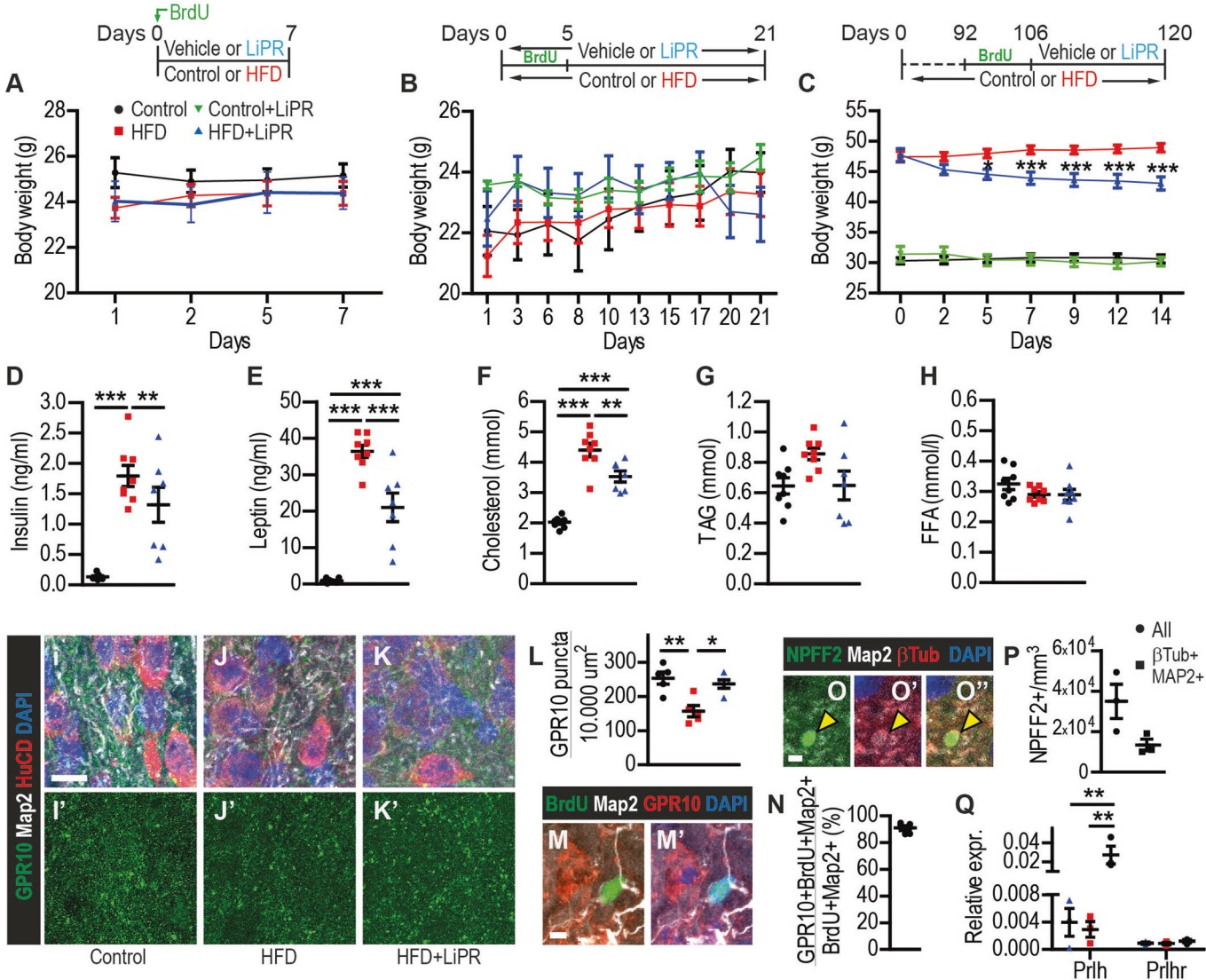

**Figure 1. Effects of LiPR on body weight, plasma metabolites and expression of PrRP and its receptors.**

(A–C) Mouse body weigh over time for the 7d (A), 21d (B) and 4mo (C) HFD protocols, which are summarized in the schematics above panels. (D–H) Plasma concentration of selected hormones and metabolites from 4mo HFD group. (I–K) Representative confocal microphotographs of MBH stained as indicated for GPR10 and neuronal markers from 21d HFD group in Control (I), HFD (J) and HFD + LiPR (K) mice. (L) Quantification of GPR10+ puncta in MBH of 21d group. (M) A representative image of GPR10+ puncta associated with Map2+ processes around a BrdU+ nucleus in ME. (N) Quantification of the proportion of BrdU+Map2 + GPR10+ cells in MBH. (O) A representative confocal image of a NPFF2 punctum (marked by arrowhead) in MBH stained as indicated from a control animal of 21d HFD group. (P) Quantification of NPFF2 puncta in MBH of 21d group. (Q) Relative mRNA fold change of Prlh and Prlhr compared to Gapdh in cDNA from MBH from mice 2 weeks on Control diet, HFD or HFD +LiPR. Data information: Scale bars (s.b.): 10 μm (I–K), 5 μm (M), 2 μm (O). $n = 5$ mice per data set in panels A,B,L,N. $n = 8$ mice per data set in panels C,D, E–H. $n = 3$ mice per data group in panels P–Q. In panels A–C, Repeated measure two-way ANOVA ((B): treatment $F_{(3, 160)} = 4.351$, $p < 0.0056$; C: treatment $F_{(3, 150)} = 130.7$, $p < 0.0001$, duration of treatment $F_{(6, 150)} = 10.42$, $p < 0.0001$, interaction $F_{(18, 150)} = 20.70$, $p < 0.0001$). In panels D–H,L,P, One-way ANOVA. In panel Q, Two-way ANOVA ($F_{(2, 12)} = 5.95$, $p = 0.016$). *$p < 0.05$, **$p < 0.01$, ***$p < 0.001$ (Bonferroni's test). Data are presented as mean ± SEM. Source data are available online for this figure.

that LiPR reverses the negative effects of HFD on plasma levels of selected hormones and metabolites and that PrRP receptors localize with adult-generated cells in the MBH.

## LiPR rescues HFD-induced decrease in number of htNSCs

Next, we determined whether LiPR treatment influenced tanycytes (Fig. 2A–C), which form heterogeneous populations in the HVZ with different proliferation and stem cell potential and are classified as α-tanycytes of the lateral walls of 3 V and β-tanycytes of the Medial Eminence (ME) (Chen et al, 2017; Yoo and Blackshaw, 2018). Treatment and its duration had a statistically very significant effect on the number of GFAP+ α-tanycytes, which are self-renewing and predominantly gliogenic (Robins et al, 2013). Mice exposed to HFD+LiPR for 7 days had higher number of GFAP+ α-tanycytes than Control or HFD mice (Fig. 2D). While the absolute number of all GFAP+ tanycytes was not changed in any protocol between any groups (data not shown),

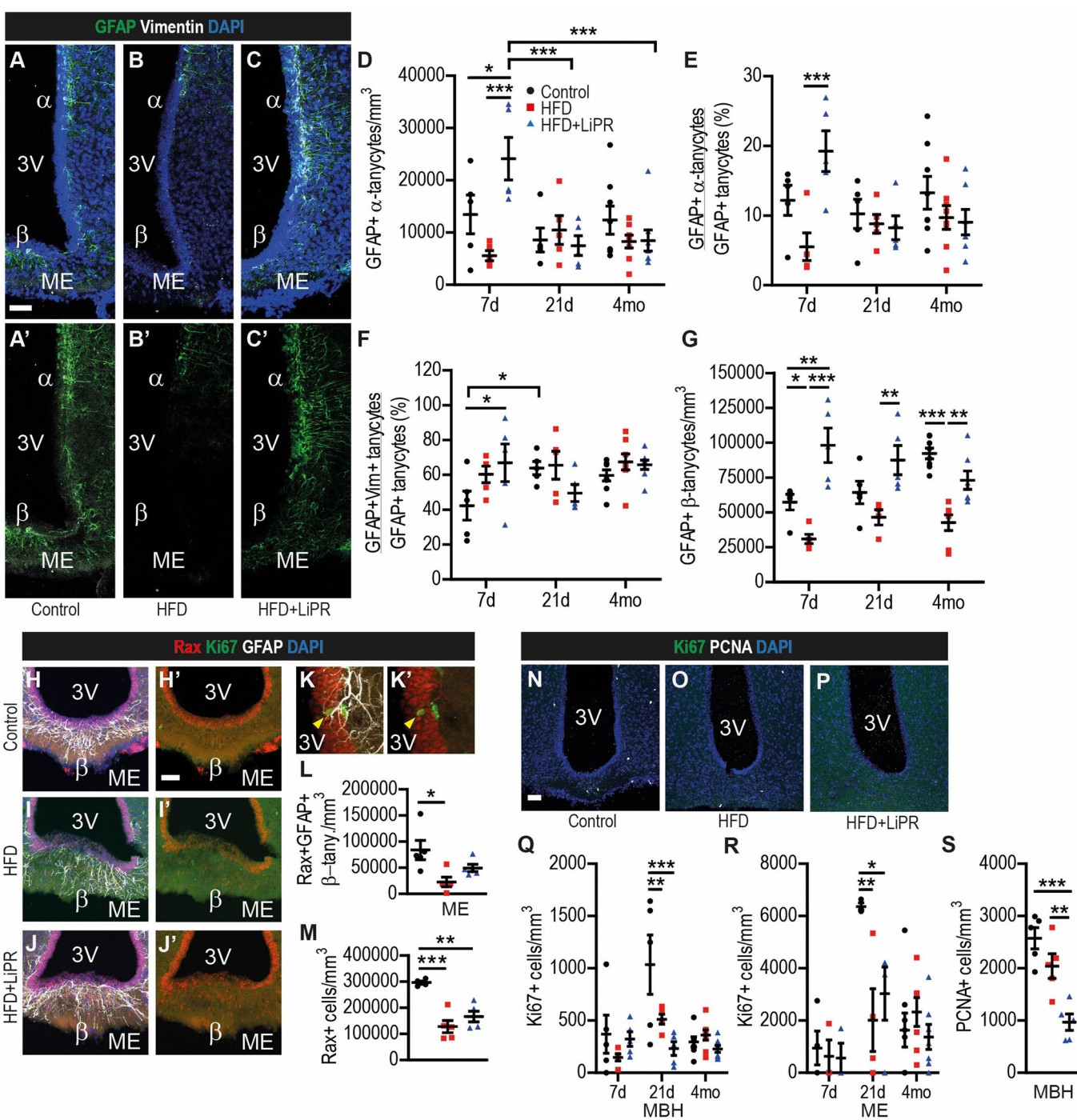

**Figure 2. Effects of LiPR on tanycytes and proliferating cells in the MBH.**

(A–C) Representative confocal images of HVZ with 3rd Ventricle (3V) and Medial Eminence (ME) stained as indicated in Control (A), HFD (B) and HFD + LiPR (C) of 7d HFD group. (D) Quantification of GFAP+ α-tanycytes per volume of MBH. (E) Proportional analysis of GFAP+ α-tanycytes in MBH. (F) Proportional analysis of GFAP+Vimentin+ tanycytes in MBH. (G) Quantification of GFAP+ β-tanycytes per volume of ME. (H–J) Representative images of MBH stained as indicated (21d group). (K) Representative image of Rax+Ki67+GFAP+ tanycyte (yellow arrowhead). (L) Quantification of Rax+GFAP+ β-tanycytes per volume of ME (21d group). (M) Quantification of Rax+ cells per volume of the HVZ (21d group). (N–P) Representative images of the MBH stained as indicated (21d group). (Q–S) Quantification of cells positive for Ki67 in MBH (Q) or in ME for 21d (R) and for PCNA (S). Data information: Scale bars: 10 μm (K), 50 μm (all other). In all panels but panel S, $n = 5$ mice per data set for 7d and 21d groups, $n = 8$ mice per data set for 4mo group. In panel S, $n = 3$–8 mice per data set. In panels **D–G**, Two-way ANOVA (D: treatment $F_{(2,45)} = 3.28$, $p = 0.047$, duration of treatment $F_{(2,45)} = 3.76$, $p = 0.031$, interaction $F_{(4,45)} = 5.29$, $p = 0.0014$; E: treatment $F_{(2,44)} = 3.58$, $p = 0.036$, interaction $F_{(4,44)} = 3.88$, $p < 0.009$; F: interaction $F_{(4,45)} = 2.61$, $p = 0.048$; G: treatment $F_{(2,42)} = 31.3$, $p < 0.0001$, interaction $F_{(4,42)} = 5.79$, $p = 0.0008$; Q: treatment $F_{(2,44)} = 6.59$, $p < 0.0031$, duration $F_{(2,44)} = 7.78$, $p = 0.0013$, interaction $F_{(4,44)} = 4.63$, $p = 0.0033$; R: treatment $F_{(2,34)} = 2.94$, $p < 0.066$, duration $F_{(2,34)} = 10.43$, $p = 0.0003$, interaction $F_{(4,34)} = 3.07$, $p = 0.029$. In other panels (**L,M,S**), One-way ANOVA (L: $F_{(2,14)} = 6.04$), $p = 0.015$; M: $F_{(2,13)} = 18.81$, $p = 0.0003$; S: $F_{(2,14)} = 16.66$, $p = 0.0003$). *$p < 0.05$, **$p < 0.01$, ***$p < 0.001$ (Bonferroni's test). Data are presented as mean ± SEM. Source data are available online for this figure.

HFD+LiPR treatment increased the proportion of GFAP+ α-tanycytes (Fig. 2E). Next, we quantified the number β-tanycytes, which show extended neurogenic capacity (Haan et al, 2013), and tanycytes expressing Vimentin, which is critical for nutrient transport and metabolic response to HFD in cells (Holmes et al, 2016; Kim et al, 2021; Roh and Yoo, 2021; Yoo and Blackshaw, 2018) (Fig. 2F). While treatment had no effect on the absolute number of GFAP+Vimentin+ tanycytes (data not shown), the interaction between treatment and its duration had a statistically significant effect on the proportion of GFAP+Vimentin+ tanycytes, increasing their proportion at 7d (Fig. 2F). Importantly, LiPR rescued the decrease in number of GFAP+ β-tanycytes driven by HFD in all three time points (Fig. 2G) suggesting that LiPR promotes the pool of reservoir tanycytes (Goodman et al, 2020). Taken together, these results suggest that LiPR treatment in the context HFD increases the number and density of tanycytes as htNSCs. Because the increase in number tanycytes could be due to increased cell proliferation, we stained brain sections for a marker of cell proliferation, Ki67 (Kempermann et al, 2004; Ohira, 2022), and the universal marker of tanycytes, the Retina and anterior neural fold homeobox transcription factor, Rax (Fig. 2H-J) (Miranda-Angulo et al, 2014; Yoo and Blackshaw, 2018). Unfortunately, we observed proliferating Ki67+Rax+ tanycytes so rarely (Fig. 2K) that it did not allow reliable quantification. However, we observed that the treatment significantly affects the number of Rax+GFAP+ β-tanycytes (Fig. 2L) and the total number of Rax+ tanycytes (Fig. 2M). Specifically, HFD reduces their number, which is not rescued by LiPR. This suggests that LiPR increases number of GFAP+ tanycytes but not all tanycytes positive by default for Rax. However, these effects of LiPR occurred in mice exposed to the HFD. To determine HFD-independent effects of LiPR, we administered it in the context of Control diet (Fig. EV1A,B) and observed that LiPR increased the number of GFAP+ α-tanycytes (Fig. EV1E) but not the proportion of GFAP+Vimentin+ tanycytes or number of GFAP+ β-tanycytes (Fig. EV1F,G). These results suggest that some but not all effects of LiPR on tanycytes are independent of HFD. Finally, to test whether effects on tanycytes are a shared feature of anti-obesity treatment, we administered Liraglutide to mice on HFD (Fig. EV1C,D). Liraglutide treatment in the 7d and 21d protocols did not change the number of GFAP+ α-tanycytes or proportion of GFAP+Vimentin+ tanycytes (Fig. EV1H,I). However, like LiPR, Liraglutide rescued the HFD-driven decrease in number of GFAP+ β-tanycytes (Fig. EV1J). These results suggest that the two anti-obesity compounds promote the population of β-tanycytes as neurogenic htNSCs (Goodman et al, 2020).

## Anti-obesity compounds reduce proliferation in the MBH

To determine if LiPR alters cell proliferation in the MBH (Fig. 2N–S), we quantified cells expressing Ki67 or Proliferating Cell Nuclear Antigen (PCNA) (Mandyam et al, 2007) in the MBH and in the ME. Treatment and its duration had a statistically very significant effects on number of Ki67+ cells in the MBH, however, LiPR did not change the reduced cell proliferation by HFD (Fig. 2Q). Similarly, treatment exerted statistically a very strong trend on number of Ki67+ cells in the ME (Fig. 2R). At the 21d time point, LiPR further reduced the number of PCNA+ cells in the MBH, which was already diminished by HFD (Fig. 2S),

suggesting that monitoring of cell proliferation in the MBH depends on proliferation markers. These anti-proliferative effects of LiPR were also maintained in the condition of Control Diet (Fig. EV1K,L), where LiPR reduced number of Ki67+ cells both in the MBH (Fig. EV1M) and the ME (Fig. EV1N) especially in the 21d group. As with the analysis of tanycytes, we wanted to address if these effects on cell proliferation are shared among anti-obesity compounds. Indeed, administration of Liraglutide (Fig. EV1O–P) also very significantly decreased the number of both Ki67+ cells both in the MBH and the ME, suggesting that different anti-obesity compounds exert similar effects on hypothalamic proliferation in the context of HFD.

## LiPR and Liraglutide show different effects on adult neurogenesis in the SGZ

Because obesity was found to alter adult neurogenesis in the SGZ (Bracke et al, 2019), we determined effects of HFD, LiPR and Liraglutide in the hippocampal neurogenic niche (Fig. EV2). LiPR treatment and its duration had statistically extremely significant effects on reduction of Ki67+ cells in the SGZ, with the 21d HFD showing reduction in proliferation, which was not rescued by LiPR (Fig. EV2E). Similarly, number of PCNA+ cells in the SGZ was extremely affected by LiPR treatment as well (Fig. EV2F). In contrast to reduced proliferation, treatment with LiPR and its duration had a significant effect on number of DCX+ neuroblasts (Fig. EV2G) and DCX+ neurons (Fig. EV2H) specifically rescuing the HFD-induced reduction of their number in the 21d protocol. Analysis of Liraglutide-treated mice revealed difference between these two anti-obesity compounds. In contrast to LiPR, Liraglutide rescued HFD-induced reduction in Ki67+ cells in 21d (Fig. EV2I), however, failed to rescue the HFD-induced decrease in number of DCX+ neuroblasts (Fig. EV2J) and even further reduced the number of DCX+ neurons (Fig. EV2K). These results suggest that LiPR reduces proliferation but increases neuronal differentiation in the SGZ, whereas Liraglutide has the opposite effects. This also suggests that LiPR can rescue neuronal differentiation reduced by the HFD in the SGZ and that, in both neurogenic niches, it reduces cell proliferation.

## LiPR treatment generates smaller MBH-derived neurospheres

The pool of proliferating cells in the MBH consists of not only proliferating htNSCs but also neural and glial precursors (Sharif et al, 2021). To observe the effects of LiPR on proliferation and cell cycle properties of htNSC, we treated mice with Control Diet or HFD with or without concurrent administration of LiPR for 14 days. After the treatment, primary htNSCs were grown as globular syncytia, so-called neurospheres (Li et al, 2012; Petrik et al, 2018), or as adherent cell cultures followed by time-lapse imaging (Petrik et al, 2018). MBH-derived neurospheres (Fig. 3A–C) from HFD-treated mice showed no difference in their diameter after 5d (Fig. 3D,E) or 10d (Fig. 3F,G) when compared with Controls. However, neurospheres from animals exposed to HFD + LiPR had smaller diameter and showed size distribution skewed towards smaller neurospheres at both 5d and 10d. While the LiPR-treated neurospheres were smaller, there was no difference in the number of neurospheres in any of the treatment groups (Fig. EV3A,B).

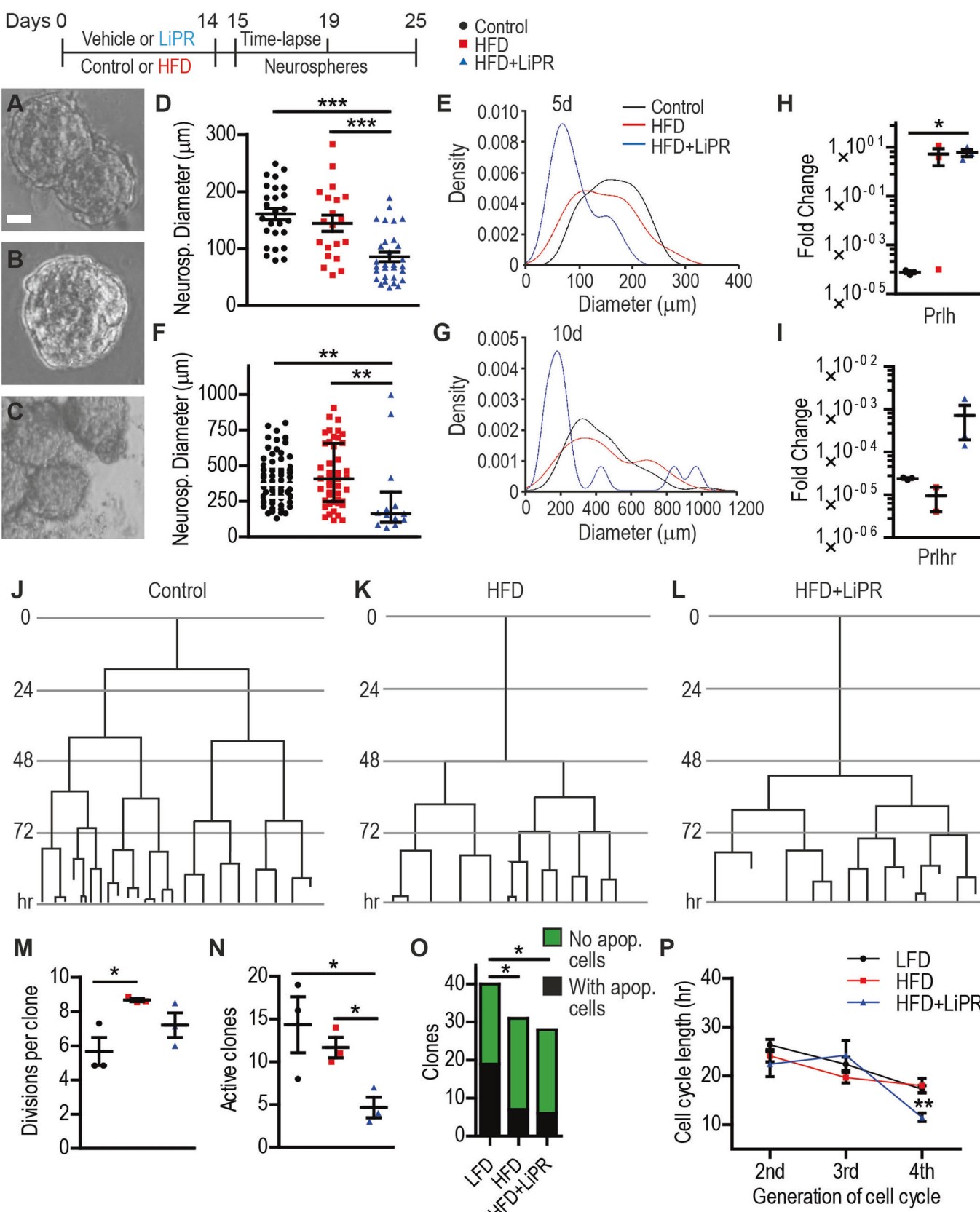

◄   **Figure 3.   LiPR reduces cell activation and proliferation of htNSCs in vitro.**

(A–C) Representative images of HVZ-derived neurospheres 5d in culture from Control (A), HFD (B) and HFD + LiPR (C) treated mice. Schematics of the experimental protocol shown above. (D,F) Quantification of diameter of neurospheres 5d (D) and 10d (F) in culture. (E,G) Kernel density plots of neurosphere diameter frequency distribution as a function of diameter for 5d (E) and 10d (G) in culture. (H,I) Relative mRNA fold change of Prlh (H) and Prlhr (I) compared to Gapdh in cDNA from MBH-derived neurospheres (10d in culture). (J–L) Example cell division trees from 4-day time-lapse imaging of aNSCs from HVZ of Control (J), HFD (K), HFD + LiPR (L) mice. (M) Quantification of the number of cell divisions per division tree clone. (N) Time-lapse quantification of active (dividing) clones per 20,000 plated cells. (O) Proportion of active clones containing at least one apoptotic cell. (P) Cell cycle length from the time-lapse imaging over observed cell divisions. Data information: Scale bars: 20 μm. In all panels, n = 3 mice for each data set. In panel D, number of neurospheres: n = 26 (Control), 20 (HFD), 30 (HFD+LiPR). In panel E, number of neurospheres: n = 66 (Control), 46 (HFD), 13 (HFD+LiPR). In panels M–P, number of traced clones: n = 48 (Control), 68 (HFD), 53 (HFD+LiPR). Data information: In panel O, Chi-square test. In panel F, Kruskal–Wallis test with Dunn's test (H = 10.30, p = 0.0058). In panels H,N,O,P, un-paired two-tailed T-Test. In other panels, One-way ANOVA with Bonferroni's test (D: F(2,73) = 15.80, p < 0.0001; M: F(2,6) = 5.61, p = 0.042). *p < 0.05, **p < 0.01, ***p < 0.001. Data are presented as median ± IQR (F) or mean ± SEM (all other panels). Source data are available online for this figure.

Interestingly, neurospheres 5d in culture had smaller diameter also from animals treated with LiPR but kept on Control diet (Fig EV3C–F). In vivo treatment with LiPR followed by 10 days in culture resulted in higher qPCR expression of Prlh, a gene that encodes PrRP, in context of HFD (Fig. 3H) and Control diet (Fig, EV3G) but not in changes of expression of Prlhr, encoding GPR10 receptor (Fig. 3I) or Vimentin (Fig, EV3I). These results suggest that LiPR reduces cell proliferation independently of diet both in vivo and in vitro and increases PrRP expression in MBH-derived stem cells and progenitors.

## LiPR affects cell activation and cell cycle of htNSCs

To further understand effects of LiPR on cell cycle and proliferation, we performed time-lapse imaging in vitro. Individual htNSCs were continuously imaged to obtain cell division trees from the post-hoc analysis (Fig. 3J–L). The analysis revealed that treatment significantly increases number of cell divisions per clone, which is significant for HFD but not HFD+LiPR (Fig. 3M). In addition, LiPR treatment reduced number active clones defined as clones with at least one cell division during the time-lapse imaging window (Fig. 3N). As with the neurospheres, we observed effects of LiPR also in the context of Control diet (Fig. EV3J), where it does not change number of cell divisions (Fig. EV3K) but decrease number of active clones (Fig. EV3L), an effect clearly independent of diet (Fig. EV3M). However, this diet-independent effect on cell activation is not mirrored by LiPR's effect on cell death and cell cycle. Time-lapse imaging reveals that there are fewer clones with at least one apoptotic cell in both HFD and HFD+LiPR (Fig. 3O) but not Control diet + LiPR (Fig. EV3N). Furthermore, the generation of observed cell cycle from 1st to 4th has a statistically a very significant effect on cell cycle length, eventually resulting in shorter cell cycle in HFD + LiPR cells (Fig. 3P), which is in contrast with extension of cell cycle in clones from LiPR treated mice on Control diet (Fig. EV3O). Of note, our data for the first time directly show the length of cell cycle in htNSCs (22 ± 0.78 h for 2nd–4th cell divisions), which is significantly shorter than cell cycle of aNSCs from the adult Subventricular Zone, SVZ (28.55 ± 2.14 h; un-paired, two-tailed T-Test, p = 0.0025) (Petrik et al, 2018) but longer than cycle length of aNSCs from the SGZ (16.62 ± 3.28 h; un-paired, two-tailed T-Test, p = 0.021) (Gupta et al, 2021) observed by in vitro time-lapse imaging under the same proliferating conditions. Finally, we tested whether PrRP signalling is cell intrinsic to htNSCs. Because LiPR is highly adhesive to cell culture plasticware, we added recombinant human PrRP31

(hPrRP31) protein, which is not plastic adhesive, to WT primary htNSC and determine its effects on cell dynamics and proliferation. Exposure to hPrRP31 in vitro significantly reduced number of cells per clone and cell divisions (Fig. EV4A–D) suggesting that PrRP signalling acts directly on htNSCs to reduce their proliferation. Taken together, these results suggest that LiPR and PrRP decrease activation and proliferation of htNSCs and changes their cell cycle in diet-dependent manner.

## LiPR rescues HFD-induced decrease in survival of new MBH neurons

Our next goal was to determine whether LiPR or Liraglutide change survival of adult-generated cells in the MBH. We exposed mice with three different BrdU protocols with varying length of post-BrdU chase periods and quantified the number of BrdU+ cells in the Region of Interest (ROI) that included the MBH parenchyma (Arcuate (Arc), Dorsal Medial (DMN) and Ventro-medial (VMN) Nuclei) and the ME. In the 7d and 21d HFD protocols, there was 6d and 16d chase periods, respectively (Fig. EV5). In the 7d protocol, there was no difference between any of the treatment groups in number of BrdU+ cells, neurons, or astrocytes (Fig. EV5A–G). In the 21d protocol (Fig. EV5H–K), LiPR did not change the number of cells, neurons, or astrocytes positive for BrdU in the MBH parenchyma in the context of HFD (Fig. EV5L–N). However, in the Arc, the primary nutrient and hormone sensing neuronal nucleus of MBH (Betley et al, 2013), treatment significantly influenced number of BrdU+ neurons. 21d HFD increased their number, which was reversed by LiPR or Liraglutide (Fig. EV5O). Interestingly, Liraglutide decreased the number of BrdU+ cells compared to Control. Taken together, these results suggest that Liraglutide in the context of HFD and in the MBH parenchyma and LiPR in the context of HFD (in the Arc) reduces the number of new BrdU+ cells in the MHB but only in the 21d and not 7d protocol.

While short-term exposure to HFD can alter functions of the MBH (Thaler et al, 2012), it does not result in DIO (Fig. 1A,B). To test the effects of DIO on the survival of newly generated cells in the MBH, mice exposed to 4mo HFD were administered BrdU for 2wk (on day 92) followed by 2wk chase (Fig. 1C). Exposure to HFD dramatically reduced the number of new BrdU+ neurons, which was rescued by LiPR administration (Fig. 4A–D). There was an extremely significant effect of the treatment on number of BrdU+ neurons with a statistically very significant decrease in the number of BrdU+ neurons by HFD rescued by LiPR to the level of Controls

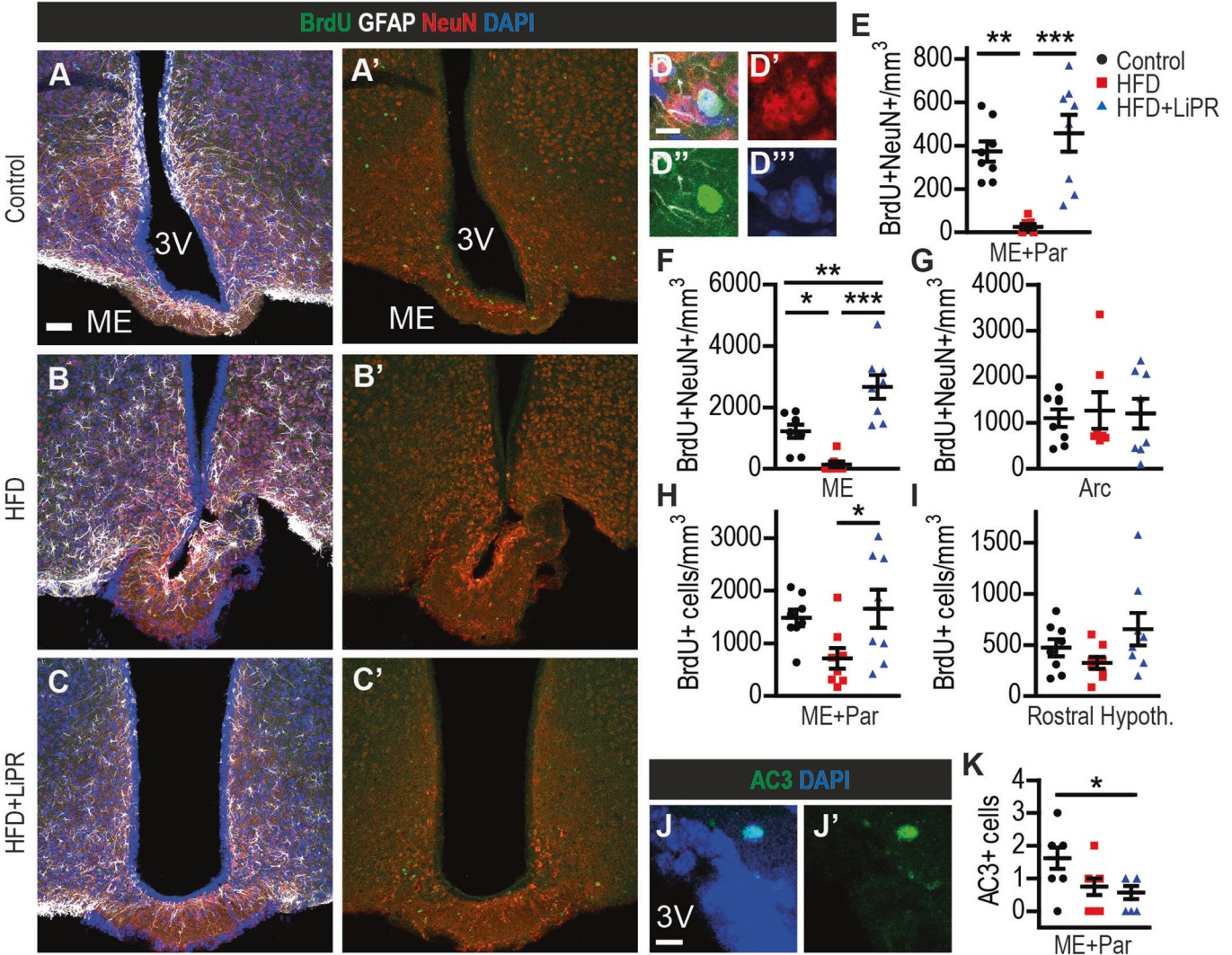

**Figure 4. LiPR improves the survival of new neurons in the MBH.**

(A–C) Representative confocal images of HVZ stained as indicated in Control (A), HFD (B) and HFD + LiPR (C) of 4 mo HFD group. (D) An example of BrdU+ neuron in the MBH parenchyma. (E–G) Quantification of BrdU+ neurons in MBH parenchyma (E), ME (F) and Arcuate Nucleus (Arc, G). H,I Quantification of all BrdU+ cells in parenchyma of MBH (H) and rostral hypothalamus (I). (J) An example of Activated Caspase 3 (AC3) positive cell near 3 V wall. (K) Quantification of AC3+ cells in MBH. Data information: Scale bars: 50 μm (A–C), 10 μm (D,J). $n = 5$ mice per data set for 7d and 21d groups and $n = 8$ mice per data set for 4mo group. Data information: In all panels, One-way ANOVA with Bonferroni's test (E: $F_{(2,20)} = 14.21$, $p < 0.0001$; F: $F_{(2,20)} = 21.37$, $p < 0.0001$; H: $F_{(2,21)} = 3.87$, $p = 0.037$; K: $F_{(2, 22)} = 2.46$, $p = 0.025$). $*p < 0.05$, $**p < 0.01$, $***p < 0.001$. Data are presented as mean ± SEM. Source data are available online for this figure.

(Fig. 4E). This statistically significant rescue of HFD-induced decrease in number of BrdU+ neurons by LiPR was observed not only in MBH parenchyma (Fig. 4E) but also in the ME (Fig. 4F) but not in Arc (Fig. 4G). This rescue effects is not limited only to new neurons but also to other BrdU+ cells (Fig. 4H) and it is confined to the MBH as there is no difference in the number of BrdU+ cells between treatment groups in the rostral hypothalamus (Fig. 4I). These results suggest that LiPR promotes survival of new MBH neurons in the context of DIO, which could be caused by reduced cell death. To test this alternative, we quantified the number of cells in the 3 V walls and MBH parenchyma positive for the Activated Caspase 3 (AC3), a marker of apoptosis (Petrik et al, 2013). Indeed, our results show a statistically significant decrease in number of AC3+ cells in HFD+LiPR compared to HFD (Fig. 4J,K). Finally,

we determined the HFD-independent effects of LiPR on new neurons in the MBH. LiPR in context of Control diet (Fig. EV5P–T) significantly increased the number of BrdU+ neurons but decreased the number of astrocytes (Fig. EV5S–T) suggesting it promotes adult neurogenesis in the MBH of older mice that are not experiencing DIO.

## LiPR shows no effects on astrocytes

Because obesity increases reactivity of glial cells (Thaler et al, 2012; Valdearcos et al, 2017), we determined the effects of HFD and LiPR on astrocytes. In the 4mo HFD protocol, there was a statistically extremely significant effects of HFD exposure on increasing the number of GFAP+ astrocytes in the MBH, which

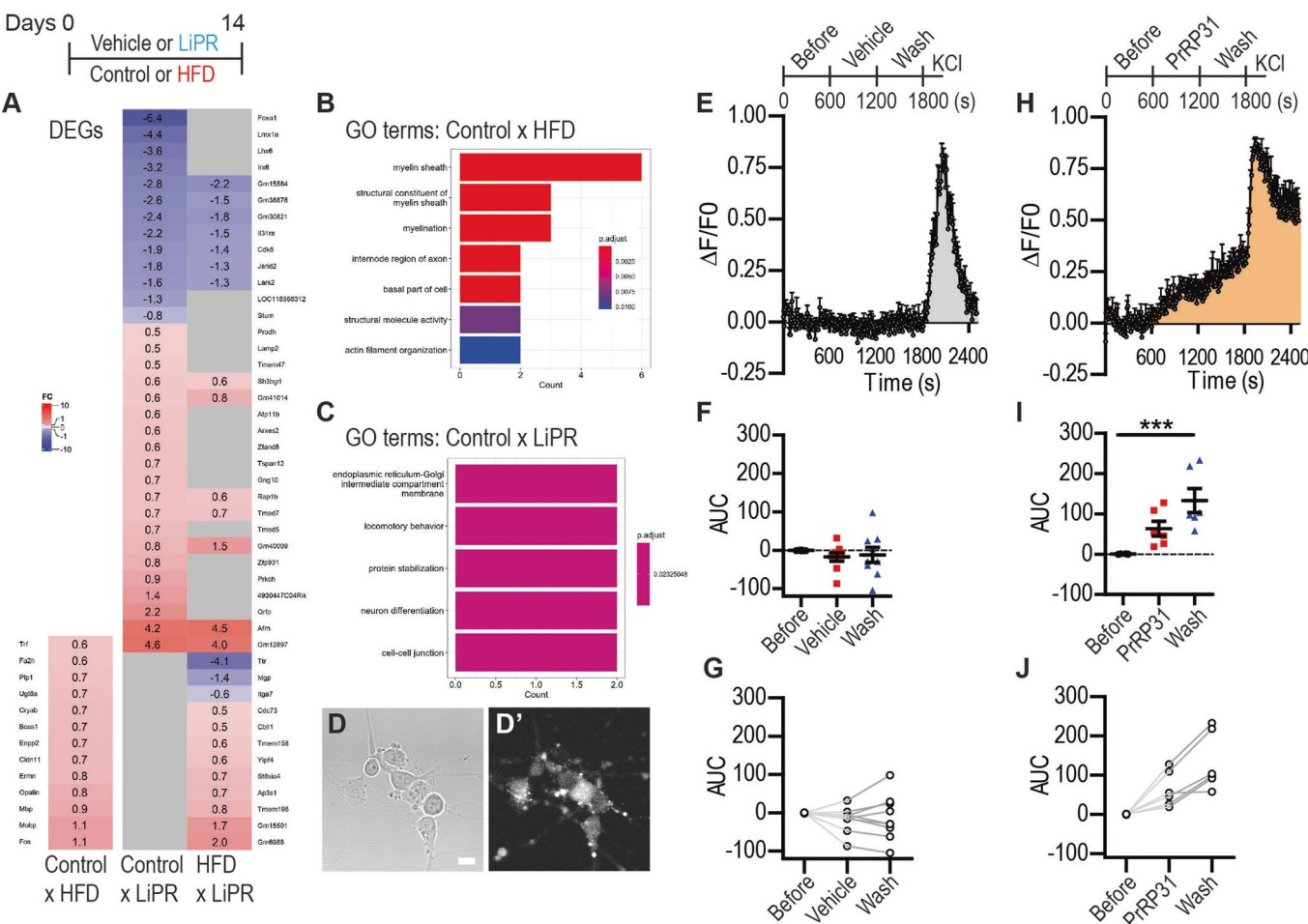

**Figure 5. Effects of LiPR on gene expression in the MBH and human iPSC-derived hypothalamic neurons.**

(A) A table of Differentially Expressed Genes (DEGs) in three different pair-wise comparisons of bulk RNAseq data. Experimental protocol is depicted above. (B) Bulk RNAseq Gene Ontology (GO) terms for the Control Diet vs HFD comparison. (C) Bulk RNAseq GO terms for Control Diet vs Control Diet+LiPR comparison. (D) A representative image in the bright field (D) and fluorescence (D') of hiPSC-derived hypothalamic neurons loaded with the Rhod-3 AM dye and used in calcium imaging. (E,H) A graph of the relative fluorescence change ($\Delta F/F0$) of Rhod-3 as a function of time before and during hPrRP31 (or Vehicle) application in medium, wash-out and KCl positive control. (E) Neurons exposed to Vehicle. (H) Neurons responding to hPrRP31. (F,I) A summary of fluorescence before (0–600 s), during hPrRP31 (F) or Vehicle (I, 600–1200 s) or wash-out (1200–1800 s) in area under curve (AUC). (G,J) Individual neurons from calcium imaging in the before-after plot from the Vehicle (G) and hPrRP31-responsive (J) groups. Data information: Scale bar: 10 µm. In panels A–C, n = 3–4 mice per data set. In panels E–G, n = 9 recorded cells. In panels H–J, n = 6 recorded cells. Data information: One-way ANOVA (I: $F_{(2,15)} = 10.90$, $p = 0.0012$). ***$p < 0.001$ (Bonferroni's test). Data are presented as mean ± SEM. Source data are available online for this figure.

was not altered by LiPR (Appendix Fig. S1A–D). Similarly, HFD increases the optical density of GFAP in the MBH independently of LiPR (Appendix Fig. S1E). However, this HFD-induced increase in reactive astrogliosis is not mirrored by an increase in astrogliogenesis as there are no effects of HFD or LiPR on number of new BrdU+ astrocytes in the MBH (Appendix Fig. S1F,G), suggesting that LiPR does not alter either reactive astrogliosis or astrogliogenesis and its pro-survival effects are confined to new neurons (Fig. 4).

## LiPR administration alters gene expression in the context of diet

To elucidate molecular mechanisms of action of LiPR, we performed bulk RNA sequencing (RNAseq) of the MBH. This brain region was chosen

because it contains the HVZ and the hypothalamic parenchyma, where we observed the changes in number of htNSCs and proliferating and surviving adult-generated cells. In addition, the MBH has a high density of cells expressing PrRP receptors (Fig. 1L–Q). The MBH tissue was sampled from mice exposed to Control or HFD with concurrent LiPR administration for 14 days (Fig. 5). Analysis of Differentially Expressed Genes (DEGs, Fig. 5A) revealed that LiPR administration changes the expression of genes involved in cell cycle regulation (e.g., Cdk8, Cdc73, Jarid2) or in function of the Golgi apparatus (e.g., Ap31s, Tmed7, Yip4), supporting the time-lapse results that suggest LiPR influences cell cycle and activation of htNSCs. In addition, LiPR changed the expression of individual DEGs with unique functions in insulin expression (Lmx1a), food intake (Qrfp), tissue inflammation (Il31ra) or thyroid hormone and vitamin dynamics (e.g., Afm, Ttr). These genes are involved in controlling metabolism and energy homeostasis. Gene Ontology (GO) analysis of

genes with changed expression (Fig. 5B,C, Appendix Fig. S2A,B) highlighted GO terms involved in neuron differentiation, cell-cell junction, or endoplasmic reticulum-Golgi compartments, suggesting that LiPR may be involved in these processes. Finally, comparison of Control diet with HFD without LiPR administration showed GO terms strongly involved in myelination, myelin sheath and its constituents, actin filament organization and internode region of axons. Several upregulated DEGs from this comparison were involved in myelination (e.g., Mbp, Mobp, Cldn11, Ugt8a, Plp1). This suggests that HFD promotes myelination, which is, however, not affected by LiPR as there are no myelin-related DEGs in the pair comparison between Control diet or HFD and LiPR groups. RT-qPCR from bulk RNA of MBH confirmed expression changes of selected genes observed by RNAseq (Appendix Fig. S2C) and showed that HFD increases expression of Prlh, which encodes the native PrRP protein, but not Prlhr, encoding GPR10 (Appendix Fig. S2D).

## PrRP increases intracellular calcium in human hypothalamic neurons

Our final goal was to determine whether PrRP can influence physiology in neurons. This should address physiological responsiveness of the target cell population. As a model, we chose human-induced Pluripotent Stem Cell (hiPSC)-derived hypothalamic neurons (Kirwan et al, 2017; Merkle et al, 2015) and determined their responsiveness to PrRP analogues by calcium imaging. Differentiated hiPSC-derived hypothalamic neurons were loaded with Rhod-3 AM dye (Fig. 5D) and calcium-induced changes in fluorescence in individual neurons were recorded over time before and during the application of hPrRP31, which was used instead of LiPR because it is not plastic adhesive (see also Fig. EV4). In contrast to vehicle-treated control neurons (Fig. 5E–G), approximately half of neurons exposed to hPrRP31 (46%, 6 out of 13) displayed a robust and steady increase in intracellular calcium (Fig. 5H). There was a statistically significant increase in calcium-dependent fluorescence during hPrRP31 application (un-paired, two-tailed $T$-Test, $p = 0.038$) that was retained after wash-out (Bonferroni's Test, $p < 0.001$; Fig. 5I,J). This hPrRP31 effect on intracellular calcium was not present in non-responsive neurons (Appendix Fig, S3A–C), which diminished the effects when pooled together with the hPrRP31-responsive neurons (Appendix Fig. S3D–F). These results demonstrate that PrRP signalling stimulates increase in the intracellular calcium in hiPSC-derived hypothalamic neurons suggesting their direct responsiveness to PrRP and LiPR.

# Discussion

## LiPR decreases body weight and promotes expression of PrRP receptors

For the first time, our results show that anti-obesity compounds influence adult neurogenesis. We demonstrate that administration of a PrRP analogue increases the number of htNSCs, decreases cell proliferation and improves survival of new neurons. These findings suggest that LiPR affects adult neurogenesis, but we cannot claim that its anorexigenic actions are conveyed through neurogenesis. Interestingly, our results indicate that different anti-obesity compounds share some but not all effects on adult neurogenesis. In agreement with previous results (Maletinska et al, 2015; Prazienkova et al, 2017), our data demonstrate that LiPR exerts

anorexigenic effects by reducing the body weight and decreasing plasma concentrations of insulin, leptin, and cholesterol that were increased by the HFD in obese mice. This ability of LiPR to normalize metabolomic response to HFD is further highlighted by our RNAseq data. LiPR upregulated expression of genes related to energy homeostasis and development of obesity such as Qrfp and Afm (Juhasz et al, 2022; Le Solliec et al, 2022) or genes involved in the regulation of insulin or thyroid hormone expression and distribution such as Lmx1a and Ttr (German et al, 1992; Liz et al, 2020) In addition to its role in metabolism, LiPR also down-regulated expression of pro-inflammatory cytokine, Il31ra (Nemmer et al, 2021) and influenced expression of Tmed7, which controls trafficking of Toll-like receptor 4 (Trl4), a key receptor in innate immunity (Liaunardy-Jopeace et al, 2014). This suggests that LiPR plays an anti-inflammatory role previously observed in a model of neurodegeneration (Holubova et al, 2019). Our results also improve the understanding of the expression of PrRP receptors in the hypothalamus, which has been demonstrated previously but without the cellular resolution or without co-localization with specific cellular markers (Fujii et al, 1999; Ibata et al, 2000; Roland et al, 1999; Takahashi et al, 2002). Interestingly, our data show that LiPR can dynamically regulate PrRP signalling by rescuing HFD-induced decrease in GPR10 protein density in the MBH and by increasing PrRP expression both in the MBH and in MHB-derived neurospheres. This LiPR-induced increase in the expression may suggest a positive feedback of the agonist on its signalling pathway as it is the case with estrogen-induced estrogen secretion (Herbison, 2008). Or it may indicate a desensitisation of GRP10 signalling by LiPR, which would resemble insulin resistance in T2DM (DeFronzo et al, 2015). Importantly, we show for the first time that GPR10 and NPFF2 receptors are expressed in the MBH-derived neurospheres and localize in new adult-generated hypothalamic neurons suggesting that PrRP signalling can act directly on adult neurogenesis, which is what we investigated here.

## LiPR increases number of htNSCs by reducing their activation

Our results from cell quantification in the MBH and the SGZ and primary MBH-derived cell cultures jointly suggest that LiPR promotes adult neurogenesis. In the context of HFD, we show that LiPR increases the number and proportion of GFAP+ α- and β-tanycytes, which are two of the putative htNSC populations that differ in the level of quiescence, proliferation and neurogenic capacity (Haan et al, 2013; Robins et al, 2013; Yoo et al, 2020; Zhang et al, 2017). Notably, short HFD was found to decrease the number of the ventral-lateral α2-tanycytes (Yoo and Blackshaw, 2018), an effect that likely contributed to the reduction in the proportion of GFAP+ α -tanycytes in our results. However, increasing the number of GFAP+ α-tanycytes by LiPR cannot be due to their higher proportion because the absolute number of GFAP+ tanycytes does not change. On the other hand, we observed that LiPR increased the number of β-tanycytes, which can serve as the reservoir neurogenic htNSCs (Goodman et al, 2020; Haan et al, 2013), but not Rax+ tanycytes. This suggests that LiPR specifically increases a subset of GFAP+ tanycytes and not all tanycytes marked by Rax (Pak et al, 2014). However, because Rax+ β-tanycytes proliferate very rarely (Mu et al, 2021) and because Rax does not cluster with GFAP in tanycytes (Mickelsen et al, 2020), the apparent increase in the number of GFAP+ β-tanycytes by LiPR may

not be due to increased proliferation but either due to the change of their phenotype (e.g., higher expression of GFAP or more prominent GFAP+ processes), lower cell death or higher self-renewal proliferation that would increase their maintenance and number (Petrik et al, 2022). Indeed, while we observed that both LiPR and Liraglutide decrease cell proliferation in the MBH, which may be critical for their anti-obesity effects, we rarely observed any proliferating tanycytes in vivo. In addition, the time-lapse imaging showed no reduction in number of clones containing apoptotic cells from HFD+LiPR vs HFD groups and HFD+LiPR decreased rather than increased cell proliferation in vitro. This suggests that changes in number of tanycytes may be a result of altered cell activation. Indeed, our time-lapse imaging results show that LiPR decreases number of active clones and shortens cell cycle length, which could be a compensatory mechanism for the lower activation. The lower cell activation may be protective against adverse effects of the HFD, which is also reflected in the rescue of the number of β-tanycytes by LiPR. In addition, our observation that adult NSCs from different niches have different duration of the cell cycle in vitro indicates that it is intrinsically controlled. The role of LiPR in regulating cell cycle is also supported by our findings that it reduces proliferation both in the MBH and the SGZ and influences the expression of genes that control the cell cycle (e.g. Cdk8, Jarid2) (Adhikari et al, 2019; Alarcon et al, 2009) Interestingly, recent RNAseq data from mouse macrophages exposed to PrRP for only 18 h reveal DEGs related not only to neuronal differentiation but also to cell cycle or inflammation, such as Cdkn2a or Il1b (Sun et al, 2021). In addition, we show that HFD+LiPR also increases the proportion of Vimentin+GFAP+ tanycytes, a subset of α-tanycytes (Goodman and Hajihosseini, 2015), which may be protective against HFD-induced changes as Vimentin is responsive to HFD (Kim et al, 2021) and is downregulated in tanycytes of mice with a hypermetabolic phenotype (Holmes et al, 2016). Taken together, our results improve our understanding of how the HFD and the anti-obesity pharmacology influence tanycytes, which act not only as htNSCs but as metabolic regulators that control vascular permeability, leptin transport (Balland et al, 2014) and the entry of anti-obesity compounds such as Liraglutide to the hypothalamic parenchyma (Imbernon et al, 2022).

## LiPR rescues reduced survival of new MBH neurons

The neurogenic process requires not only active adult NSCs but also the survival of new adult-generated neurons. The exposure to the HFD was shown to increase or decrease hypothalamic adult neurogenesis depending on the length of HFD treatment and timing of cell birth-dating protocols (Gouaze et al, 2013; Lee et al, 2012, 2014; Li et al, 2012; McNay et al, 2012). Our results show that the short and intermediate exposure to HFD does not change the number of newly generated, BrdU+ cells, neurons, or astrocytes in the MBH parenchyma, however, it increases the number of BrdU+ neurons in the primary sensing Arc, which is reversed by the con-current administration of LiPR or Liraglutide. These findings are also in agreement with previous studies that used short and intermediate HFD protocols (Gouaze et al, 2013; Lee et al, 2014), which can induce reactive gliosis or inflammation but not obesity (Nakandakari et al, 2019; Thaler et al, 2012). In addition, our results show that while LiPR does not change the number of new cells in the MBH parenchyma, it can rescue the increased production of new neurons in the Arc in the

context of the intermediate HFD exposure. This is in contrast with the long HFD exposure that induces obesity and reduces the survival of new hypothalamic neurons both in our results and previous studies (Li et al, 2012; McNay et al, 2012). Importantly, we show that LiPR rescues the reduction in the number of new neurons caused by the long HFD exposure and increases their number also in the context of Control diet. This suggests that LiPR increases survival of new hypothalamic neurons in older but not younger mice in both physiological and pathophysiological conditions. These pro-survival effects, however, are confined to the MBH because there is no difference in the number of new cells in the rostral hypothalamus. Because DIO can increase cell death in the adult hypothalamus (McNay et al, 2012; Moraes et al, 2009), we also determined if LiPR can reduce apoptosis. Indeed, our results show that LiPR reduces the number of AC3+ cells in the MBH suggesting that the increased number of new neurons is thanks to lower apoptosis during DIO but not in physiological conditions. Alternatively, LiPR could rescue neuronal survival also by mitigating HFD-induced reactive gliosis, which contributes to the increased inflammation and cell death (Seong et al, 2019; Thaler et al, 2012). However, we did not observe that LiPR mitigated gliosis suggesting that the improved neuronal survival is a cell-intrinsic feature as glia do not likely express PrRP receptors. The neurogenic effects of LiPR are summarized in a schematic of Appendix Fig. S4.

## LiPR activates human hypothalamic neurons

To determine whether LiPR can directly activate new neurons, we performed calcium imaging. PrRP or LiPR were shown to increase intracellular calcium in cell lines expressing human GPR10 (Elhabazi et al, 2013; Karnosova et al, 2021; Roland et al, 1999) and to reduce the frequency of GABAergic miniature Inhibitory Postsynaptic Currents (mIPSCs) in murine hypothalamic neurons (Ma et al, 2009). This suggests that PrRP signalling decreases neuronal inhibition. However, to our best knowledge, physiological responsiveness to PrRP has not been demonstrated in immature or human neurons (Prazienkova et al, 2019). To fill this knowledge gap, we performed calcium imaging in hiPSC-derived neurons. We chose these neurons for two principal reasons. First, they display immature characteristics when compared to the tissue-derived counterparts (Cornacchia and Studer, 2017; Miller et al, 2013; Patterson et al, 2012), which makes them relevant as a model of immature, adult neurogenesis-generated neurons. Second, they provide relevance to human biology and medicine. Specifically, hiPSC-derived hypothalamic neurons display MBH lineage and phenotype (Kirwan et al, 2017; Merkle et al, 2015) and thus are ideal for testing responsiveness to PrRP. This is especially relevant considering that highest concentration of PrRP in the human brain was detected in the hypothalamus (Takahashi et al, 2000). Our results from calcium imaging show for the first time that PrRP can directly stimulate human iPSC-derived hypothalamic neurons. Interestingly, the calcium concentration increases even when hPrRP31 was washed-out. This might be caused by persistent signalling downstream of PrRP receptors or by internalization of the receptors. Finally, almost half of hypothalamic neurons randomly chosen for calcium imaging displayed responsiveness to PrRP. However, anorexigenic POMC-expressing neurons, which are expected to be primary targets of PrRP, represent less than 10% of total hypothalamic neurons differentiated from hiPSCs (Merkle et al, 2015). This suggests that other hypothalamic neuronal subtypes may be responsive to PrRP as well.

## LiPR rescues HFD-induced decrease in SGZ neurogenesis

Because diet influences adult neurogenesis in the hippocampus (Poulose et al, 2017), we determined the effects of HFD and LiPR in the SGZ. In the context of the intermediate HFD protocol, we observed that LiPR reduces cell proliferation already lowered by the HFD, whereas it rescues reduction in number of neuroblasts and new immature neurons, which was induced by the HFD, a finding reported previously in the SGZ (Jiranugrom et al, 2020; Robison et al, 2020). These results suggest that LiPR exhibits pro-neurogenic functions in both the HVZ and the SGZ, which is in line with its other neuroprotective effects (Popelova et al, 2018). The dual effects of LiPR on two neurogenic niches are especially intriguing given the fact that the hypothalamus can modulate adult neurogenesis in the hippocampus (Li et al, 2022).

## Diet-independent effects of LiPR on hypothalamic neurogenesis

As LiPR displays a broad range of roles independent from the HFD (Maletinska et al, 2019; Mengr et al, 2021), we determined its effects on hypothalamic neurogenesis in the context of Control diet. For the Control diet + LiPR, we chose 21d and 4mo time points because they complement results for two different phases of the HFD exposure. The 21-day time-point represents pre-obesity, whereas the 4-month time-point is for obesity. In the MBH, LiPR increased the number of new, adult-generated neurons and reduced the number of adult-generated astrocytes suggesting it promotes adult neurogenesis in older mice outside the pathophysiological context of DIO and possibly skews the cell potency of htNSCs towards the neurogenic instead of gliogenic lineage. In addition, we observed that LiPR decreases cell proliferation both in vitro and in vivo, increases number of α-tanycytes and decreases activation of htNSC, which suggests that PrRP signalling increases self-maintenance of htNSCs and preserves their reserve pool. These results jointly suggest that certain LiPR effects on htNSCs, especially its anti-proliferative potential, are diet independent. Importantly, in vitro activation of PrRP signalling in naive WT htNSCs reduces cell proliferation, which supports the conclusion that LiPR actions on htNSCs are cell intrinsic.

## Liraglutide shares some neurogenic effects with LiPR

Given the effects of LiPR on adult neurogenesis, we were intrigued if anti-obesity compounds share the neurogenic potential as many anti-depressants do (Eisch and Petrik, 2012; Malberg et al, 2000). Contrary to our expectation, our results show some differences between LiPR and Liraglutide, a compound already approved for human medicine, which is indicated for T2DM and has anti-obesity activity (Bray et al, 2016; van Can et al, 2014). Both compounds reduce cell proliferation and increase the number of β-tanycytes in the MBH suggesting that these cellular effects are important for their anti-obesity action. In contrast, Liraglutide does not rescue HFD-driven reduction in the number of neuroblasts and immature neurons in the SGZ. Instead, it further decreases their number in the context of the HFD. The difference in their action may originate in the difference of the cellular expression profile and signalling in their respective pathways. Nevertheless, this dissimilarity in neurogenic effects is intriguing because both compounds share

other functions such as neuroprotection (Holubova et al, 2019). More importantly, the positive effects of LiPR on adult neurogenesis warrant further research because healthy neurogenesis is essential for many brain functions such as learning and mood control (Eisch and Petrik, 2012). The future work should determine whether specific ablation of adult neurogenesis in the hypothalamus would diminish anorexigenic effects of anti-obesity compounds along the lines of previous causative evidence that anti-depressants require neurogenesis in the hippocampus for their action (Santarelli et al, 2003).

# Methods

## Mice

Animal experiments in this study were performed either in the United Kingdom (UK) or in the Czech Republic as it is specified below. All experimental procedures in the in UK were performed in accordance with the United Kingdom's the Animals (Scientific Procedures) Act 1986 (ASPA) and were approved by the UK Home Office (HO) and the Biological Services of the College of Biomedical and Life Sciences, Cardiff University. All mice were 8–9 weeks old at the start of the regulated procedures. All mice were C57BL/6j males purchased from a HO accredited vendor, Charles River, UK. After transfer to our animal facility, mice were acclimatized for a minimum of 5 days before entering a regulated procedure. Mice were housed at a constant 22 °C, a relative humidity of 45–65% and in a 12:12 h light-dark cycle and group housed (3–5 mice per group) with ad libitum access to water and diet. Mice used for the in vitro experiments, time-lapse imaging and RNAseq were treated in the UK.

In the Czech Republic, all animal experiments followed the ethical guidelines for animal experiments and the Act of the Czech Republic Nr. 246/1992 and were approved by the Committee for Experiments with Laboratory Animals of the Academy of Sciences of the Czech Republic. C57BL/6 male mice that were 4 weeks old were obtained from Charles River Laboratories (Sulzfeld, Germany). The mice were housed under controlled conditions at a constant temperature of $22 \pm 2$°C, a relative humidity of 45–65% and a fixed daylight cycle (6 am–6 pm), with 5 mice per cage. The animals were provided free access to water and the standard rodent chow diet. The 7d, 21d and 4mo protocols with BrdU treatment were conducted in the Czech Republic.

## Diets

Mice were given either Control Diet (Low Fat Diet, LFD) or High Fat Diet (HFD) in form of hard pellets. HFD was custom-made in the laboratory based on (Maletinska et al, 2015) and had energy content of 5.3 kcal/g, with 13%, 60% and 27% of calories derived from proteins, fats, and carbohydrates, respectively. This HFD is based on Ssniff Germany Diet (Ssniff® R/M-H #V1530-000 RM-Haltung, Mehl (ground/powder), Ssniff Spezialdiäten GmbH, Soest, Germany), which is the basis of the Control (LFD) pellet diet (#V1535 RM-Haltung, 15 mm pellets, with 24%, 7%, 67% of calories derived from proteins, fat, and carbohydrates, respectively). HFD composition was 400 g of Ssniff base, 340 g of powdered cow-milk-based human baby formula (Sunar Complex 3, Dr Max

Pharmacy, Mirakl, a.s., Bratislava, Slovakia), 10 g of Corn Starch and 250 g of pork lard (Lidl UK, Surbiton, UK). Ingredients were thoroughly mixed by a table-top kitchen robot, made into pellets and stored at −80 °C. Before there were given to animals, pellets were warmed to room temperature. Both control and HFD pellets were changed twice per week.

## Prolactin releasing peptide and its lipidized analogue

Full-length, 31 amino acid long human Prolactin Releasing Peptide (hPrRP31), SRTHRHSMEIRTPDINPAWYASRGIRPVGRF-NH$_2$ was used in selected in vitro experiments as described below. Purified, desicated, crystal hPrRP31 (M.W. = 3665.15) was stored at −20 °C. Stock solution of 1 mM hPrRP31 in water was added to cell media to the final concentration of 1 μM. A human palmitoylated analogue of PrRP (LiPR) with the sequence SRTHRHSMEIK (N-γ-E(N-palmitoyl)) TPDINPAWYASRGIRPVGRF-NH$_2$ was synthesized and purified at the Institute of Organic Chemistry and Biochemistry, Czech Academy of Sciences, Prague (CAS), Czech Republic, as previously described (Prazienkova et al, 2017). Purified, desicated, crystal LiPR (M.W. = 4004) was stored at −20 °C until it was reconstituted in the sterile physiological solution (0.9% NaCl, pH = 7.4) as 1 mg/ml concentration (corresponding to ~0.25 mM). This working stock solution was aliquoted to the protein LoBind tubes (Eppendorf, Stevenage, UK) and kept at −20 °C for up to 2 weeks. Each day, an aliquot was thawed before administration to animals. Mice received a single subcutaneuos (s.c.) injection per day of 5 mg LiPR/kg of body weight before the onset of the dark cycle. As a control, LiPR vehicle was injected s.c.

## Liraglutide

Glucagon Like Peptide 1 (GPL-1) analogue Liraglutide (Saxenda, PubChem CID: 16134956, (van Can et al, 2014) was obtained from Pharmacy (Victoza®, Novo Nordisk A/S, Bagsværd, Denmark). Liraglutide was administered in the physiological solution (40 μg/ml) s.c. daily in 0.2 mg/kg concentration.

## Determination of hormonal and biochemical parameters

Blood was sampled from tail vein of overnight fasted animals and the blood plasma was separated and stored at −20 °C. The plasma insulin concentrations were measured using an RIA assay (Millipore, St. Charles, MI, USA). Leptin concentrations were determined by ELISA (Millipore, St. Charles, MI, USA). The blood glucose levels were measured using a Glucocard glucometer (Arkray, Kyoto, Japan). The plasma triglyceride levels were measured using a quantitative enzymatic reaction (Sigma, St. Louis, MO, USA). The free fatty acids (FFA) levels were determined using a colorimetric assay (Roche, Mannheim, Germany). Cholesterol was determined by colorimetric assay (Erba Lachema, Brno, Czech Republic). All measurements were performed according to the manufacturer's instructions.

## BrdU

Animals were given bromodeoxyuridine, BrdU (Sigma-Aldrich, Gillingham, UK) either by intraperitoneal (i.p.) injections or in drinking water. For i.p. injections, BrdU (10 mg/ml) was dissolved in the physiological solution (0.9% NaCl, pH = 7.4) and injected as 150 mg/kg of body weigh (Petrik et al, 2012). Animals received 2

i.p. injections of BrdU 6 h apart in the 7-day HFD protocol (see below). In drinking water, BrdU (1 mg/ml) was dissolved in sterile tap water with 1% sucrose (Petrik et al, 2018). Animals were administered BrdU drinking water for 5 or 14 days (see the HFD protocols below). BrdU drinking water was protected against photolysis and changed every 3 days.

## HFD protocols

Mice were exposed to four different protocols. In the short protocol, they were given control or HFD with concurrent daily s.c. injections of LiPR or Liraglutide for 7 days and BrdU i.p. injections on day 1 and culled on day 7 (BrdU chase period of 6 days). In the intermediate protocol, they were exposed to 21 days of control or HFD with concurrent daily s.c. injections of LiPR or Liraglutide and administration of BrdU in the first 5 days and culled on day 21 (BrdU chase of 16 days). In the long protocol, mice were exposed to control or HFD for 4 months. On day 92, they were given BrdU drinking water for 14 days followed by daily s.c. LiPR injections for 14 days and culled on day 120 (BrdU chase of 14 days). In these 3 protocols, mice were culled by transcardial perfusion with brains used for immunohistochemistry. Finally, mice were exposed to 14 days of control or HFD with concurrent daily s.c. injections of LiPR and culled by cervical dislocation with brains used to prepare primary cell cultures or to isolate RNA for RNA sequencing. In the 14-day and 21-day protocols, LiPR was also administered to animals on Control (LFD) diet.

## Immunohistochemistry

Animals were transcardially perfused with ice-cold PBS followed by perfusion with ice-cold 4% paraformaldehyde. The brains were isolated and postfixed in 4% PFA for overnight. The PFA was replaced by 30% sucrose and brains let to sink at room temperature. 40 μm coronal sections (spanning entire hippocampus) were cut in serial sets of 12 for stereological evaluation. Slide-mounted or free-floating immunohistochemistry (IHC) was performed. The following primary antibodies were used: rabbit polyclonal anti-GPR10 receptor (1:200; PA5-29809 Thermo Scientific, Paisley, UK; RRID:AB_2547283); rabbit polyclonal anti-NPFF2 receptor (1:200; ab1420 AbCam, Cambridge, UK); chicken polyclonal anti-MAP2 (1:400; ab92434 AbCam; RRID:AB_2138147); mouse monoclonal anti-HuC/HuD (1:500; A21271 Thermo; RRID:AB_221448); mouse monoclonal anti-NeuN (1:500; MAB377 Thermo; RRID:AB_2298772); mouse monoclonal anti-beta-III-Tubulin (1:400; ab14545 AbCam); rabbit polyclonal anti-GFAP (1:500; Z0334 Agilent/Dako, Stockport, UK; RRID:AB_10013382); goat polyclonal anti-Vimentin (1:50; AB1620 Chemicon, Watford, UK); mouse monoclonal anti-Ki67 (1:400; CPCA-Ki67-100ul EnCor Biotechnology/2BScientific, Upper Heyford, UK; RRID:AB_2637049); mouse monoclonal anti-PCNA (1:300; M087901-2 Agilent); guinea pig polyclonal anti-Doublecortin (DCX; 1:400; AB2253 Thermo; RRID:AB_1586992); rat monoclonal anti-BrdU (1:400; MCA6144 BioRad, Watford, UK); guinea pig polyclonal anti-RAX (1:200; M229 Takara Bio Europe Ab, Goteborg, Sweden; RRID:AB_2783559); rabbit polyclonal anti-activated caspase 3 (AC3; 1:500; 9661S Cell Signalling, Leiden, The Netherlands; RRID:AB_2341188). Blocking with the carrier (1× PBS, 0.5% Triton X-100, 5% normal donkey serum or 2% bovine

serum albumin) for a minimum of 20 minutes at room temperature (RT) was followed by incubation with the primary antibodies at RT overnight. After three washes, the secondary antibodies were incubated for 1.5 hours at RT. Sections were counterstained with DAPI (1:1000; Roche). Slides were mounted and cover-slipped using ProLong Diamond antifade mountant (Thermo). For BrdU staining, brain sections were incubated with primary antibodies for GFAP and HuC/HuD or NeuN and followed by their respective antibodies as described above. After that, the sections were post-fixed in 4% PFA in PBS for 15 min in RT and treated with 2 N HCl (20 min in RT) followed by 0.1 M sodium tetraborate (pH = 8.5; 10 min in RT). After equilibration in PBS, primary antibody for BrdU was applied overnight in 4 ºC, followed by three washes, and the secondary antibody.

## Stereologic and proportional cell analyses

Quantification of marker-positive cells in the Medial Basal Hypothalamus (MBH) and the Subgranural Zone (SGZ) of the hippocampus was performed stereologically as described before (Petrik et al, 2012). All coronal sections containing the hippocampus (in general, from bregma −0.6 to −4.0 mm) and MHB (in general, from bregma −1.2 to −2.3 mm, (Paxinos and Franklin (2004)) from one well in the series of 12 were stained as described above as tanycytes are present in coordinates from bregma −1.3 to −2.5 mm (Pasquettaz et al, 2021). Z-stack images were obtained in the ZEN Blue software (Zeiss) using 20X or 40X apochromatic objectives on the Observer.Z1 Zeiss LSM780 confocal microscope by an observer blind to the genotype of the brains. Density of GPR10-positive (GPR10+) and NPFF2+ puncta (co-localized with Map2 and βIII-Tubulin neuronal cytoskeleton structures) was quantified in the DMN of the hypothalamus and expressed per area of tissue quantified. BrdU+Map2+ neurons expressing GPR10 in the MBH (including the ME) were quantified as a proportion of all BrdU+Map2+ cells. A cell was recognized as BrdU+Map2 + GPR10+ if it had Map2+ processes around or extending from a BrdU+ nucleus and there was co-localization of GPR10+ puncta with the BrdU-associated Map2+ processes. To obtain estimates of absolute number of marker-positive cells per volume of tissue, all cells positive for one or more cell markers (e.g. BrdU, NeuN and DAPI) were quantified in brain sections containing MBH. The Region of Interest (ROI) quantified included the MBH parenchyma with the Arc, DMN and VMN and the Medial Eminence (ME). In the hippocampus, the ROI included SGZ only. The area of quantified ROI in 2–4 brain section was measured in the ZEN software and multiplied by the brain section thickness to obtain the ROI volume. The cell density was quantified as number of cells per mm cubic of tissue. Quantifications of tanycytes was done in sections containing MBH, including the number of GFAP+ α-tanycytes in the lateral-ventral 3 V wall, number of GFAP+ β-tanycytes in the ME and the proportion of GFAP+Vimentin+ tanycytes. The number of GFAP+ α-tanycytes which project from the 3 V wall into the parenchyma were quantified. α-tanycytes were defined as being located in the lateral walls of 3 V and having a cell process longer than two cell soma lengths and projected into MBH parenchyma. Similarly, GFAP+ β-tanycytes were defined as cells with soma next to the 3 V in the ME and as having a single GFAP+ cell process longer than two cell soma lengths and projected to the ME parenchyma. These GFAP+

β-tanycytes had distinctly different morphology than GFAP+ astrocytes with multiple GFAP processes radiating from soma. The number of GFAP+ α- or β-tanycytes in 1 mm³ of MBH or ME, respectively, per section was calculated as (1/volume (mm³) x number of GFAP+ α- or β-tanycytes, to allow comparison between brain sections of different sizes. Rax+GFAP+ tanycytes and Rax+ cells were quantified in the HVZ, which comprised of both ventral lateral walls of 3 V and the ME.

## Cell cultures

Primary cell cultures were prepared from the dissected walls of 3 V of adult C57BL/6j male mice exposed to control or HFD with or without concurrent LiPR administration for 14 days or from untreated mice. Cells were prepared as adherent primary cell cultures or as neurospheres. Isolated brain was washed three times in ice-cold HBSS with 10 mM HEPES. Dissection was carried out in the same medium as brainwashing. First, anterior, and posterior coronal cuts were made to isolate the hypothalamus in the rostro-caudal axis, corresponding to approximately bregma −1.0 to −4.0 mm. Then, a horizontal cut was made at the level of the thalamus to separate the cerebral cortex and to level the dorsal aspect. With the ventral aspect facing up, using microdissection surgical scissors, cuts parallel to the 3 V wall were made to isolate the 3 V walls. Finally, cust were made in the rostro-caudal axis to separate the segment containing the Hypothalamic Ventricular Zone (HVZ) corresponding approximately to bregma −1.0 to −2.5 mm. The dissected tissue was digested for 30 min in 37 ℃ with trypsin (3.5 mg/5 ml; Sigma) and hyaluronidase (3.5 mg/ 5 ml, Sigma) as described previously (Petrik et al, 2018). After gradient centrifugation, the cells were resuspended in DMEM/F12 Gluta-MAX medium (Thermo) supplemented with penicillin (100 units/ml), streptomycin (100 µg/ml, Gibco) and B27 supplement (1:50, Thermo). For neurospheres essay, the medium was also supplemented with EGF and α (both 10 ng/ml; Gibco) every 3–4 days and primary neurospheres were counted and imaged on day 5 and 10 of the culture to measure their diameter. On day 10, neurospheres were collected for RNA isolation. For primary HVZ cultures, EGF and bFGF2 (both 10 ng/ml) was added once at plating of cells. 20,000 cells per 0.5 ml of medium per well (24-well plate) were plated in PDL-coated wells. Cells were allowed to adhere overnight in a cell incubator before they were subjected to time-lapse imaging. After imaging, cell identity was determined by immuno-cytochemistry (ICC) for proteins specific to different cell types as described in greater details in the following sections.

## Time-lapse imaging

Primary HVZ cells from HFD/LiPR-treated mice were prepared as described above. Primary HVZ cells from untreated mice were exposed to LiPR in vitro for 8 days (1 µM added every day during the first 3 days of the culture). Alternatively, primary HVZ cells from naïve C57BL/6j mice were prepared and subjected to 1 µM hPrRP31 for 4 days (hPrRP31 added once per day) before the time-lapse imaging. Cells were continuously imaged as described previously (Ortega et al, 2013; Petrik et al, 2018). Briefly, cells were kept in 37 °C in 5% CO₂ and pictures were taken every 10 min for 4 days. Four imaging regions (of 2 × 2 tiles) were selected per well in the ZEN Blue software (Zeiss). Imaging was done with an

inverted 10X apochromat objective on a Zeiss Axio Observer 7 microscope equipped with an environmental chamber, motorized stage and Zeiss Definitive Focus module. After imaging, cells were stained for Vimentin, Sox2 and/or GFAP and their immunocyto-chemical phenotype was determined. Marker positive cells with elongated soma and one or two elongated processes were distinguished from rare astrocytes with stellate morphology. The cell dynamics of individual cell clones from a single seeding stem cell was analysed either in ZEN software in less mitotically active clones (up to 8 cells per clone at the end of imaging) or in the The Tracking Tool for more active clones as described previously Timm's Tracking Tool software as described previously (Hilsen-beck et al, 2016; Petrik et al, 2018).

## Immunocytochemistry

Cells were stained in 1× PBS with 2% bovine serum albumin (BSA, Sigma) and 0.5% Triton-X overnight in RT. After washout, the cells were stained with secondary antibodies (1:300) for 2 h at room temperature and nuclei stained with DAPI (1:1000, Roche, Munich, Germany). The cells were mounted using the ProLong Diamond antifade mountant (Thermo) and imaged in the ROIs identical to the time-lapse ROIs.

## RT-qPCR

Total RNA was isolated from HVZ neurospheres using the RNeasy Mini plus kit (Qiagen, Manchester, UK) according to manufac-turer's manual. RNA was re-transcribed into cDNA using Super Script III polymerase with random primers and RNAse Out (Thermo). qPCR reaction was performed using Fast SYBR Green dye (Thermo) in technical duplicates for each sample on a Step One Plus Real time PCR system (Life Technologies) to determine relative expression of selected mRNA transcripts (denaturation for 5 min at 95 °C, followed by 40 cycles of 95 °C denaturation and 60 °C annealing and elongation). The results were analysed using the AccuSEQ software (Life Technologies) with the amplification cycle (Ct) values determined by maximum 2nd derivation method and following the $2^{-\Delta\Delta Ct}$ method (Livak and Schmittgen, 2001) with normalization to the expression levels of Gapdh. The sequences of used primers (125 nM final concentration for Prlh, Prlhr and Npffr2 primers, 250 nM for all other) were designed by using the Primer-BLAST online tool (NCBI-NIH): Gapdh forward = TTCACCACCATGGAGAAGG, Gapdh reverse = CACACCCAT-CACAAACATGG (Petrik et al, 2018); Prlh forward = GCTGCTGCTAGGCTTAGTCC, Prlh reverse = ACTTGG-CACTTCCATCCAGT (Bjursell et al, 2007); Prlhr forward = ACTTCCTCATTGGCAACCTG; Prlhr reverse = TGGTGAGTGT-GAACACCGAT (Bjursell et al, 2007); Npffr2 forward = TGCCTATCACATTGCTGGAC; Npffr2 reverse = CGTGA-CAAAGGCTGTCTTGA (Bjursell et al, 2007); Vim forward = CACTAGCCGCAGCCTCTATTC, Vim reverse = GTCCACC-GAGTCTTGAAGCA (Wang et al, 2018); Ttr forward = TCCTCATTTTTCTCCCCTGCT, Ttr reverse = CGGTTGGT CCACTCTGCTTT; Il31ra forward = CCAGAAGCTGCCATGTC GAA; Il31ra reverse = TCTCCAACTCGGTGTCCCAAC (Arai et al, 2022); Stxbp3 forward = CTGCGACAAAATACGGGCAG; Stxbp3 reverse = GGGGAACAATGGGAACACCA; Rap1b

forward = GTGAATATAAGCTCGTCGTGC, Rap1b reverse = ACACTGCTGTGCATCTACTTC (Chen et al, 2020).

## RNA sequencing

Mice were exposed to Control diet or HFD for 14 days with or without the concurrent administration of LiPR (as described above). There were these mouse groups used for the RNAseq: Control (LFD) Diet, HFD, Control Diet + LiPR, and HFD + LiPR. The weight of each mouse was recorded every 2 days. At the end of the 14 days all the mice were culled. Mice were culled by cervical dislocation and brains collected in ice-cold HBSS with 10 mM HEPES (Thermo). Coronal, 500 μm sections were cut on a vibratome (Leica) and MBH tissue was sampled with Reusable rapid biopsy punch kit with 1.0 mm in diameter (WPI, Hitchin, UK). Sampled tissue punches were flash frozen on dry ice and store at −80 ºC until RNA isolation by RNeasy Plus mini kit (Qiagen). RNA quality checking was completed with the Qubit fluorometer system (Invitrogen) and assessed using RIN (RNA integrity number, maximum score is 10). All samples had RIN above 8.5, most above 9. RNA was re-transcribed to cDNA, tagmented and finally sequenced on the Illumina next-gen sequencer (75 bp cartridges) following the Illumina user guide at at the Genomic Hub of School of Biosciences, Cardiff University. The RNA input concentrations for cDNA library preparation were normalized to 380 ng. cDNA libraries were carried out with 13 rounds of amplification.

## RNAseq data analysis

Raw reads were quality checked using FastQC (Andrews, 2010) and quality filtered with fastp (Chen et al, 2018). Paired-end reads for each sample were mapped to the *Mus musculus* genome GRCm39 using STAR (Dobin et al, 2013). Duplicates were marked using Picard (Broad-Institute, 2019) and alignments were quanti-fied using subread (Liao et al, 2013). For each of the three pairwise comparisons (Control Diet vs HFD, Control Diet vs Control Diet + LiPR, HFD vs HFD + LiPR), differentially expressed genes were identified using the DESeq2 pipeline as implemented in SARTools (Love et al, 2014; Varet et al, 2016). Genes were excluded from the analysis unless they had <100 reads in at least one sample, as were all mitochondrial genes. Differentially expressed genes were defined as having an absolute $\log_2$ fold change >= 0.5 and an adjusted $P$ value of <0.05. $P$ values were adjusted for multiple testing using the Benjamini-Hochberg method (Benjamini and Hochberg, 1995). Heatmaps were created using ComplexHeatmap (Gu, 2022) and GO term enrichment analysis was run using clusterProfiler and GO.db (Carlson, 2022; Yu et al, 2012). Analysis was run using R v4.2.1 in RStudio (R Development Core Team, 2011; RStudio Team, 2016). The data RNAseq data were deposited to the European Nucleotide Archive under accession PRJEB63681.

## hESCs-derived hypothalamic neuron differentiation

The maintenance and directed differentiation of embryonic stem cells to hypothalamic neurons were performed as previously described with slight modifications (Kirwan et al, 2017; Merkle

et al, 2015). Specifically, differentiated HUES9 hESCs (Passage 12–18; RRID:CVCL_0057) were plated on 12-well iMatrix-511 (Cat.No. T303; Takara Bio Inc) coated plates at a density of $\sim3\times10^6$ cells/well in Synaptojuice medium (adapted from (Telezhkin et al, 2016). Hypothalamic cultures were maintained in Synaptojuice medium and replated onto iMatrix-511-coated on Ibidi $35\times12$ mm imaging dishes (Cat.No. 81156; Ibidi) at a density of $\sim2\times10^6$ cells/well on days 33–40 post-differentiation. Calcium imaging experiments were then carried out on replated hypothalamic cultures at days 40+ post-differentiation.

## Calcium imaging

Hanks' Balanced Salt Solution Phenol Red free (HBSS, Cat. No. 14025092; ThermoFisher scientific) with synaptic blockers was used as an extracellular bath solution. The final composition of the extracellular bath solution contained (in mM) NaCl 138, KCl 5.3, $CaCl_2$ 1.26, $MgCl_2$ 0.49, $MgSO_4$ 0.4, $KH_2PO_4$, $NaHCO_3$, $Na_2HPO_4$ 0.34, D-Glucose 5.5, DL-AP5 100 µM (Cat. No. ab120271; Abcam), Picrotoxin 50 µM (Cat. No. 1128; Tocris), CNQX 30 µM (Cat. No. ab120044; Abcam), and Strychnine 20 µM (Cat. No. ab120416; Abcam). Concentrated stock solutions of 1 mM hPrRP31 were prepared in $H_2O$ and stored at $-20\,°C$ until the day of each experiment. Test concentration of 1 µM hPrRP31 were prepared in HBSS plus synaptic blockers immediately before each experiment. Differentiated stem cell-derived hypothalamic neurons were replated on ibidi $35\times12$ mm imaging dishes (Cat.No. 81156; Ibidi) and loaded with the Fluorescent Dye-Based Rhod-3 AM following the manufactures' instructions (Rhod-3 Calcium Imaging Kit, Cat.No. R10145; ThermoFisher scientific). Loading, incubation, and wash buffers were prepared in HBSS solution in the presence of synaptic blockers. Imaging dishes were placed on an Olympus BX51WI Fixed Stage Upright Microscope (RRID:SCR_023069) and imaged using a 16-bit high-speed ORCA Flash4.0 LT plus digital sCMOS camera (RRID:SCR_021971). Neurons were identified by Rhod-3 AM fluorescence using an excitation wavelength of 560 nm and an emission wavelength of 600 nm. Images were taken using a 40× objective and acquired at a rate of one frame per second (100 ms exposure/frame) using a CoolLED pE-300^white (RRID:SCR_021073) illumination system and HCImage (RRID:SCR_015041) software for acquisition. Neurons were perfused continuously at room temperature with HBSS solution (3 mL/min) using a gravity-driven perfusion system for the entire length of the experiment and the fluorescence time course was recorded. The time course of each experiment was as follow: After a washout period of 10 min, cells were perfused with extracellular bath solution containing 1 µM hPrRP31 for 10 min. After second washout of a period of 10 min neuron were stimulated by 50 mM KCl for two minutes. A final washout of 10 minutes followed the application of KCl. Each fluorescence time course was exported to Microsoft Excel (RRID:SCR_016137), and the change in fluorescence intensity as a function of time was expressed as $(F-F_0)/F_0$ or $\Delta F/F_0$, where $F$ is the total fluorescence and $F_0$ is the baseline fluorescence. The extent of hPrRP31 response was evaluated by comparing the change in the area under the curve (AUC) of the fluorescence time-course before, during, and after the perfusion of hPrRP31.

## Statistical analysis

Numbers of biological and technical repeats are provided in the Figure Legends and in the Method Details. Datasets were analysed with Microsoft Excel and GraphPad Prism. The statistical analysis has adhered to the following procedural algorithm. First, data sets were analysed by the Grubb's test (ESD method) to identify statistically significant outliers (alpha = 0.05), which were removed from following statistical analyses. Second, it was determined if the data are normally distributed by D'Agostino & Pearson's omnibus normality test. If the data set is smaller than $N=5$, the Kolmogorov–Smirnov normality test was used. If data were not normally distributed or if this could not be reliably determined due to small number of replicates, non-parametric statistical tests were used such as Mann–Whitney for un-paired experiments, Wilcoxon matched-pairs signed rank test for paired experiments, or Kruskal–Wallis test with Dunn's post-hoc test (for group comparison). If data are normally distributed, parametric tests were used. For simple comparison of two data groups, the un-paired two-tailed $T$-Test was used. The $T$-Test was also used for selected data groups containing three data sets that were not interdependent. For multiple factor or group comparison, One-way or two-way analysis of variance (ANOVA) was used with the Bonferroni's or Tukey's post-hoc test for the cross-comparison of individual data sets. The contingency distribution of data is tested by either Chi-square or Fisher's exact tests. For non-parametric tests, the data were presented as median ± interquartile range (IQR). The IQR was calculated as $R = P * (n+1)/100$, where P is the desired percentile (25 or 75 for quartiles) and n is the number of values in the data set (Hyndman and Fan, 1996). For parametric tests, the data were presented as mean ± standard error of mean (SEM). Results were considered significant with $P < 0.05$ (one asterisk). In graphs, two asterisks represent values of $P < 0.01$, three asterisks for $P < 0.001$.

# Data availability

The data RNAseq data were deposited to the European Nucleotide Archive under accession PRJEB63681.

# Peer review information

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

## Acknowledgements

We thank all members of Petrik, Maletinska and Merkle labs for their collegiality and support. We thank the Genomic Research Hub of Cardiff School of Biosciences for assisting us with the preparation of libraries for RNA sequencing. This work was supported by following grants: Wellcome Trust (WT) - 211221/Z/18/Z and New York Stem Cell Foundation (NYSCF) - NYSCF-R-156 to FTM, Chan Zuckerberg Initiative (CZI) – 191942 to FTM and supporting H-JCC and SM; Czech Academy of Sciences - RVO:61388963 to LM; Wellcome Trust (WT) - ISSF3 518511 and The Academy of Medical Sciences - SBF007\100124 to DP.

## Author contributions

**Sara KM Jörgensen**: Investigation. **Alena Karnošová**: Investigation. **Simone Mazzaferro**: Investigation. **Oliver Rowley**: Investigation. **Hsiao-Jou Cortina Chen**: Investigation. **Sarah J Robbins**: Investigation. **Sarah Christofides**: Data curation. **Florian T Merkle**: Supervision; Funding acquisition; Methodology; Writing—review and editing. **Lenka Maletínská**: Resources; Supervision; Methodology; Writing—review and editing. **David Petrik**: Conceptualization; Resources; Supervision; Investigation; Methodology; Writing—original draft; Writing—review and editing.

## Disclosure and competing interest statement

The authors declare no competing interests.

# Expanded View Figures

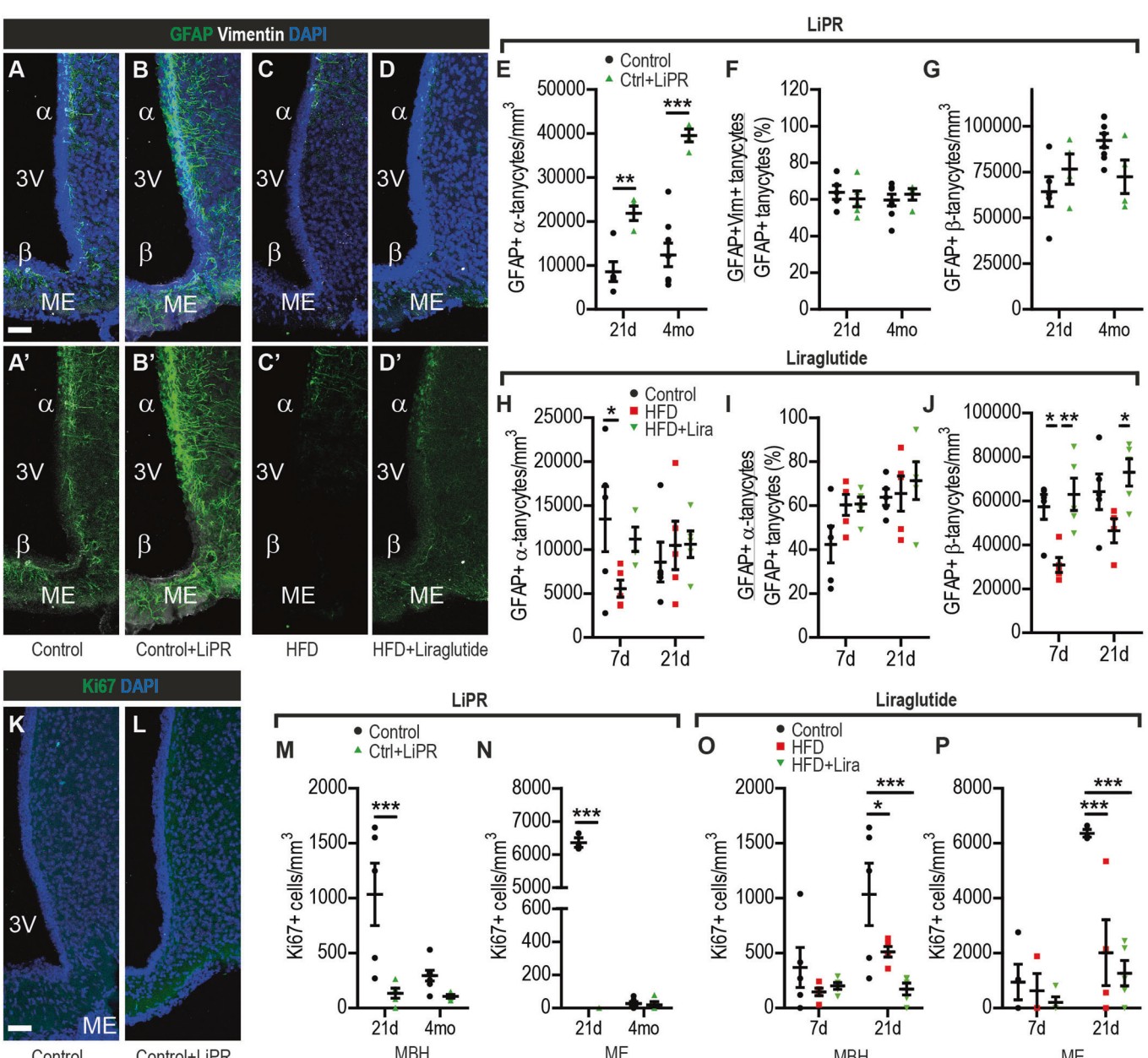

**Figure EV1. Effects of LiPR in Control Diet and Liraglutide on tanycytes and proliferating cells in the MBH.**

(A–D) Representative confocal images of HVZ stained as indicated in Control (**A**), Control+LiPR (**B**), HFD (**C**) and HFD + Liraglutide (Lira) (**D**) of 21d HFD group. Images in panels **A** and **C** are identical with representative confocal images in Fig. 2A and B and are shown for direct comparison with the control in both figures. (E–G) Effects of LiPR on number of GFAP+ α-tanycytes per volume of MBH (**E**), the proportion of GFAP+Vimentin+ tanycytes (**F**) and the number of GFAP+ β-tanycytes in MBH (**G**). (H–J) Effects of Liraglutide on GFAP+ α-tanycytes (**H**), the proportion of GFAP+Vimentin+ tanycytes (**I**) and the number of GFAP+ β-tanycytes in MBH (**J**). (K,L) Representative confocal images of MBH stained for Ki67 in Control (**K**) and Control+LiPR (**L**) of 4mo treatment group. (M,N) Quantification of Ki67+ cells in the MBH (**M**) and in the ME (**N**) from Control+LiPR mice. (O,P) Quantification of Ki67+ (**O**) and PCNA+ cells (**P**) in the MBH of Liraglutide-treated mice. Data information: Scale bars: 50 μm. $n = 5$ mice (7d and 21d), $n = 8$ mice (4mo Control), $n = 4$ mice (4mo Control + LiPR). In panel **P**, One-Way ANOVA. In all other panels, Two-Way ANOVA (**E**: treatment $F_{(1,17)} = 59.74$, $p < 0.0001$, duration $F_{(1,17)} = 16.87$, $p = 0.0007$, interaction $F_{(1,17)} = 7.04$, $p = 0.017$; **J**: treatment $F_{(2,23)} = 11.36$, $p = 0.0004$, duration $F_{(1,23)} = 4.42$, $p = 0.047$; **M**: treatment $F_{(1,17)} = 14.52$, $p = 0.0014$, duration $F_{(1,17)} = 7.24$, $p = 0.016$, interaction $F_{(1,17)} = 6.2$, $p = 0.023$; **N**: treatment $F_{(1,14)} = 4481$, $p < 0.0001$, duration $F_{(1,14)} = 4404$, $p < 0.0001$, interaction $F_{(1,14)} = 6.2$, $p < 0.0001$; **O**: treatment $F_{(2,24)} = 7.04$, $p < 0.0039$, duration $F_{(1,24)} = 8.35$, $p < 0.0081$, interaction $F_{(2,24)} = 3.01$, $p < 0.068$; **P**: treatment $F_{(2,17)} = 10.11$, $p < 0.0013$, duration $F_{(1,17)} = 21.99$, $p < 0.0002$, interaction $F_{(2,17)} = 6.1$, $p < 0.01$). *$p < 0.05$, **$p < 0.01$, ***$p < 0.001$ (Bonferroni's test). Data are presented as mean ± SEM.

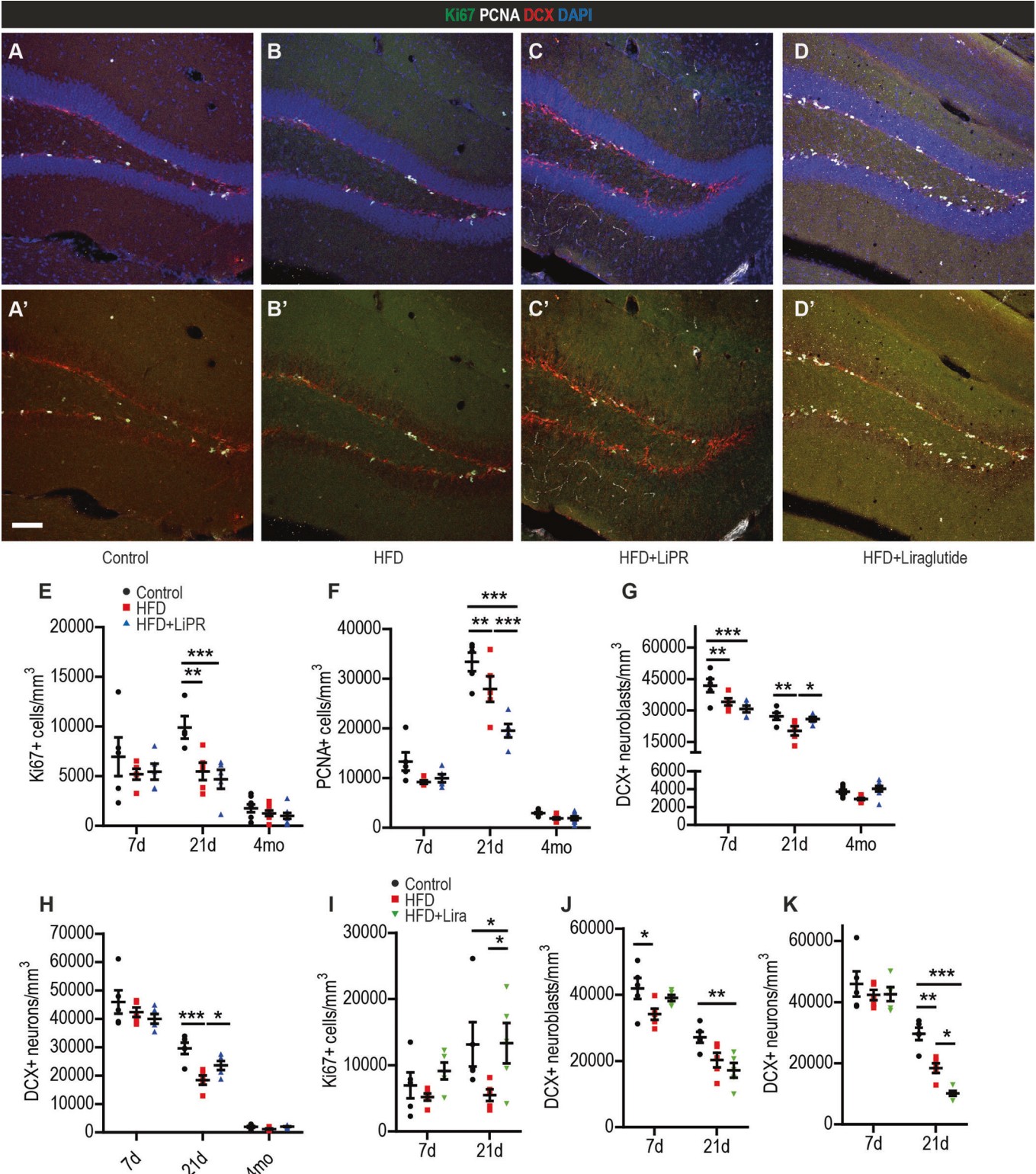

◀ **Figure EV2. LiPR promotes neurogenesis in the SGZ.**

(**A–D**) Representative confocal images of the Dentate Gyrus (DG) of the hippocampus showing the Subgranular Zone (SGZ) stained as indicated in Control (**A**), HFD (**B**), HFD + LiPR (**C**) and HFD + Lira of 21d HFD group. (**E–H**) Quantification of Ki67+ (**E**), PCNA+ cells (**F**), DCX+ neuroblasts (**G**) and DCX+ neurons (**H**) in SGZ for Control, HFD and HFD + LiPR. (**I–K**) Cell quantification in SGZ as described for Control, HFD and HFD + Liraglutide. Data information: Scale bars: 50 µm. $n = 5$ mice per data set for 7d and 21d groups, $n = 8$ mice per data set for 4mo group. All panels, Two-Way ANOVA (**E**: treatment $F_{(2,44)} = 9.88$, $p = 0.0003$, duration $F_{(2,44)} = 6.86$, $p = 0.0002$, interaction $F_{(4,44)} = 9.88$, $p = 0.0003$; **F**: treatment $F_{(2,45)} = 21.19$, $p < 0.0001$, duration $F_{(2,45)} = 374.3$, interaction $F_{(4,45)} = 9.88$, $p < 0.0001$; **G**: treatment $F_{(2,45)} = 11.38$, $p < 0.0001$, duration $F_{(2,45)} = 463.03$, $p < 0.0001$; interaction $F_{(4,45)} = 6.25$, $p = 0.0004$; **H**: treatment $F_{(2,45)} = 8.77$, $p = 0.0006$, duration $F_{(2,54)} = 552.73$, $p < 0.0001$, interaction $F_{(4,45)} = 3.78$, $p = 0.011$; **I**: treatment $F_{(2,24)} = 4.3$, $p = 0.025$, duration $F_{(1,24)} = 4.22$, $p = 0.05$; **J**: treatment $F_{(2,24)} = 7.21$, $p = 0.0035$, duration $F_{(1,24)} = 95.86$, $p < 0.0001$; **K** treatment $F_{(2,24)} = 12.29$, $p = 0.0002$, duration $F_{(1,24)} = 161.57$, $p < 0.0001$, interaction $F_{(2,24)} = 5.96$, $p = 0.0079$). *$p < 0.05$, **$p < 0.01$, ***$p < 0.001$ (Bonferroni's test). Data are presented as mean ± SEM.

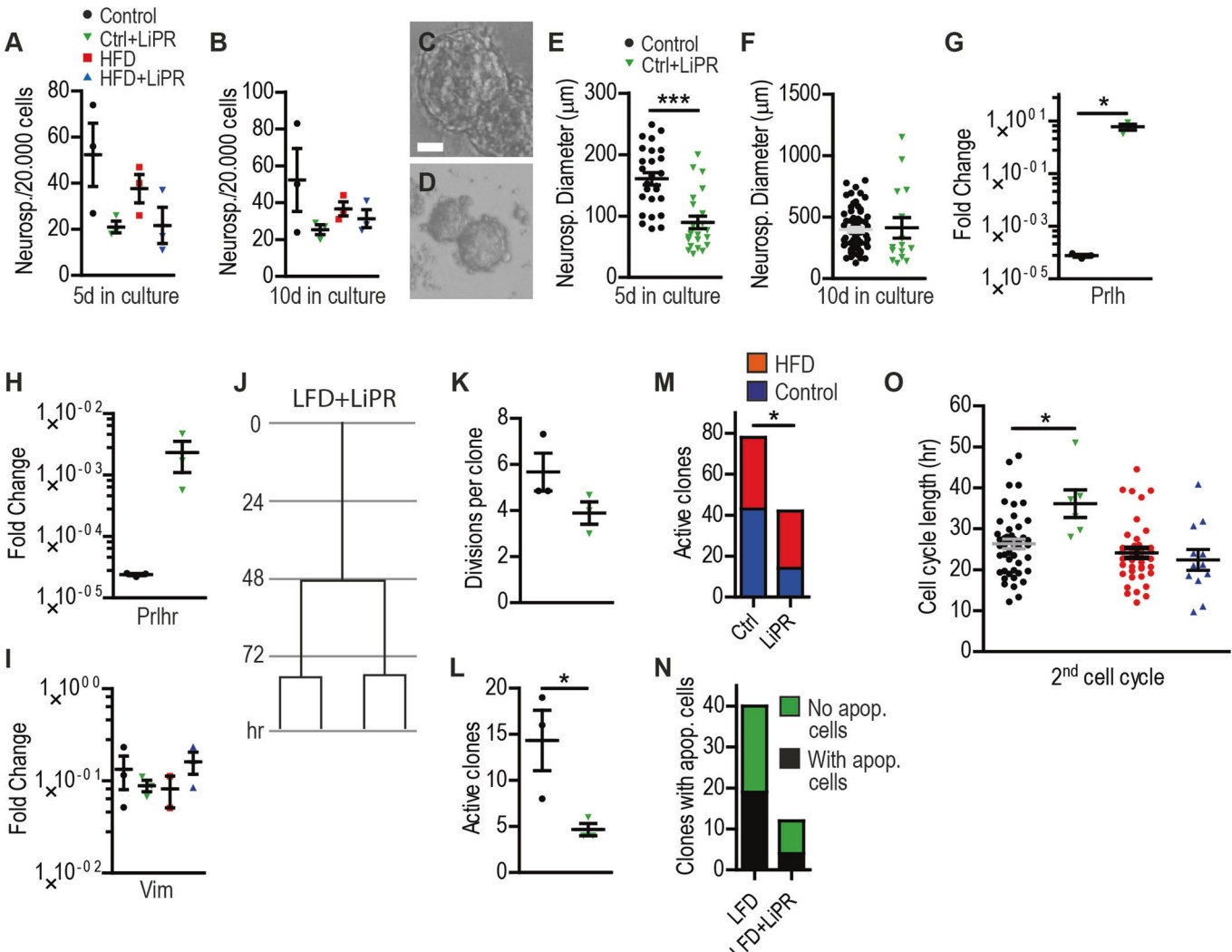

**Figure EV3. LiPR reduces proliferation of naïve neurospheres.**

(A,B) Quantification of number of HVZ-derived neurospheres per 20.000 plated cells after 5d (A) and 10d (B) in culture. (C,D) Representative images of HVZ-derived neurospheres 5d in culture from Control (C) and Control + LiPR (D) treated mice. The Image in panel C is identical with the image in Fig. 3A and is shown for direct comparison with the control in both figures. (E,F) Quantification of diameter of neurospheres 5d (E) and 10d (F) in culture. (G-I) Relative mRNA fold change of Prlh (G), Prlhr (H) and Vimentin (I) compared to Gapdh in cDNA from MBH-derived neurospheres (10d in culture). (J) An example cell division tree from 4-day time-lapse imaging of aNSCs from HVZ of Control + LiPR treated mouse. (K) Quantification of the number of cell divisions per division tree. (L) Time-lapse quantification of active (dividing) clones per 20.000 plated cells. (M) Number of all active clones of HVZ aNSCs from Control or LiPR-treated mice exposed to Control or HFD. Active clones pooled from all observed time-lapse imaging regions of interests per given treatment group. (N) Proportion of active clones containing at least one apoptotic cell. (O) Cell cycle length from the time-lapse imaging in the 2nd observed cell division. Data information: Scale bar: 20 μm. In all panels, $n = 3$ mice per data set. In panel E, number of neurospheres: $n = 26$ (Control), 22 (Control+LiPR). In panel F, number of neurospheres: $n = 66$ (Control), 15 (Control+LiPR). For panel K,L, number of traced cells: $n = 25$ per data set. In panel O, number of traced cells: $n = 47$ (Control), 6 (Control+LiPR), 35 (HFD), 12 (HFD+LiPR). Fisher's exact test (M,N), Un-paired two-tailed T-Test (E-H,K,L,O) and Two-Way ANOVA with Bonferroni's test (A,B,I). $*p < 0.05$, $***p < 0.001$. Data are presented as mean ± SEM.

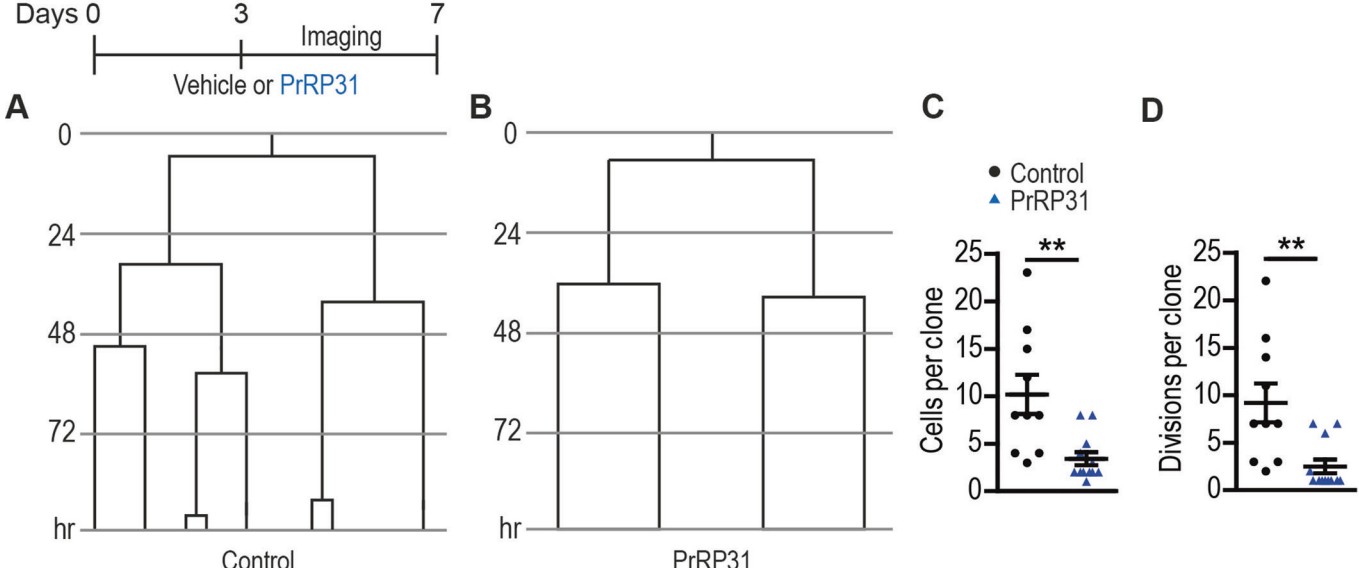

**Figure EV4. LiPR reduces proliferation of naïve htNSCs.**

(A,B) Example cell division trees from 4-day time-lapse imaging of aNSCs from HVZ of Control (A) and hPrRP31-treated (B) cells. A schematic of the experimental protocol shown above. (C,D) Quantification of cells per clone (C) and divisions per clone (D). Data information: In all panels, $n = 3$ mice per data set. In panels C,D, number of traced clones: $n = 10$ (Control), 12 (Control+LiPR). Un-paired two-tailed T-Test. **$p < 0.01$. Data are presented as mean ± SEM.

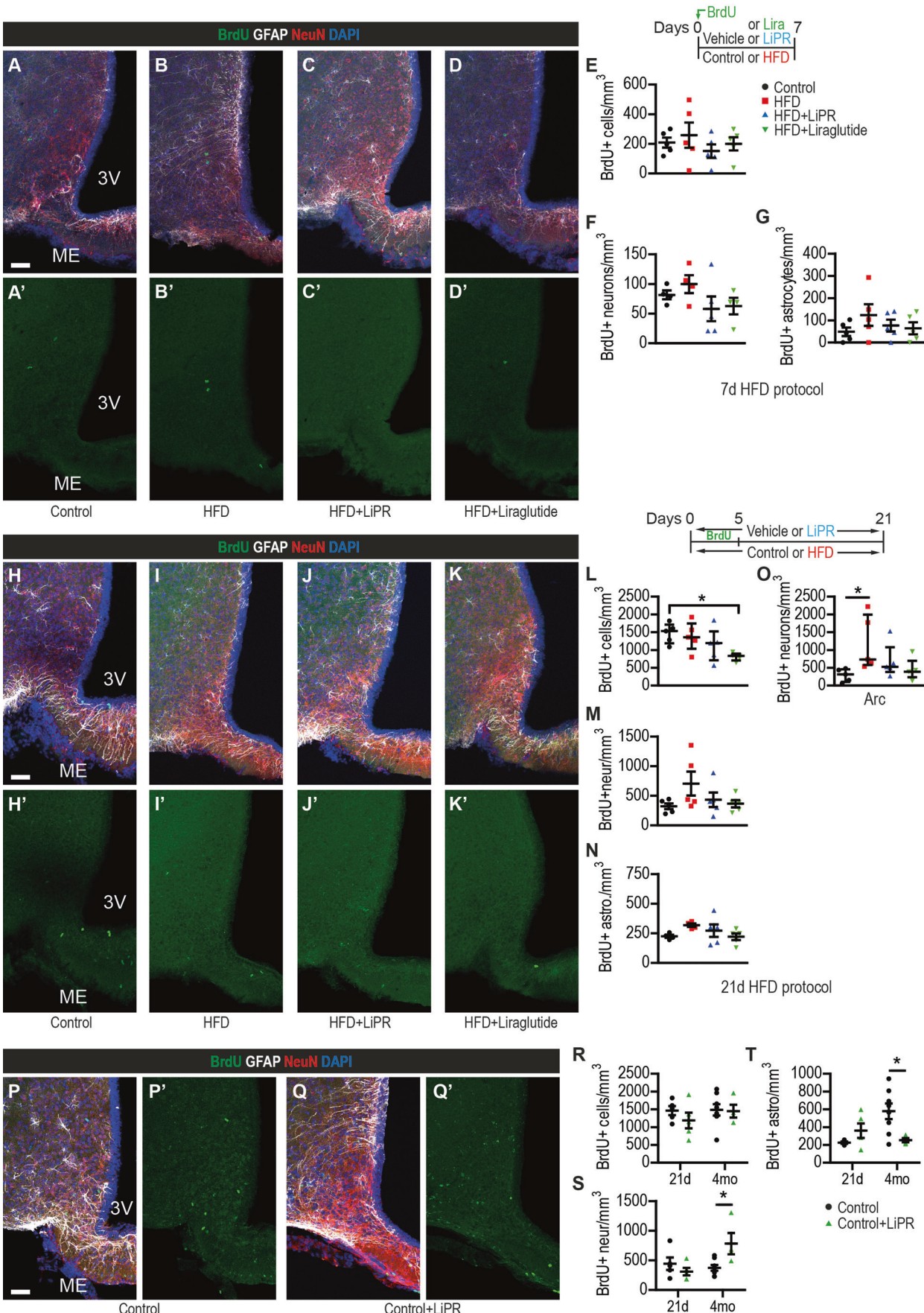

◀ **Figure EV5.  Effects of LiPR on new adult-generated cells in shorter HFD protocols and in the context of Control Diet.**

(A–D, H–K) Representative confocal images of HVZ and MBH stained as indicated in Control (**A,H,P**), HFD (**B,I**), HFD + LiPR (**C,J**), HFD + Liraglutide (Lira) (**D,K**) and Control Diet + LiPR (**Q**) of 7d (**A–D**), 21d (**H–K**) and 4mo (**P,Q**) groups. (**E–G**) Quantification of BrdU+ cells (**E**), BrdU+ neurons (**F**) and BrdU+ astrocytes (**G**) in the MBH of 7d HFD group. (**L–N**) Quantification of BrdU+ cells (**L**), BrdU+ neurons (**M**) and BrdU+ astrocytes (**N**) in the MBH parenchyma and BrdU+ neurons in the Arc (**O**) of 21d HFD group. (**R–T**) Quantification of BrdU+ cells (**R**), BrdU+ neurons (**S**) and BrdU+ astrocytes (**T**) in MBH parenchyma of Control and Control Diet + LiPR treated mice of 21d and 4mo groups. Data information: Scale bars: 50 μm. $n = 5$ mice per data set (7d, 21d), $n = 8$ mice (Control 4mo), $n = 4$ mice (Control + LiPR 4mo) per data set. In panel **O**, Kruskal–Wallis test with Dunn's test ($H = 9.88$, $p = 0.02$). In panels **E**–G and **L**–N, One-Way ANOVA with Tukey's test (**L**: $F_{(3,19)} = 3.58$, $p = 0.038$) or Bonferroni's test (**N**: $F_{(3,19)} = 4.3$, $p = 0.021$). In panels **R–T**, Two-Way ANOVA with Bonferroni's test (**S**: treatment $F_{(1,18)} = 5.71$, $p = 0.021$; duration $F_{(1,18)} = 9.9$, $p = 0.0056$; interaction $F_{(1,18)} = 6.43$, $p = 0.02$; **T**: treatment $F_{(1,17)} = 1.32$, $p = 0.27$; duration $F_{(1,17)} = 2.2$, $p = 0.15$; interaction $F_{(1,17)} = 7.75$, $p = 0.013$). *$p < 0.05$, ***$p < 0.001$. Data are presented as median ± SEM (**O**) or mean ± SEM (all other).

