## [Peer Review File · EMBO Reports]

An analogue of the Prolactin Releasing Peptide reduces obesity and promotes adult neurogenesis

Sara Jørgensen, Alena Karnošová, Simone Mazzaferro, Oliver Rowley, Hsiao-Jou Cortina Chen, Sarah Robbins, Sarah Christofides, Florian Merkle, Lenka Maletínská, and David Petrik

DOI: [10.15252/embr.202357414](https://doi.org/10.15252/embr.202357414)

Corresponding author(s): David Petrik (PetrikD@cardiff.ac.uk)

Review Timeline:

Transfer from Review Commons:	28th Apr 23
Editorial Decision:	5th May 23
Revision Received:	13th Jul 23
Editorial Decision:	18th Aug 23
Appeal Received:	14th Sep 23
Editorial Decision:	20th Sep 23
Revision Received:	21st Sep 23
Editorial Decision:	22nd Oct 23
Revision Received:	2nd Nov 23
Accepted:	17th Nov 23

Editor: Ioannis Papaioannou / Bernd Pulverer

Transaction Report: This manuscript was transferred to EMBO reports following peer review at Review Commons.

Review
COMMONS

Review #1

1. Evidence, reproducibility and clarity:

Evidence, reproducibility and clarity (Required)

In this manuscript, Jorgensen and colleagues describe their findings on the action of a palmitoylated form of prolactin-release peptide (LiPR) on neural stem cells (NSC) in the adult mouse hypothalamus and adult mouse hippocampus. Their main conclusion is that LiPR can counteract the effects of high-fat diet (HFD) and rescue some of the adverse effects of HFD. Specifically, the authors provide evidence that:

- Exposure to HFD reduces the number of presumptive adult neural stem cells (NSCs) in the adult hypothalamus, whereas exposure to LiPR reverses this trend.
- The results suggest that LiPR reduces the proliferation of alpha-tanycytes and/or their progeny in the hypothalamus in the context of HFD, with Liraglutide acting similarly. In contrast, while LiPR also suppresses proliferation in the SGZ, Liraglutide works there in the opposite direction.
- LiPR also helps the survival of adult-born hypothalamic neurons.
- Reduction of proliferation by LiPR suggests a model where LiPR increases the number of NSCs presumably by reducing their rate of activation.
- The results suggest that LiPR promotes expression of PrRP receptors in the hypothalamic neurons, suggesting that PrRP may act directly on such neurons (and tanycytes?) in vivo.
- The authors also show that HFD and LiPR alter gene expression profiles of the MBH cells, with HFD, but not LiPR, inducing myelination-related genes.
- Finally, they show that PrRP stimulates an increase in Ca²⁺ in in vitro-derived human hypothalamic neurons.
- The authors conclude that LiPR may be reducing activation and proliferation of the hypothalamic stem cells and thereby preserve their pool from exhaustion, which was stimulated by HFD.

The manuscript presents interesting data and is clearly written. There are several comments, mainly editorial.

1. It is unclear why most of the experiments do not include the control+LiPR group. Even though the focus of the study was the action of LiPR in the context of HFD, questions remain regarding the action of LiPR per se. Is LiPR (or Liraglutide, for that matter) completely inactive on the normal diet background, with respect to neurogenesis in the hypothalamus and the hippocampus? Whether the answer is positive or negative, it would give a much better understanding of the action of LiPR -

does it regulate neurogenesis in various physiological contexts, or does it only kick in with a particular type of diet? In fact, this was examined (see Supplementary figures), but only for the cells in culture and, when performed with animals, was limited to 7 and 21 days, rather than 4 months, which would have been much more informative.

2. The question above is also relevant when considering the conclusions on the potential depletion of the stem cell pool (again, whether in the hypothalamus or the hippocampus), particularly at the 4-months time point. The mice are ~6 months old by that time, and neurogenesis in both regions is expected to decrease by that time. Are LiPR or Liraglutide able to suppress or exacerbate this decrease? Can they be used to mitigate this decrease when mice are on a regular diet?
3. A somewhat related issue is that, in most cases, only the percentage or the density of cells are shown on the graphs, rather than the absolute numbers (at least for some cases). This sometimes complicates the comparisons; for instance, does the surface of the hypothalamus change between 2 and 6 months of age? The tanycytes' number stays, apparently, the same (e.g., Fig. 2) but the production of new neurons is supposed to fall dramatically.
4. The authors write "LiPR may prevent stem cells from exhaustion, induced by HFD" - but it is not clear that HFD indeed leads to exhaustion - there is no statistically significant difference in the number of the stem cells (alpha-tanycytes) between the control and HFD or between HFD at 1, 3, or 12 weeks.
5. Numerous papers show that the rate of production of new adult hypothalamic neurons (mainly those derived from beta-tanycytes) drops drastically within the first several weeks of mouse life. Does HFD accelerate, and LiPR mitigate, this decrease? Perhaps one can calculate the numbers from the graphs, but it would help if this is explained in the text of the manuscript. Also, it is not always clear whether specific experiments were performed with the zones of the hypothalamic wall that only contain alpha-tanycytes.
6. A sharp increase in PCNA+ cells in the hippocampus at the 21-day time point, both in the control and in the HFD and HFD/LiPR groups (Fig. S2f) is a little puzzling because neither the Dcx+ nor the Ki67+ cells show this increase.
7. The study deals with several agents and several processes; a simple scheme that summarizes authors' conclusions might help to better understand the relationships between those agents and processes.

****Referee cross-commenting****

I agree, the lack of the LiPR group complicates the interpretation of the results. I also agree that the experiments with vimentin staining, calcium increase, and even with neurospheres do not add much to the main questions that this study attempts to answer, and I'd rather see a more thorough analysis of the activation and

differentiation data. I also want to reiterate that the concept of LiPR/PrRP preventing the exhaustion of the hypothalamic stem cell pool is not clear, because it is not shown that this pool does actually get exhausted under normal or HFD conditions. This latter issue again requires the LiPR-alone group.

Also, as a clarification - I wrote about 1 month required to complete the revision assuming that the authors actually have the data on the Control+LiPR group or at least the specimens available, mainly because the supplementary material shows results with this group, at least with the neurospheres. If this group is fully missing, then the effort will obviously take a longer time.

2. Significance:

Significance (Required)

The provided evidence suggests, for the first time, that PrRP prevents the loss of the neural stem cells population in the adult hypothalamus that was diminished by obesity and HFD. This finding might be interesting to a broad audience.

3. How much time do you estimate the authors will need to complete the suggested revisions:

Estimated time to Complete Revisions (Required)

(Decision Recommendation)

Less than 1 month

Yes

Review #2

1. Evidence, reproducibility and clarity:

Evidence, reproducibility and clarity (Required)

The authors examine the effect of an anorexigenic drug, LiPR in the context of treatment with high fat diet (HFD) and with a special focus on hypothalamic neural stem/progenitor cells and neurogenesis. The work is mostly based on mice and a barrage of different techniques (confocal imaging, cell cultures with time lapse, gene expression...) are used.

The results are interesting because they address the yet-poorly understood implication of hypothalamic neurogenesis in food intake and energy balance.

The results point at complex effects at different levels (neural stem cells, neurons, division, survival...). The experimental approach is sometimes thorough in the treatment of details on the one hand, it also lacks of consistency on the other, and as a result the conclusions lack strength.

There is a number of experiments that sometimes seem unrelated and this hurts the comprehension of the manuscript, specially in lieu of the complexity of the results obtained.

****These are the detailed comments:****

A major issue is the lack of a LiPR-only group, which would much facilitate the interpretation of the results. The effect of LiPR alone is however tested, but only in comparison with the Control in one of the in vitro experiments (S.Fig. 3)

As plotted, in Fig 1B is difficult to interpret the effect of HFD and LiPR, might be using percentage and noting the statistical differences as in the other would help. It looks like HFD has no effect compared to control on weight and only at the end LiPR could have an effect. On the other hand, after 4 months, HFD mice are clearly above the controls and it is then, albeit when weight gain has reached a plateau, that LiPR has an effect. The election of these arbitrary paradigms and their drawbacks has to be better explained.

Why was the proportion of GPR10+BrdU+MAP2+ cells only assessed in control mice and no in the experimental groups if its expression in overall neurons changes?

This suggests that the receptor is expressed in neurons. Interestingly, exposure to 21d

HFD reduced density of GPR10, which was rescued by LiPR administration (Fig.1L). Why was this time point chosen and not the longer-term one? What is the consequence of the alterations in the potential number of GPR10, specially in relation to the administration of LiPR?

This clarification is important because a 14-day treatment was chosen for the in vitro experiments in which LiPR, but not HFD, seems to have an effect on cell proliferation. Might be it would have been more useful to use a paradigm in which HFD has an effect to better compare with in vivo work and for the rationale of the work.

"Besides GPR10, we co-localized neuronal cytoskeleton structures with NPFFR2 in the MBH (Fig.1O-P)..." Why were not GPR10 and NPFF2 analyzed in a similar and consistent manner ? It is confusing.

The number of GFAP+ α -tancytes is not significantly changed by HFD therefore LiPR does not rescue, but rather increases the number of GFAP+ α -tancytes in the 7-day setting. There are no differences among groups later, the effect is lost by 21 days, therefore there is a transient excess of GFAP+ α -tancytes which later "disappear" in the LiPR group. The authors state that LiPR rescues the decrease in "htNSCs", but after 21 days the number of the GFAP+ α -tancytes is the same in all groups without the need of LiPR. There is no experimental follow up (addressing proliferation and survival of these cells) and the conclusions stated in the text (results and discussion) are not really supported by the data. The in vitro experiments could be a complement, but are no substitute for the missing in vivo exploration.

The fact that cell division is "rarely found" (Rax GFAP) experiments also push for further investigation.

It is difficult to see that relevance of the inclusion of the vimentin staining experiment if there is no further exploration. The effect of LiPR is only transient, in the 7-day paradigm and as the parameter evaluated is the proportion of vimentin+ tancytes among GFAP+ tancytes it could only be reflecting increased expression of the filament.

"Nevertheless, we did not observe a statistically different change in the area occupied by Rax+ tancytes (Fig.2H)."

Why did the authors use Rax only for this experiment if "GFAP+ α -tancytes which are considered the putative htNSCs?" What is the justification for not seeing changes in relation to the results reported in Fig 2D-F?

"Because Vimentin is associated with nutrient transport in cells and with metabolic response to HFD 52-54, we quantified the proportion of GFAP+ tanycytes expressing Vimentin (Fig.2F)."

It is difficult to see that relevance of the inclusion of the vimentin staining experiment if there is no further exploration. The effect of LiPR is only transient, in the 7-day paradigm and as the parameter evaluated is the proportion of vimentin+ tanycytes among GFAP+ tanycytes it could only be reflecting increased expression of the filament.

Why there is no Ki67 experiment in the 7-day paradigm if that is the timepoint in which changes in the number or proportion of GFAP+ tanycytes are observed? PCNA was then used but only in the 21-day paradigm. What is the interpretation and relevance of these data? What are the non-htNSCs proliferating cells, whose dynamics are different from the changes in the number or proportion of htNSCs that could be potentially related to changes in mitosis? Again, I think it would be much useful for the work to explore in detail the changes in the putative htNSCs than investing in experiments that only add confusion.

The inclusion of Liraglutide + HFD, (not Liraglutide alone) only in some of the experiments is pointless if there is no direct comparison with LiPR and a timepoint is missing.

In S.Fig 3, Fig. 5 and S.Fig 7 LFD (low fat diet?) is used in several occasions as in: "on reducing number of PCNA+ cells in 21d protocol (one-way ANOVA (OWA), $F(2,12) = 16.66$, $p = 0.0003$) when compared to both LFD and HFD groups". Is this the control diet?

The final experiment shows that application of hPrRP31, a variation of LiPR, causes an immediate calcium increase in human induced pluripotent stem cell-derived hypothalamic nucleus. This finding is interesting in itself because it brings light about the function of the receptor/s. It would have been very useful to test what other receptors mentioned to bind LiPR is mediating the effect. In any case, the focus of the work are the neural stem/progenitor cells responsible for neurogenesis and the changes in their properties because of HFD and LiPR, therefore I would trade these experiments for a more thorough and detailed dissection of these effects.

****Minor points:****

Introduce "GLP-1RA"

"HFD-induced inflammation and astrogliosis in the hypothalamus 45,46, whereas the

long (4mo) protocol leads to DIO" Are these notions exclusive?

"LiPR displays no effects on astrocytes" "Displays" is not the correct term.

****Referee cross-commenting****

I think we all referees agree for the most part. The main concern stated by all of us is the lack of a LiPR-alone group. The rest of the concerns are also related or complementary. In my opinion the mostly common view by the referees is reassuring.

2. Significance:

Significance (Required)

The strengths of the work are its novelty in the field and the variety of techniques employed. The work has the potential of unveiling mechanistic insight into the regulation of neural stem/progenitor cells and neurogenesis. The main audience of this work would be the community working on this field. The lack of experiments testing that the changes observed actually participate in food intake prevent the work from being of relevance for a broader audience (food intake, energy balance, obesity...).

The limitations are the descriptive nature of the work and the lack of a consistent and systematic experimental design that would allow to extract solid conclusions upon to which build upon future research.

3. How much time do you estimate the authors will need to complete the suggested revisions:

Estimated time to Complete Revisions (Required)

(Decision Recommendation)

More than 6 months

4. *Review Commons* values the work of reviewers and encourages them to get credit for their work. Select 'Yes' below to register your reviewing activity at Web of Science Reviewer Recognition Service (formerly Publons); note that

the content of your review will not be visible on Web of Science.

No

Review #3

1. Evidence, reproducibility and clarity:

Evidence, reproducibility and clarity (Required)

The work of Jørgensen et al describes the effect of a lipidized analogue of the prolactin releasing peptide (LiPR) on the mouse metabolism in response to high fat diet (HFD) and on hypothalamic and subgranular zone (SGZ) neurogenesis. They conclude that LiPR reduces body weight and improves metabolic parameters affected by HFD as well as it concomitantly stimulates neurogenesis in both niches the SGZ and the hypothalamus. The link between both effects is not demonstrated. The work is well conducted, the hypothesis is interesting and the experimental approach is adequate. The scope is wide and results are interesting, however a few aspects need to be further clarified. The manuscript is well written although the modification of some aspects would facilitate the reading such as the use of non described abbreviations for example.

****Major comments:****

1. One concern in this study is the experimental groups. Authors analyze three groups control, HFD and HFD treated with LiPR. Authors conclude that the effects of LiPR are diet independent. However, given the results obtained by the authors on the effect of LiPR, the main question that arises in here is whether LiPR would have an effect on control mice. It seems that a group is missing in the experimental design in which control mice are treated with LiPR during 7, 21 and the last two weeks of the 4 months. Author must include this information or at least argue the election of the experimental design.
2. Body weight is found reduced by LiPR as well as other metabolic parameters in mice treated with LiPR during the last two weeks of the 4 Mo HFD. However, no

effects on hypothalamic or SGZ neurogenesis are not observed in this experimental group. How do authors explain this results?

3. In figure 1 I-K images are not clear and better resolution images would help.

4. Authors conclude that LiPR is increasing the number of NSC by reducing their activation. However, authors show an induced increase in htNSC only in mice fed HFD for 7 days and not in the 21 day fed mice or the 4 mo fed mice (fig 2 d-f). In addition, authors test for the number of cells expressing Ki67 (fig 2 L), however, the number of Ki67+ alpha tanicytes is not shown.

5. On figure 2B it seems that is alpha 2 tanicytes that are missing in response to HFD.

6. Are Fig 2 A-C images representative of mice fed HFD for 7 days?

7. By looking at figure 2B it seems like the proportion of alpha tanicytes is higher in HFD since no or very few tanicytes are observed and almost all of them are alpha tanicytes.

8. In fig 2 d-f, an increase in the number of GFAP+ alpha tanicytes and its proportion as well as labelled with vimentin is observed in control mice fed with normal diet for 7 days compared with mice fed normal diet for 21 days. How do authors explain this difference?

9. In fig 2 Why are the differences in RAX, KI67 and PCNA only present in mice fed HFD for 21 days?

10. Authors test for adult hippocampal neurogenesis in the three groups. DO images in fig S2 correspond to the 21 day treatment group?

11. On fig S2 C, it seems that in HFD fed mice treated with LiPR newly generated neuroblasts are more differentiated have authors looked at DCX+ cell morphology?

12. In this same figure, it seems like the number of DCX+ neuroblasts and the number of newly generated neurons is reduced in mice of the 21 d group compared to the 7 day group. Is this statistically significant?

13. There is a large reduction in the number of DCX+ cells from control 21 d treated mice to control 4 month treated mice. Is this statistically significant? How do authors explain this dramatic reduction?

14. Authors do not show the effect of HFD on BrdU+ neurons in the Arcuate. However, all data need to be shown.

2. Significance:

Significance (Required)

In general the manuscript includes a great amount of work to demonstrate the effect of LiPR on neurogenesis (hippocampal and hypothalamic). The scope is wide, and the hypothesis is really interesting. Authors may need to solve some issues in order to completely demonstrate their claims and conclusions, but once the work is done, it will be very valuable to understand the effect of pharmacological agents used in the

field of endocrinology to treat metabolic disorders such as type 2 diabetes di type 2 diabetes.

So far, no studies have been done in which the effect of this molecules have been described on SGZ and hypothalamic neurogenesis. Both the field of endocrinology and metabolism as well as the field of adult neurogenesis may benefit of a study of this type.

3. How much time do you estimate the authors will need to complete the suggested revisions:

Estimated time to Complete Revisions (Required)

(Decision Recommendation)

Between 1 and 3 months

Yes

Revision Plan

Manuscript number: RC-2023-01900

Corresponding author(s): David Petrik

[The “revision plan” should delineate the revisions that authors intend to carry out in response to the points raised by the referees. It also provides the authors with the opportunity to explain their view of the paper and of the referee reports.]

The document is important for the editors of affiliate journals when they make a first decision on the transferred manuscript. It will also be useful to readers of the reprint and help them to obtain a balanced view of the paper.

*If you wish to submit a full revision, please use our "Full Revision" template. **It is important to use the appropriate template to clearly inform the editors of your intentions.**]*

1. General Statements [optional]

We thank the reviewers for a thorough review that will help us to improve the manuscript in the revision process. In our opinion, all three reviewers found the manuscript interesting, novel, and relevant for a broader readership. The reviewers suggested performing additional analyses of cell quantification from existing brain tissue or from newly generated tissue. All reviewers identified several shared concerns that we are happy to address by additional experiments and analyses to improve our manuscript. The reviewers suggested including the Control Diet + LiPR treatment group to further characterize the effects of LiPR on adult neurogenesis outside the context of the High Fat Diet. Also, the reviewers suggested including built upon the analysis of tanycytes and their proliferation. Some of these analyses will require generating new experimental animals, however, most analyses can be performed from already available brain tissue or previously collected confocal microscope images. Because we had anticipated some of the possible concerns, we have placed mice in the experiment already in February 2023. These mice are in the 4-month treatment group of Control Diet + LiPR. We will collect the brain tissue at the end of May 2023 and will analyze it in June and July 2023. In April and May 2023, we will work on analyses from existing tissue or images as described in detail below. We estimate that the suggested analyses are all feasible and should be manageable in 3 months. In fact, we are pleasantly surprised by the favorable nature of the reviews, especially from the reviewer 1 and 3, which allowed us to address around 50% of comments already as demonstrated in this revision plan (see section 3). Therefore, we are confident that we will be able to address the remaining concerns to full satisfaction of all relevant reviewers' comments.

2. Description of the planned revisions

Insert here a point-by-point reply that explains what revisions, additional experimentations and analyses are planned to address the points raised by the referees.

Reviewer 1

In this manuscript, Jorgensen and colleagues describe their findings on the action of a palmitoylated form of prolactin-release peptide (LiPR) on neural stem cells (NSC) in the adult mouse hypothalamus and adult mouse hippocampus. Their main conclusion is that LiPR can counteract the effects of high-fat diet (HFD) and rescue some of the adverse effects of HFD. Specifically, the authors provide evidence that:

- Exposure to HFD reduces the number of presumptive adult neural stem cells (NSCs) in the adult hypothalamus, whereas exposure to LiPR reverses this trend.
- The results suggest that LiPR reduces the proliferation of alpha-tanycytes and/or their progeny in the hypothalamus in the context of HFD, with Liraglutide acting similarly. In contrast, while LiPR also suppresses proliferation in the SGZ, Liraglutide works there in the opposite direction.
- LiPR also helps the survival of adult-born hypothalamic neurons.
- Reduction of proliferation by LiPR suggests a model where LiPR increases the number of NSCs presumably by reducing their rate of activation.
- The results suggest that LiPR promotes expression of PrRP receptors in the hypothalamic neurons, suggesting that PrRP may act directly on such neurons (and tanycytes?) in vivo.
- The authors also show that HFD and LiPR alter gene expression profiles of the MBH cells, with HFD, but not LiPR, inducing myelination-related genes.
- Finally, they show that PrRP stimulates an increase in Ca²⁺ in in vitro-derived human hypothalamic neurons.
- The authors conclude that LiPR may be reducing activation and proliferation of the hypothalamic stem cells and thereby preserve their pool from exhaustion, which was stimulated by HFD.

The manuscript presents interesting data and is clearly written. There are several comments, mainly editorial.

ANSWER: We thank the reviewer for the favorable and positive assessment of our manuscript and for finding our study to be interesting to a broad audience and well written, with most comments described by the reviewer as “editorial”. Below, we address the reviewer’s concerns in a detailed revision plan.

1. It is unclear why most of the experiments do not include the control+LiPR group. Even though the focus of the study was the action of LiPR in the context of HFD, questions remain regarding the action of LiPR per se. Is LiPR (or Liraglutide, for that matter) completely inactive on the normal diet background, with respect to neurogenesis in the hypothalamus and the hippocampus? Whether the answer is positive or negative, it would give a much better understanding of the action of LiPR - does it regulate neurogenesis in various physiological contexts, or does it only kick in with a particular type of diet? In fact, this was examined (see Supplementary figures), but only for the cells in culture and, when performed with animals, was limited to 7 and 21 days, rather than 4 months, which would have been much more informative.

ANSWER: We thank the reviewer for this suggestion. We agree that including the Control Diet + LiPR group for the 4-month HFD group would complement the results from the 7 and 21 days. We will generate this treatment group for the 4-month HFD group and analyze the effect of LiPR on aNSC and adult-generated neurons. These mice in the 4-month treatment are in the experiment already from February 2023 and we plan to analyze their brain sections in June and July 2023.

Revision Plan

2. The question above is also relevant when considering the conclusions on the potential depletion of the stem cell pool (again, whether in the hypothalamus or the hippocampus), particularly at the 4-month time point. The mice are ~6 months old by that time, and neurogenesis in both regions is expected to decrease by that time. Are LiPR or Liraglutide able to suppress or exacerbate this decrease? Can they be used to mitigate this decrease when mice are on a regular diet?

ANSWER: This concern will be addressed by analyzing the Control + LiPR mice for the 4-month HFD group (see our response to the point 1 above). We will analyze neural stem cells in the Hypothalamic Ventricular Zone and neural progenitors in the Median Eminence of these mice to address whether LiPR treatment changes the time-dependent decrease in both cell populations.

Reviewer 2

The authors examine the effect of an anorexigenic drug, LiPR in the context of treatment with high fat diet (HFD) and with a special focus on hypothalamic neural stem/progenitor cells and neurogenesis. The work is mostly based on mice and a barrage of different techniques (confocal imaging, cell cultures with time lapse, gene expression...) are used.

The results are interesting because they address the yet-poorly understood implication of hypothalamic neurogenesis in food intake and energy balance.

The results point at complex effects at different levels (neural stem cells, neurons, division, survival...). The experimental approach is sometimes thorough in the treatment of details on the one hand, it also lacks of consistency on the other, and as a result the conclusions lack strength.

There is a number of experiments that sometimes seem unrelated and this hurts the comprehension of the manuscript, specially in lieu of the complexity of the results obtained.

ANSWER: We thank the reviewer for finding our results interesting and relevant. We will strive to improve the consistency of our results in the revised manuscript to satisfy the reviewer's concerns.

1. A major issue is the lack of a LiPR-only group, which would much facilitate the interpretation of the results. The effect of LiPR alone is however tested, but only in comparison with the Control in one of the in vitro experiments (S.Fig. 3)

ANSWER: We agree with the reviewer that expanding on the LiPR-only effect would facilitate the interpretation of the results (see concern 1 and 2 of reviewer 2). We want to emphasize, however, that we analyzed the HFD-independent LiPR effects not only *in vitro* but also *in vivo* by quantifying the number of BrdU+ cells and neurons in the MBH of mice exposed to 21-day HFD (S.Fig. 5 O-Q) and by including the Control Diet + LiPR in the RNAseq experiment (Fig.5C). Nevertheless, we will analyze the number of alpha tancytes and proliferating cells for the 21-day Control Diet + LiPR treatment group. And we will generate mice treated with Control Diet + LiPR to complement the 4-month group. In this Control Diet + LiPR group, we will quantify the number of tancytes and number of BrdU+ cells and neurons.

Revision Plan

5. The fact that cell division is "rarely found" (Rax GFAP) experiments also push for further investigation.

It is difficult to see that relevance of the inclusion of the vimentin staining experiment if there is no further exploration. The effect of LiPR is only transient, in the 7-day paradigm and as the parameter evaluated is the proportion of vimentin+ tanycytes among GFAP+ tanycytes it could only be reflecting increased expression of the filament. "Nevertheless, we did not observe a statistically different change in the area occupied by Rax+ tanycytes (Fig.2H)." Why did the authors use Rax only for this experiment if "GFAP+ α -tanycytes which are considered the putative htNSCs?" What is the justification for not seeing changes in relation to the results reported in Fig 2D-F?

"Because Vimentin is associated with nutrient transport in cells and with metabolic response to HFD 52-54, we quantified the proportion of GFAP+ tanycytes expressing Vimentin (Fig.2F)."

It is difficult to see that relevance of the inclusion of the vimentin staining experiment if there is no further exploration. The effect of LiPR is only transient, in the 7-day paradigm and as the parameter evaluated is the proportion of vimentin+ tanycytes among GFAP+ tanycytes it could only be reflecting increased expression of the filament.

ANSWER: Because Vimentin is a marker of neural stem cells and alpha tanycytes, we quantified the number of GFAP+Vimentin+ tanycytes to complement the quantification of GFAP+ alpha tanycytes. We are sorry that this was not clear, and we highlighted this connection in the revised manuscript (line 165). Because Rax is expressed in alpha tanycytes, we expected that LiPR will increase Rax in the Hypothalamic Ventricular Zone (HVZ). We agree with the reviewer that further investigation may be useful, and we will quantify the number of alpha tanycytes positive for Rax instead of determining only the volume of Rax+ tissue. We will quantify Rax+GFAP+ neural stem cells in the HVZ and Rax+GFAP+ neural progenitors (so-called beta tanycytes) in the Median Eminence to improve characterization of the cell dynamics *in vivo*.

6. Why there is no Ki67 experiment in the 7-day paradigm if that is the timepoint in which changes in the number or proportion of GFAP+ tanycytes are observed? PCNA was then used but only in the 21-day paradigm. What is the interpretation and relevance of these data? What are the non-htNSCs proliferating cells, whose dynamics are different from the changes in the number or proportion of htNSCs that could be potentially related to changes in mitosis? Again, I think it would be much useful for the work to explore in detail the changes in the putative htNSCs than investing in experiments that only add confusion.

ANSWER: We apologize if the data presentation is confusing. We will include the quantification of the Ki67+ cells for the 7-day time point. In the MBH, many cell types undergo mitosis, including the oligodendrocyte precursor cells, microglia, astrocytes, and infiltrating macrophages. However, characterizing the identify of all these different cell types in response to the HFD and/or LiPR is beyond the scope of this study. To resolve whether HFD and/or LiPR influence proliferating aNSCs, we will quantify the proliferating cells in the HVZ, which will allow us to separate the proliferating aNSCs from all other proliferating cell types in the MBH.

Reviewer 3

The work of Jörgensen et al describes the effect of a lipidized analogue of the prolactin releasing peptide (LiPR) on the mouse metabolism in response to high fat diet (HFD) and on

Revision Plan

hypothalamic and subgranular zone (SGZ) neurogenesis. They conclude that LiPR reduces body weight and improves metabolic parameters affected by HFD as well as it concomitantly stimulates neurogenesis in both niches the SGZ and the hypothalamus. The link between both effects is not demonstrated. The work is well conducted, the hypothesis is interesting and the experimental approach is adequate. The scope is wide and results are interesting, however a few aspects need to be further clarified. The manuscript is well written although the modification of some aspects would facilitate the reading such as the use of non described abbreviations for example.

ANSWER: We thank the reviewer for the positive assessment of our manuscript and for recognizing its novelty and importance for the research in neurogenesis, endocrinology, and metabolism. We will strive to clarify and facilitate our conclusions to improve the manuscript.

1. One concern in this study is the experimental groups. Authors analyze three groups control, HFD and HFD treated with LiPR. Authors conclude that the effects of LiPR are diet independent. However, given the results obtained by the authors on the effect of LiPR, the main question that arises in here is whether LiPR would have an effect on control mice. It seems that a group is missing in the experimental design in which control mice are treated with LiPR during 7, 21 and the last two weeks of the 4 months. Author must include this information or at least argue the election of the experimental design.

ANSWER: We thank the reviewer for this insight. We agree that including the Control Diet + LiPR in some of our analyses would improve the revised manuscript as also noted by Reviewer 2 (comment 1 and 2) and by Reviewer 2 (comment 1 and 2). In the original manuscript, we included the quantification of BrdU+ cells in the MBH for the Control Diet + LiPR in the 21-day group. To expand on these results, we will quantify the effects of LiPR on alpha tanyocytes in the 21-day group. In addition, we will generate Control Diet + LiPR mice for the 4-month group to complement the HFD and HFD + LiPR data.

4. Authors conclude that LiPR is increasing the number of NSC by reducing their activation. However, authors show an induced increase in htNSC only in mice fed HFD for 7 days and not in the 21 day fed mice or the 4 mo fed mice (fig 2 d-f). In addition, authors test for the number of cells expressing Ki67 (fig 2 L), however, the number of Ki67+ alpha tanyocytes is not shown.

ANSWER: We thank the reviewer for this insight. In the revised manuscript (line 158), we corrected the inaccurate statement that LiPR increased the number of aNSCs and did not rescue their number, which was also noted by Reviewer 1 (comment 5) and by Reviewer 2 (comment 4). In addition, we will quantify the number of Ki67+ cells in the Hypothalamic Ventricular Zone (HVZ), which will address whether LiPR affects proliferation of aNSCs. This concern parallels comment 6 of Reviewer 2.

9. In fig 2 Why are the differences in RAX, KI67 and PCNA only present in mice fed HFD for 21 days?

ANSWER: We thank the reviewer for this question, which reflects a similar comment 6 of Reviewer 2. To improve consistency of the presented data, we will quantify the proliferating cells also for the 7-day time point. In addition, we will quantify the number of proliferating cells in the HVZ, which will allow us to address whether HFD and/or LiPR alter proliferation of tanyocytes.

Revision Plan

3. Description of the revisions that have already been incorporated in the transferred manuscript

Please insert a point-by-point reply describing the revisions that were already carried out and included in the transferred manuscript. If no revisions have been carried out yet, please leave this section empty.

Reviewer 1

4. The authors write "LiPR may prevent stem cells from exhaustion, induced by HFD" - but it is not clear that HFD indeed leads to exhaustion - there is no statistically significant difference in the number of the stem cells (alpha-tanocytes) between the control and HFD or between HFD at 1, 3, or 12 weeks.

ANSWER: We thank the reviewers for their insights. We adjusted the interpretation to better reflect our results. On line 442, we replaced the original statement "The lower cell activation may protect the stem cell pool from exhaustion elicited by the HFD" with a new one, "The lower cell activation may protect the stem cell pool from exhaustion elicited by the HFD".

5. Numerous papers show that the rate of production of new adult hypothalamic neurons (mainly those derived from beta-tanocytes) drops drastically within the first several weeks of mouse life. Does HFD accelerate, and LiPR mitigate, this decrease? Perhaps one can calculate the numbers from the graphs, but it would help if this is explained in the text of the manuscript. Also, it is not always clear whether specific experiments were performed with the zones of the hypothalamic wall that only contain alpha-tanocytes.

ANSWER: Our results show that LiPR rescues the HFD-induced reduction in adult-generated hypothalamic neurons only in the context of 4-month HFD but not in the 7- and 21-day HFD. In the methods (line 877), we specify that "the Region of Interest (ROI) quantified included the MBH parenchyma with the Arcuate (Arc), DMN and Ventromedial (VMN) Nuclei and the Medial Eminence (ME)". In the results of the revised manuscript (lines 301-303), we highlighted the areas of the ROI. Upon the request of Reviewer 3 (comment 14), we included new data on quantification of BrdU+ neurons in the Arcuate Nucleus (S.Fig.5O). This data show that 21d HFD increases the number of new neurons in ArcN, which is reversed by LiPR or Liraglutide (text added to results and discussion on lines 309-313 and 468-474, respectively). Finally, in the discussion (lines 464-488), it is stated that HFD and/or LiPR had no effect on number of new hypothalamic neurons or cells in the MBH parenchyma in the 7- and 21-day groups and this is discussed in the context of relevant literature.

7. The study deals with several agents and several processes; a simple scheme that summarizes authors' conclusions might help to better understand the relationships between those agents and processes.

ANSWER: We thank the reviewer for this useful suggestion. We included a summarizing schematic in the revised manuscript as the new Figure 6. We will update the schematic for the final revised manuscript, when we will incorporate the new analyses.

Reviewer 2

2. As plotted, in Fig 1B is difficult to interpret the effect of HFD and LiPR, might be using percentage and noting the statistical differences as in the other would help. It looks like HFD has no effect compared to control on weight and only at the end LiPR could have an effect. On the other hand, after 4 months, HFD mice are clearly above the controls and it is then, albeit when weight gain has reached a plateau, that LiPR has an effect. The election of these arbitrary paradigms and their drawbacks has to be better explained.

ANSWER: We thank the reviewer for the comment. We analyzed the effect of HFD and/or LiPR on the body weight for the 21-day group (Fig.1B) in the original manuscript (lines 111-115). The two-way, repeated measure ANOVA revealed no effect of the treatment on the body weight in the 7-day group, however, it revealed the effect of the duration of treatment on the body weight in the 21-day group. As suggested by the reviewer, we included the Control Diet + LiPR in the 21-day group (Fig.1B). We analyzed the data with ANOVA and found that the treatment has a statistically significant effect on the body weight, however, without any statistical difference between treatment groups (lines 112-116 in the revised manuscript). In addition, we will include the Control Diet + LiPR in the 4-month group.

4. The number of GFAP+ α -tanyocytes is not significantly changed by HFD therefore LiPR does not rescue, but rather increases the number of GFAP+ α -tanyocytes in the 7-day setting. There are no differences among groups later, the effect is lost by 21 days, therefore there is a transient excess of GFAP+ α -tanyocytes which later "disappear" in the LiPR group. The authors state that LiPR rescues the decrease in "htNSCs", but after 21 days the number of the GFAP+ α -tanyocytes is the same in all groups without the need of LiPR. There is no experimental follow up (addressing proliferation and survival of these cells) and the conclusions stated in the text (results and discussion) are not really supported by the data. The *in vitro* experiments could be a complement, but are no substitute for the missing *in vivo* exploration.

ANSWER: We thank the reviewer for this comment. We agree that we did not correctly interpret the data. On line 158, we replaced the original statement "This suggests that short LiPR rescues HFD-induced reduction in the number of htNSCs" with a new one that reflects of date correctly, "This suggests that short LiPR increases the number of htNSCs. In our revision plan, we will quantify the number of proliferating tanyocytes to complement our *in vitro* results.

7. The inclusion of Liraglutide + HFD, (not Liraglutide alone) only in some of the experiments is pointless if there is no direct comparison with LiPR and a timepoint is missing. In S.Fig 3, Fig. 5 and S.Fig 7 LFD (low fat diet?) is used in several occasions as in: "on reducing number of PCNA+ cells in 21d protocol (one-way ANOVA (OWA), $F(2,12) = 16.66$, $p = 0.0003$) when compared to both LFD and HFD groups". Is this the control diet?

ANSWER: We apologize for the confusion caused by labelling the conditions of the Control Diet inconsistently. In some figures (e.g., Fig.2, S.Fig.3, Fig.4), we labelled the Control Diet as "Control", whereas in some other figures (e.g., Fig.5, S.Fig.7) we labelled the Control Diet as "LFD" (Low Fat Diet). In all experiments and figures, the used Control Diet was identical. We unified the labelling of the Control Diet in all figures and in the text of the revised manuscript. Respectfully, we do not agree that including the Liraglutide data is pointless. We included the

Revision Plan

Liraglutide in the context of the HFD as a direct comparison with the HFD + LiPR group to demonstrate that the two anti-obesity compounds exert differential effects on adult neurogenesis. Such comparison has not been done before in analyzing adult neurogenesis and is valuable for better understanding of functions of these anti-obesity compounds.

Minor points:

A. Introduce "GLP-1RA"

ANSWER: We thank the reviewer for identifying this omission. We introduced the term in the revised manuscript (line 50).

B. "HFD-induced inflammation and astrogliosis in the hypothalamus 45,46, whereas the long (4mo) protocol leads to DIO" Are these notions exclusive?

ANSWER: This statement emphasized that HFD-induced inflammation and astrogliosis precede obesity. We prefer to leave the statement as it is.

C. LiPR displays no effects on astrocytes" "Displays" is not the correct term.

ANSWER: We replaced the term "display" with the word "show" in the revised manuscript (line 342).

Reviewer 3

3. In figure 1 I-K images are not clear and better resolution images would help.

ANSWER: We provided images with higher resolution for Figure 1I-K of the revised manuscript.

6. Are Fig 2 A-C images representative of mice fed HFD for 7 days?

ANSWER: Yes, the representative images in panels of Fig. 2A-C are from the 7-day group. However, the legend states that these images are from the 21-day group. This is an error that we corrected in the revised manuscript in the legend of Figure 2 (line 572). We apologize for this and thank the reviewer for double-checking.

7. By looking at figure 2B it seems like the proportion of alpha tanicytes is higher in HFD since no or very few tanicytes are observed and almost all of them are alpha tanicytes.

ANSWER: Indeed, 7 days of HFD reduced the number of alpha 2 tanicytes, which occupy the ventral-lateral aspect of the 3rd ventricle. This reduction of alpha 2 tanicytes drives the lower proportion of GFAP+ alpha-tanicytes out of all GFAP+ tanicytes. We emphasized this in the text of the revised manuscript (line 435-437).

13. There is a large reduction in the number of DCX+ cells from control 21 d treated mice to control 4 month treated mice. Is this statistically significant? How do authors explain this dramatic reduction?

ANSWER: Yes, there is statistically significant reduction in the number of DCX+ cells and DCX+ neurons in the SGZ between the 21-day and 4-month group (S.Fig.2). This reduction is most likely a result of aging. The mice of the 21-day group were around 2.5 months of age when culled, whereas the 4-month group mice were over 6.5-month-old. The decline in SGZ neurogenesis with age is well documented. Because this decrease in DCX+ cells in the SGZ is an obvious consequence of the animals' age and because the hippocampal neurogenesis is not the primary focus of this manuscript, we prefer not to discuss this feature in the manuscript.

14. Authors do not show the effect of HFD on BrdU+ neurons in the Arcuate. However, all data need to be shown.

ANSWER: We stated (on page 12 of the original manuscript) that in the Arcuate Nucleus of the 21-day group, there was "a statistically significant increase of BrdU+ neurons by HFD compared to Control (data not shown)". To satisfy reviewer's comment, we incorporated this data in the S.Fig.5 as the new panel S.Fig.5O and added the following text (lines 309-313) to the revised manuscript: "However, in the ArcN, the primary nutrient and hormone sensing neuronal nucleus of MBH⁴, there was a statistically significant difference in number of BrdU+ neurons due to treatment (OWA, $F(3,15) = 3.97$, $p = 0.0029$). Exposure to 21d HFD significantly increased the number of BrdU+ neurons in the ArcN, which was reversed by co-administration of LiPR or Liraglutide (S.Fig.5O)." In addition, we adjusted the relevant discussion (lines 468-472): "Our results show that the short and intermediate exposure to HFD does not change the number of newly generated, BrdU+ cells, neurons, or astrocytes in the MBH parenchyma, however, it increases the number of BrdU+ neurons in the primary sensing ArcN, which is reversed by the con-current administration of LiPR or Liraglutide" and (lines 474-476): "In addition, our results show that while LiPR does not change the number of new cells in the MBH parenchyma, it can rescue the increased production of new neurons in the ArcN in the context of the intermediate HFD exposure."

4. Description of analyses that authors prefer not to carry out

Please include a point-by-point response explaining why some of the requested data or additional analyses might not be necessary or cannot be provided within the scope of a revision. This can be due to time or resource limitations or in case of disagreement about the necessity of such additional data given the scope of the study. Please leave empty if not applicable.

Reviewer 1

3. A somewhat related issue is that, in most cases, only the percentage or the density of cells are shown on the graphs, rather than the absolute numbers (at least for some cases). This sometimes complicates the comparisons; for instance, does the surface of the hypothalamus change between 2 and 6 months of age? The tanycytes' number stays, apparently, the same (e.g., Fig. 2) but the production of new neurons is supposed to fall dramatically.

ANSWER: We thank the reviewer for this comment. We agree that the quantification of absolute number of cells is the preferable approach that we have used in our previous publications on subventricular (SVZ) or subgranular (SGZ) neurogenesis. However, hypothalamic adult neurogenesis is dispersed over much larger volume of tissue than neurogenesis in the SVZ or SGZ, which is confined to narrow tissue compartments. As we do not have access to a confocal

Revision Plan

microscope with stereological software, absolute quantification in entire MBH is not feasible. Nevertheless, we believe that our quantification of cell density provides an unbiased and informative approach that allowed us to compare the effects of LiPR and diet on the neurogenic process.

6. A sharp increase in PCNA+ cells in the hippocampus at the 21-day time point, both in the control and in the HFD and HFD/LiPR groups (Fig. S2f) is a little puzzling because neither the Dcx+ nor the Ki67+ cells show this increase.

ANSWER: We agree with the reviewer that this increase in the number of PCNA+ cells is puzzling. We quantified the number of PCNA+ cells twice by two different people, always getting the same result. Given that this is a minor result in a supplementary figure, we would prefer not analyzing this again, unless the reviewer would insist on it.

Reviewer 2

3. Why was the proportion of GPR10+BrdU+MAP2+ cells only assessed in control mice and not in the experimental groups if its expression in overall neurons changes? This suggests that the receptor is expressed in neurons. Interestingly, exposure to 21d HFD reduced density of GPR10, which was rescued by LiPR administration (Fig.1L). Why was this time point chosen and not the longer-term one? What is the consequence of the alterations in the potential number of GPR10, specially in relation to the administration of LiPR? This clarification is important because a 14-day treatment was chosen for the *in vitro* experiments in which LiPR, but not HFD, seems to have an effect on cell proliferation. Might be it would have been more useful to use a paradigm in which HFD has an effect to better compare with *in vivo* work and for the rationale of the work. "Besides GPR10, we co-localized neuronal cytoskeleton structures with NPFFR2 in the MBH (Fig.1O-P)..." Why were not GPR10 and NPFF2 analyzed in a similar and consistent manner ? It is confusing.

ANSWER: The proportion of GPR10+BrdU+Map2+ neurons was quantified to address whether new neurons express the PrRP receptor. We chose to analyze the proportion of GPR10+BrdU+Map2+ neurons at the 21d time-point because we had the most robust data for this or related time points *in vitro* and *in vivo*. We will emphasize this in the text. But we prefer not to analyze the effect of LiPR on the density or expression of GPR10 or NPFF2 for all time points. We consider this to be beyond the scope and focus of the manuscript.

8. The final experiment shows that application of hPrRP31, a variation of LiPR, causes an immediate calcium increase in human induced pluripotent stem cell-derived hypothalamic nucleus. This finding is interesting in itself because it brings light about the function of the receptor/s. It would have been very useful to test what other receptors mentioned to bind LiPR is mediating the effect. In any case, the focus of the work are the neural stem/progenitor cells responsible for neurogenesis and the changes in their properties because of HFD and LiPR, therefore I would trade these experiments for a more thorough and detailed dissection of these effects.

ANSWER: We thank the reviewer for recognizing the relevance of the experiments with the hiPSC-derived neurons. As described in the comments above, we will conduct additional experiments to address the effect of LiPR on aNSCs and proliferation to more thoroughly as suggested by the reviewer.

Reviewer 3

2. Body weight is found reduced by LiPR as well as other metabolic parameters in mice treated with LiPR during the last two weeks of the 4 Mo HFD. However, no effects on hypothalamic or SGZ neurogenesis are not observed in this experimental group. How do authors explain this results?

ANSWER: The 4-month group contains animals that are over 6-month-old, which display very low levels of cell proliferation and differentiation in comparison with the 7 and 21-day groups that contain mice that are 2 and 2.5 months old, respectively. It is possible that these low levels of neurogenesis did not allow us to detect any pro-neurogenic effects of LiPR. Alternatively, the low neurogenesis in older animals precludes us from detecting the adverse effects of the HFD, which are rescued by LiPR in younger animals.

5. On figure 2B it seems that is alpha 2 tanicytes that are missing in response to HFD.

ANSWER: Indeed, the panel in Figure 2B shows that the HFD reduces the number of alpha tanicytes, including the alpha 2 tanicytes. This representative image supports our quantification results in Figure 2D-E.

8. In fig 2 d-f, an increase in the number of GFAP+ alpha tanicytes and its proportion as well as labelled with vimentin is observed in control mice fed with normal diet for 7 days compared with mice fed normal diet for 21 days. How do authors explain this difference?

ANSWER: There is no difference in the number of GFAP+ alpha tanicytes or proportion of GFAP+ alpha tanicytes between 7-day and 21-day Control Diet mice. We used the two-way, repeated measure ANOVA with the Bonferroni's pots-hoc test and did not observe any statistical difference between these 2 quantifications for the Control Diet mice at 7 and 21 days. There is a statistical difference between 7-day and 21-day Control Diet mice in the proportion of GFAP+Vimentin+ tanicytes. This could be due to expansion of the Vimentin+ tanicytes in relatively young adult mice. Given that this is not a major point, we prefer not expanding its discussion in the manuscript.

10. Authors test for adult hippocampal neurogenesis in the three groups. DO images in fig S2 correspond to the 21 day treatment group?

ANSWER: Yes, the representative images in the Supplementary Figure 2 are from the 21-day group. This is stated in the figure legend.

11. On fig S2 C, it seems that in HFD fed mice treated with LiPR newly generated neuroblasts are more differentiated have authors looked at DCX+ cell morphology?

ANSWER: We thank the reviewer for this observation. We have not analyzed the morphology of DCX+ cells or DCX+ neuroblasts in the SGZ. As the manuscript focuses on the hypothalamic and not hippocampal neurogenesis, we prefer not to analyze the morphology in the revised manuscript.

Revision Plan

12. In this same figure, it seems like the number of DCX+ neuroblasts and the number of newly generated neurons is reduced in mice of the 21 d group compared to the 7 day group. Is this statistically significant?

ANSWER: We used the two-way, repeated measure ANOVA with the Bonferroni's pots-hoc test to analyze the DCX+ neuroblasts and neurons. We observed a statistically very significant effect of LiPR treatment on the number of DCX+ neuroblasts and neurons (page 10 of the original manuscript). However, the Bonferroni's test did not reveal any difference between 7-day and 21-day treatment groups.

Dear Dr. Petrik,

Thank you for transferring your research manuscript from Review Commons to EMBO reports, along with the comments of the three referees who assessed it for Review Commons, and your revision plan describing additional data that you have already obtained and outlining further experiments that you are willing to perform to address the referees' concerns.

The referees acknowledge the novelty and significance of the findings, but they also identify a number of limitations and raise some shared concerns, and they provide several suggestions for the improvement of the study and the manuscript. In particular, they suggest that a control diet and LiPR treatment group should be analyzed to characterize the effects of LiPR on adult neurogenesis outside the context of the high-fat diet and to facilitate interpretation of the results. They also recommend more in-depth characterization of tanycytes and their proliferation, as well as additional analyses of cell quantification, and they also provide a number of additional suggestions regarding technical aspects of the work, which should be addressed. We also agree with the comments of the referees regarding readability of the manuscript, and we think that the text should be improved (e.g. by clearly explaining the rationale and connections between the presented experiments, avoiding the excessive use of abbreviations, moving results of statistical tests to relevant Figure legends, if possible).

Although we agree with referee #2 that evidence that the observed changes actually participate in food intake regulation would be desirable to considerably strengthen the manuscript, it will not be required for further consideration of your manuscript for publication in EMBO reports.

Given the constructive comments of the referees, we would like to invite you to revise your manuscript with the understanding that the referee concerns (as detailed above and in their reports) must be fully addressed, along the lines of your revision plan, and their suggestions taken on board. Please address all referee concerns in a complete point-by-point response. Acceptance of the manuscript will depend on a positive outcome of another round of review. It is EMBO reports policy to allow a single round of revision only and acceptance or rejection of the manuscript will therefore depend on the completeness of your responses included in the next, final version of the manuscript. If you have any questions or comments, we can also discuss the revisions in a video chat, if you like.

We realize that it is difficult to revise to a specific deadline. In the interest of protecting the conceptual advance provided by the work, we usually recommend a revision within 3 months (August 4th). Please discuss with me the revision progress ahead of this time if you require more time to complete the revisions.

IMPORTANT NOTE:

We perform an initial quality control of all revised manuscripts before re-review. Your manuscript will FAIL this control and the handling will be DELAYED if the following APPLIES:

- 1) If a data availability section providing access to data deposited in public databases is missing.
- 2) If your manuscript contains statistics and error bars based on $n=2$. Please use scatter plots in these cases. No statistics should be calculated if $n=2$.

- 1) A .docx formatted version of the manuscript text (including legends for main figures, EV figures and tables). Please make sure that the changes are highlighted to be clearly visible.
- 2) Individual production quality figure files as .eps, .tif, .jpg (one file per figure). Please download our Figure Preparation Guidelines (figure preparation pdf) from our Author Guidelines pages <https://www.embopress.org/page/journal/14693178/authorguide> for more info on how to prepare your figures.
- 3) A .docx formatted letter INCLUDING the reviewers' reports and your detailed point-by-point responses to their comments. As part of the EMBO Press transparent editorial process, the point-by-point response is part of the Review Process File (RPF), which will be published alongside your paper unless you opt out of this (please see below for further information).
- 4) A complete author checklist, which you can download from our author guidelines (). Please insert information in the checklist

that is also reflected in the manuscript. The completed author checklist will also be part of the RPF (please see below for more information).

5) Please note that all corresponding authors are required to supply an ORCID ID for their name upon submission of a revised manuscript (). Please find instructions on how to link your ORCID ID to your account in our manuscript tracking system in our Author guidelines
()

6) We replaced Supplementary Information with Expanded View (EV) Figures and Tables that are collapsible/expandable online. A maximum of 5 EV Figures can be typeset. EV Figures should be cited as "Figure EV1, Figure EV2" etc... in the text and their respective legends should be included in the main text after the legends of regular figures.

7) Before submitting your revision, primary datasets (and computer code, where appropriate) produced in this study need to be deposited in appropriate public databases (see < <https://www.embopress.org/page/journal/14693178/authorguide#dataavailability>>).

Specifically, we would kindly ask you to provide public access to the following datasets/data:

- RNA-sequencing data

The accession numbers and database should be listed in a formal "Data availability " section (placed after Materials & Methods) that follows the model below (see also < <https://www.embopress.org/page/journal/14693178/authorguide#dataavailability>>):

Data availability

- RNA-seq data: Gene Expression Omnibus GSE46843 (<https://www.ncbi.nlm.nih.gov/geo/query/acc.cgi?acc=GSE46843>)
- [data type]: [name of the resource] [accession number/identifier/doi] ([URL or identifiers.org/DATABASE:ACCESSION])

*** Note: all links should resolve to a page where the data can be accessed. ***

*** Note: the Data Availability Section is restricted to new primary data that are part of this study. ***

8) We request authors to consider both actual and perceived competing interests. Please review the new policy () and update your competing interests statement if necessary. Please name this section 'Disclosure and competing interests statement' and place it after the Acknowledgements section.

9) Figure legends and data quantification:

- the name of the statistical test used to generate error bars and P values,
- the number (n) of independent experiments (please specify technical or biological replicates) underlying each data point,
- the nature of the bars and error bars (s.d., s.e.m.)
- If the data are obtained from n {less than or equal to} 2, use scatter plots showing the individual data points.

Discussion of statistical methodology can be reported in the Materials and Methods section, but figure legends should contain a basic description of n, P and the test applied.

10) We now request publication of original source data with the aim of making primary data more accessible and transparent to

the reader. Our source data coordinator will contact you to discuss which figure panels we would need source data for and will also provide you with helpful tips on how to upload and organize the files.

11) Our journal encourages inclusion of *data citations in the reference list* to directly cite datasets that were re-used and obtained from public databases. Data citations in the article text are distinct from normal bibliographical citations and should directly link to the database records from which the data can be accessed. In the main text, data citations are formatted as follows: "Data ref: Smith et al, 2001" or "Data ref: NCBI Sequence Read Archive PRJNA342805, 2017". In the Reference list, data citations must be labeled with "[DATASET]". A data reference must provide the database name, accession number/identifiers and a resolvable link to the landing page from which the data can be accessed at the end of the reference. Further instructions are available at .

12) Please also note our reference format:

13) We now use CRediT to specify the contributions of each author in the journal submission system. CRediT replaces the author contribution section, which should be removed from the manuscript. Please use the free text box to provide more detailed descriptions. See also guide to authors:

14) As part of the EMBO publications' Transparent Editorial Process, EMBO reports publishes online a Review Process File to accompany accepted manuscripts. This File will be published in conjunction with your paper and will include the referee reports, your point-by-point response and all pertinent correspondence relating to the manuscript.

You can opt out of this by letting the editorial office know (emboreports@embo.org). If you do opt out, the Review Process File link will point to the following statement: "No Review Process File is available with this article, as the authors have chosen not to make the review process public in this case."

I look forward to seeing a revised version of your manuscript when it is ready. Please let me know if you have any questions or comments regarding the revision.

Yours sincerely,

Ioannis Papaioannou, PhD
Editor
EMBO reports

Authors' Response

Manuscript number: EMBOR-2023-57414V1 (Review Commons RC-2023-01900)

Corresponding author(s): David Petrik

General Statement

We thank the reviewers for a thorough review that helped us to improve the manuscript in the revision process. In our opinion, all three reviewers found the manuscript interesting, novel, and relevant for a broader readership. The reviewers suggested performing additional analyses of cell quantification from existing brain tissue or from newly generated tissue. All reviewers identified several shared concerns that we were happy to address by additional experiments and analyses to improve our manuscript.

First, the reviewers suggested expanding analyses for the Control Diet + LiPR treatment group to further characterize the effects of LiPR on adult neurogenesis outside the context of the High Fat Diet (HFD). To address this, we generated new mice for the 4-month time point and quantified tanycytes, proliferating cells and BrdU-retaining neurons and other cells in the Medial Basal Hypothalamus 21-day and 4-month time points. Our new analyses in Figures 2 and 4 and Extended View (EV) Figures 1 and 5 show that LiPR maintains some neurogenic effects outside the HFD exposure. This suggests that PrRP signaling is generally neurogenic. We provide more details in individual responses to reviewer's comments below.

Second, the reviewers suggested building upon the analysis of tanycytes and cell proliferation. We extended the analysis of tanycytes and proliferating cells and included the new results in Figure 2. These new results show that LiPR rescues HFD-driven decrease in the beta-tanycytes, which are neurogenic, and that LiPR treatment decreases cell proliferation in the MBH in the context of both the HFD and Control Diet. Taken together, these results suggest that LiPR reduces cell proliferation and increases populations of hypothalamic adult Neural Stem Cells.

Finally, to improve readability and to adhere to EMBO Reports formatting, we moved the specific results of statistical analyses (i.e., F and P values) to the figure legends. To comply with the EMBO Reports formatting, we also re-labeled the original Supplementary Figures 1-5 as Extended View Figures 1-5 (Fig EV 1-5) and combined the original Supplementary Figures 6-8 to one Appendix Figure 1-3.

All changes in the text of the revised manuscript are highlighted by red font to simplify identifying these changes. Below we address reviewer's concerns point-by-point. We are confident that our new results satisfied most of the concerns, improved our manuscript and made it suitable for publication in the EMBO Reports.

Reviewer 1

In this manuscript, Jorgensen and colleagues describe their findings on the action of a palmitoylated form of prolactin-release peptide (LiPR) on neural stem cells (NSC) in the adult mouse hypothalamus and adult mouse hippocampus. Their main conclusion is that LiPR can counteract the effects of high-fat diet (HFD) and rescue some of the adverse effects of HFD. Specifically, the authors provide evidence that:

- Exposure to HFD reduces the number of presumptive adult neural stem cells (NSCs) in the

adult hypothalamus, whereas exposure to LiPR reverses this trend.

- The results suggest that LiPR reduces the proliferation of alpha-tanycytes and/or their progeny in the hypothalamus in the context of HFD, with Liraglutide acting similarly. In contrast, while LiPR also suppresses proliferation in the SGZ, Liraglutide works there in the opposite direction.

- LiPR also helps the survival of adult-born hypothalamic neurons.

- Reduction of proliferation by LiPR suggests a model where LiPR increases the number of NSCs presumably by reducing their rate of activation.

- The results suggest that LiPR promotes expression of PrRP receptors in the hypothalamic neurons, suggesting that PrRP may act directly on such neurons (and tanycytes?) in vivo.

- The authors also show that HFD and LiPR alter gene expression profiles of the MBH cells, with HFD, but not LiPR, inducing myelination-related genes.

- Finally, they show that PrRP stimulates an increase in Ca²⁺ in in vitro-derived human hypothalamic neurons.

- The authors conclude that LiPR may be reducing activation and proliferation of the hypothalamic stem cells and thereby preserve their pool from exhaustion, which was stimulated by HFD.

The manuscript presents interesting data and is clearly written. There are several comments, mainly editorial.

RESPONSE: We thank the reviewer for the favorable and positive assessment of our manuscript and for finding our study to be interesting to a broad audience and well written, with most comments described by the reviewer as “editorial”. Below, we address the reviewer’s concerns in a detailed revision plan.

1. It is unclear why most of the experiments do not include the control+LiPR group. Even though the focus of the study was the action of LiPR in the context of HFD, questions remain regarding the action of LiPR per se. Is LiPR (or Liraglutide, for that matter) completely inactive on the normal diet background, with respect to neurogenesis in the hypothalamus and the hippocampus? Whether the Response is positive or negative, it would give a much better understanding of the action of LiPR - does it regulate neurogenesis in various physiological contexts, or does it only kick in with a particular type of diet? In fact, this was examined (see Supplementary figures), but only for the cells in culture and, when performed with animals, was limited to 7 and 21 days, rather than 4 months, which would have been much more informative.

RESPONSE: We thank the reviewer for this suggestion. We agree that including the Control Diet + LiPR group for the 4-month HFD group complements the results from the 7 and 21 days. We generated new animals for the 4-month group that were exposed to Control Diet in their entire life and to LiPR in the last 2 weeks to complement the other three treatment groups in this time point. In addition, we further analyzed the Control Diet + LiPR brains from the 21-day group. At both time-points (21-day and 4-month) for the Control Diet + LiPR mice, we quantified the number of proliferating cells, tanycytes and new, adult-generated neurons in the MBH and the Median Eminence (ME). New data from these analyses are presented in Figure 1B-C (body weight), Figure EV1A-G, K-N (quantification of tanycytes and proliferating cells), Figure EV5 (BrdU+ neurons, astrocytes, and cells). In these new analyses, we observed that LiPR co-administered with the Control Diet increases the number of alpha-tanycytes (Fig EV1E), reduces cell proliferation (Fig EV1M-N) and increases the number of new BrdU+ neurons in the MBH (Fig EV5P-T). We incorporated this new data in the Results on page 8 and 13 and in the Discussion on page 21-22. Taken together, these new results suggest that LiPR exerts some of its neurogenic effects also in physiological conditions of Control Diet.

2. The question above is also relevant when considering the conclusions on the potential depletion of the stem cell pool (again, whether in the hypothalamus or the hippocampus), particularly at the 4-month time point. The mice are ~6 months old by that time, and neurogenesis in both regions is expected to decrease by that time. Are LiPR or Liraglutide able to suppress or exacerbate this decrease? Can they be used to mitigate this decrease when mice are on a regular diet?

RESPONSE: This concern is addressed by analyzing the Control + LiPR mice for the 4-month HFD group (see our response to point 1 above). We analyzed the alpha- and beta-tanycytes in the Hypothalamic Ventricular Zone (HVZ) and in the ME of these mice to address whether LiPR treatment changes the time-dependent decrease in both cell populations (Fig EV1). Our new results suggest that LiPR promotes adult neurogenesis by reducing cell proliferation and extending self-maintenance of htNSCs.

3. A somewhat related issue is that, in most cases, only the percentage or the density of cells are shown on the graphs, rather than the absolute numbers (at least for some cases). This sometimes complicates the comparisons; for instance, does the surface of the hypothalamus change between 2 and 6 months of age? The tanycytes' number stays, apparently, the same (e.g., Fig. 2) but the production of new neurons is supposed to fall dramatically.

RESPONSE: We thank the reviewer for this comment. We agree that the quantification of absolute number of cells is the preferable approach that we have used in our previous publications on subventricular (SVZ) or subgranular (SGZ) neurogenesis. However, hypothalamic adult neurogenesis is dispersed over a much larger volume of tissue than neurogenesis in the SVZ or SGZ, which is confined to narrow tissue compartments. As we do not have access to a confocal microscope with stereological software, absolute quantification in entire MBH is not feasible. Nevertheless, using 10X objective on our confocal microscope, we quantified the area covered by the MBH in 21d and 4mo groups (see the graph below). We did not observe any statistical difference between any treatment groups at both time points suggesting that the surface or volume of the MBH does not change with treatment or time. Thus, we believe that our quantification of cell density provides an unbiased and informative approach that allowed us to compare the effects of LiPR and diet on the neurogenic process.

4. The authors write "LiPR may prevent stem cells from exhaustion, induced by HFD" - but it is not clear that HFD indeed leads to exhaustion - there is no statistically significant difference in the number of the stem cells (alpha-tanycytes) between the control and HFD or between HFD at 1, 3, or 12 weeks.

RESPONSE: We thank the reviewers for their insights. We agree that our initial quantification of alpha-tanycytes only did not show that HFD exhausts tanycytes. We over-interpreted the data.

In the revision, however, we show that LiPR rescues HFD-driven decrease in beta-tanycytes, which show neurogenic potential and higher cell quiescence than alpha-tanycytes (Haan *et al.*, 2013). These new results support our original statement, which we modified as following (line 458): “The lower cell activation may be protective against adverse effects of the HFD, which is reflected in the rescue of the number of β -tanycytes by LiPR.”

5. Numerous papers show that the rate of production of new adult hypothalamic neurons (mainly those derived from beta-tanycytes) drops drastically within the first several weeks of mouse life. Does HFD accelerate, and LiPR mitigate, this decrease? Perhaps one can calculate the numbers from the graphs, but it would help if this is explained in the text of the manuscript. Also, it is not always clear whether specific experiments were performed with the zones of the hypothalamic wall that only contain alpha-tanycytes.

RESPONSE: The literature on the temporal extension of hypothalamic adult neurogenesis is not uniform. It appears that there is a substantial heterogeneity in the neurogenic potential of hypothalamic adult Neural Stem Cells (htNSCs). For example, Fgf10-positive beta tanycytes generate as robust output of neurons at P400 as they do at P30 (Haan *et al.*, 2013). Nevertheless, our results show that LiPR rescues the HFD-induced reduction in adult-generated hypothalamic neurons only in the context of 4-month HFD but not in the 7- and 21-day HFD. In the methods (line 962), we specify that “the Region of Interest (ROI) quantified included the MBH parenchyma with the Arcuate (Arc), DMN and Ventromedial (VMN) Nuclei and the Medial Eminence (ME)”. In the results of the revised manuscript (lines 300-301), we highlighted the areas of the ROI. Upon the request of Reviewer 3 (comment 14), we included new data on quantification of BrdU+ neurons in the Arcuate Nucleus (Fig EV5O). This data show that 21d HFD increases the number of new neurons in Arc, which is reversed by LiPR or Liraglutide (text added to results and discussion on lines 306-309 and 488-491, respectively). Finally, in the Discussion (lines 493-498), we point out that the neurogenic potential of LiPR occurs both in Control and HFD conditions, however, is observed only in older mice of the 4-month group and not in the younger mice of the 7- and 21-day groups.

6. A sharp increase in PCNA+ cells in the hippocampus at the 21-day time point, both in the control and in the HFD and HFD/LiPR groups (Fig. S2f) is a little puzzling because neither the Dcx+ nor the Ki67+ cells show this increase.

RESPONSE: We agree with the reviewer that this increase in the number of PCNA+ cells is puzzling. We observed a similar trend also in Ki67+ cells in the MBH, although there is no statistically significant difference between the number of Ki67+ cells in Controls at 7-day and 21-day time points (Fig 2R). It is possible that the daily i.p. injections are stressful and decrease number of proliferating cells in the 7-day protocol. When animals habituate to the stress during the 21-day protocol, the proliferation is restored. We quantified the number of PCNA+ cells in the SGZ and Ki67+ cells in the MBH twice by two different people blinded to the treatment groups, always getting the same result. Given that the PCNA quantification in the SGZ is not a major point in the main figures, we would prefer not analyzing this again.

7. The study deals with several agents and several processes; a simple scheme that summarizes authors' conclusions might help to better understand the relationships between those agents and processes.

RESPONSE: We thank the reviewer for this useful suggestion. We made a summarizing schematic for the revised manuscript as a part of Appendix Figure 4.

Referee cross-commenting

I agree, the lack of the LiPR group complicates the interpretation of the results. I also agree that the experiments with vimentin staining, calcium increase, and even with neurospheres do not add much to the main questions that this study attempts to Response, and I'd rather see a more thorough analysis of the activation and differentiation data. I also want to reiterate that the concept of LiPR/PrRP preventing the exhaustion of the hypothalamic stem cell pool is not clear, because it is not shown that this pool does actually get exhausted under normal or HFD conditions. This latter issue again requires the LiPR-alone group.

Also, as a clarification - I wrote about 1 month required to complete the revision assuming that the authors actually have the data on the Control+LipR group or at least the specimens available, mainly because the supplementary material shows results with this group, at least with the neurospheres. If this group is fully missing, then the effort will obviously take a longer time.

Reviewer #1 (Significance (Required)):

The provided evidence suggests, for the first time, that PrRP prevents the loss of the neural stem cells population in the adult hypothalamus that was diminished by obesity and HFD. This finding might be interesting to a broad audience.

Reviewer 2

The authors examine the effect of an anorexigenic drug, LiPR in the context of treatment with high fat diet (HFD) and with a special focus on hypothalamic neural stem/progenitor cells and neurogenesis. The work is mostly based on mice and a barrage of different techniques (confocal imaging, cell cultures with time lapse, gene expression...) are used.

The results are interesting because they address the yet-poorly understood implication of hypothalamic neurogenesis in food intake and energy balance.

The results point at complex effects at different levels (neural stem cells, neurons, division, survival...). The experimental approach is sometimes thorough in the treatment of details on the one hand, it also lacks of consistency on the other, and as a result the conclusions lack strength. There is a number of experiments that sometimes seem unrelated and this hurts the comprehension of the manuscript, specially in lieu of the complexity of the results obtained.

RESPONSE: We thank the reviewer for finding our results interesting and relevant. We will strive to improve the consistency of our results in the revised manuscript to satisfy the reviewer's concerns.

1. A major issue is the lack of a LiPR-only group, which would much facilitate the interpretation of the results. The effect of LiPR alone is however tested, but only in comparison with the Control in one of the in vitro experiments (S.Fig. 3)

RESPONSE: We agree with the reviewer that expanding on the LiPR-only effect would facilitate the interpretation of the results (see concern 1 of reviewer 1 and 3). We want to emphasize, however, that we analyzed the HFD-independent LiPR effects not only *in vitro* but also *in vivo*

by quantifying the number of BrdU+ cells and neurons in the MBH of mice exposed to 21-day HFD (originally in S.Fig. 5O-Q, now in panels Fig EV5R-T) and by including the Control Diet + LiPR in the RNAseq experiment (Fig 5C). In the revision, we generated new experimental animals and further expanded the analysis of Control+LiPR mice to the 4-month time point and to tanycytes and proliferating cells (EV1A-G, K-N) and to BrdU-retaining neurons and cells for both 21-day and 4-month groups (Figure EV5P-T). Our new results suggest that LiPR without the HFD increases the number of alpha-tanycytes, reduces cell proliferation and increases the number of new BrdU+ neurons in the MBH suggesting it exerts neurogenic effects also outside the context of obesity. We incorporated this new data in Results on page 8 and 13 and in Discussion on page 22. Taken together, these new results suggest that LiPR exerts some of its neurogenic effects also in physiological conditions of Control Diet.

2. As plotted, in Fig 1B is difficult to interpret the effect of HFD and LiPR, might be using percentage and noting the statistical differences as in the other would help. It looks like HFD has no effect compared to control on weight and only at the end LiPR could have an effect. On the other hand, after 4 months, HFD mice are clearly above the controls and it is then, albeit when weight gain has reached a plateau, that LiPR has an effect. The election of these arbitrary paradigms and their drawbacks has to be better explained.

RESPONSE: We thank the reviewer for the comment. In addition to the justification for the chosen HFD protocols provided at the beginning of the Result section (lines 121-124), we added the following statement to further explain the experimental design (lines 124-126): "The short and intermediate HFD protocols allowed us to determine the effects of LiPR in the context of neuroinflammation and developing metabolic syndrome, whereas the long protocol provided the context of developed obesity."

We analyzed the effect of HFD and/or LiPR on the body weight for the 21-day group (Fig 1B) in the original manuscript (lines 111-115). The two-way, repeated measure ANOVA revealed no effect of the treatment on the body weight in the 7-day group, however, it revealed the effect of the duration of treatment on the body weight in the 21-day group. As suggested by the reviewer, we included the Control Diet + LiPR in the 21-day group (Fig 1B). We analyzed the data with ANOVA and found that the treatment has a statistically significant effect on the body weight, however, without any statistical difference between treatment groups (lines 128-129 in the revised manuscript). In addition, we included the body weight of Control Diet + LiPR mice in the 4-month group (Fig 1C) and observed no statistical difference when compared to Controls suggesting that LiPR reduces body weight only in the context of fully developed obesity.

3. Why was the proportion of GPR10+BrdU+MAP2+ cells only assessed in control mice and no in the experimental groups if its expression in overall neurons changes? This suggests that the receptor is expressed in neurons. Interestingly, exposure to 21d HFD reduced density of GPR10, which was rescued by LiPR administration (Fig.1L). Why was this time point chosen and not the longer-term one? What is the consequence of the alterations in the potential number of GPR10, specially in relation to the administration of LiPR? This clarification is important because a 14-day treatment was chosen for the in vitro experiments in which LiPR, but not HFD, seems to have an effect on cell proliferation. Might be it would have been more useful to use a paradigm in which HFD has an effect to better compare with in vivo work and for the rationale of the work. "Besides GPR10, we co-localized neuronal cytoskeleton structures with NPFFR2 in the MBH (Fig.1O-P)..." Why were not GPR10 and NPFF2 analyzed in a similar and consistent manner ? It is confusing.

RESPONSE: The proportion of GPR10+BrdU+Map2+ neurons was quantified to address whether new neurons express the PrRP receptor. We chose to analyze the proportion of GPR10+BrdU+Map2+ neurons at the 21d time-point because we had the most robust data for this or related time points *in vitro* and *in vivo*. We will emphasize this in the text. But we prefer not to analyze the effect of LiPR on the density or expression of GPR10 or NPFF2 for all time points. We consider this to be beyond the scope and focus of the manuscript.

*4. The number of GFAP+ α -tanycytes is not significantly changed by HFD therefore LiPR does not rescue, but rather increases the number of GFAP+ α -tanycytes in the 7-day setting. There are no differences among groups later, the effect is lost by 21 days, therefore there is a transient excess of GFAP+ α -tanycytes which later "disappear" in the LiPR group. The authors state that LiPR rescues the decrease in "htNSCs", but after 21 days the number of the GFAP+ α -tanycytes is the same in all groups without the need of LiPR. There is no experimental follow up (addressing proliferation and survival of these cells) and the conclusions stated in the text (results and discussion) are not really supported by the data. The *in vitro* experiments could be a complement, but are no substitute for the missing *in vivo* exploration.*

RESPONSE: We thank the reviewer for this comment. We agree that we did not correctly interpret the original data on alpha-tanycytes. However, our new data show that LiPR rescues HFD-driven decrease in beta-tanycytes, which have neurogenic potential and higher cell quiescence than alpha-tanycytes (Haan *et al.*, 2013). These new results support our original statement, which we modified as following (on line 458): "The lower cell activation may be protective against adverse effects of the HFD, which is reflected in the rescue of the number of β -tanycytes by LiPR." In addition, we discuss the effects of LiPR on different tanycyte populations in the Discussion (page 18).

5. The fact that cell division is "rarely found" (Rax GFAP) experiments also push for further investigation. It is difficult to see that relevance of the inclusion of the vimentin staining experiment if there is no further exploration. The effect of LiPR is only transient, in the 7-day paradigm and as the parameter evaluated is the proportion of vimentin+ tanycytes among GFAP+ tanycytes it could only be reflecting increased expression of the filament. "Nevertheless, we did not observe a statistically different change in the area occupied by Rax+ tanycytes (Fig.2H)." Why did the authors use Rax only for this experiment if "GFAP+ α -tanycytes which are considered the putative htNSCs?" What is the justification for not seeing changes in relation to the results reported in Fig 2D-F? "Because Vimentin is associated with nutrient transport in cells and with metabolic response to HFD 52-54, we quantified the proportion of GFAP+ tanycytes expressing Vimentin (Fig.2F)." It is difficult to see that relevance of the inclusion of the vimentin staining experiment if there is no further exploration. The effect of LiPR is only transient, in the 7-day paradigm and as the parameter evaluated is the proportion of vimentin+ tanycytes among GFAP+ tanycytes it could only be reflecting increased expression of the filament.

RESPONSE: Because Vimentin is a marker of neural stem cells and alpha tanycytes, we quantified the number of GFAP+Vimentin+ tanycytes to complement the quantification of GFAP+ alpha tanycytes. We are sorry that this was not clear, and we highlighted this connection in the revised manuscript (page 7 and 8). We agree that further quantification of tanycytes would improve our manuscript. Therefore, we added new analyzes tanycytes positive for Rax as the pan-tanycyte marker (Fig 2H-M). Together with more detailed analyzes of beta-tanycytes and Vimentin-expressing tanycytes (Fig 2E-G), these new results allowed us to refine the interpretation of the effects of LiPR on htNSCs. Specifically, we discuss that LiPR increases

the number of the neurogenic beta-tanycytes but not all (Rax+) tanycytes. In the discussion, we hint that there is cell heterogeneity among htNSCs (page 18), however, we did not want to over-extend our interpretation towards one of the competing models of adult hypothalamic neurogenesis that either beta-tanycytes (Haan *et al.*, 2013; Goodman *et al.*, 2020) or alpha-tanycytes (Robins *et al.*, 2013) serve as the primary htNSCs. We did not interpret our result this way because our results do not contain lineage-tracing data.

6. Why there is no Ki67 experiment in the 7-day paradigm if that is the timepoint in which changes in the number or proportion of GFAP+ tanycytes are observed? PCNA was then used but only in the 21-day paradigm. What is the interpretation and relevance of these data? What are the non-htNSCs proliferating cells, whose dynamics are different from the changes in the number or proportion of htNSCs that could be potentially related to changes in mitosis? Again, I think it would be much useful for the work to explore in detail the changes in the putative htNSCs than investing in experiments that only add confusion.

RESPONSE: We apologize if the data presentation is confusing. In the revised manuscript, we added new analyses of Ki67+ cells for the 7-day time point and newly analyzed proliferating cells in the Median Eminence (ME), which contains neurogenic progenitors, for all 3 time points. In the Medial Basal Hypothalamus (MBH) and the ME, many cell types undergo mitosis, including the oligodendrocyte precursor cells, microglia, astrocytes, and infiltrating macrophages. However, characterizing the identify of all these different cell types in response to the HFD and/or LiPR is beyond the scope of this study. To resolve whether HFD and/or LiPR influence proliferating aNSCs, we attempted to specifically quantify proliferating cells in the Hypothalamic Ventricular Zone (HVZ), which consists of Rax+ tanycytes lining the ventral half of the 3rd ventricle surrounded by the MBH. To resolve whether LiPR affects proliferating neural progenitors, we quantified Rax+Ki67+ cells in the ME, which are expected to be mostly neurogenic. However, because htNSCs proliferate so rarely and infrequently, we were able to identify only a single Rax+Ki67+ tanycyte in the HVZ of 20 mice from 4 treatment groups of the 21-day time point. Similarly, in the ME of these 20 mice, we were able to identify only 2 Rax+Ki67+ cells. These results suggest that quantifying proliferating htNSCs is not feasible as noted by previous studies (Robins *et al.*, 2013; Haan *et al.*, 2013). We understand that combining different thymidine analogues (e.g., EdU and BrdU) in pulse-chase paradigms with shorter chase periods could potentially provide us with some insight into the proliferation dynamics of tanycytes. However, because the focus of this manuscript is on the overall neurogenic effects of anti-obesity compounds rather than just stem cell dynamics, we opted not to include these additional extensive experiments in our revision. On the other hand, our results from *in vitro* time-lapse imaging of MBH-derived cells show that LiPR decreases their proliferation and activation. We extensively discuss the effects of LiPR and HFD on proliferation and cell activation of htNSCs (page 18) and emphasize that changes in cell activation rather than proliferation explain the changes in number of tanycytes.

7. The inclusion of Liraglutide + HFD, (not Liraglutide alone) only in some of the experiments is pointless if there is no direct comparison with LiPR and a timepoint is missing. In S.Fig 3, Fig. 5 and S.Fig 7 LFD (low fat diet?) is used in several occasions as in: "on reducing number of PCNA+ cells in 21d protocol (one-way ANOVA (OWA), $F(2,12) = 16.66$, $p = 0.0003$) when compared to both LFD and HFD groups". Is this the control diet?

RESPONSE: We apologize for the confusion caused by labelling the conditions of the Control Diet inconsistently. In some figures (e.g., Fig 2, Fig EV3, Fig 4), we labelled the Control Diet as "Control", whereas in some other figures (e.g., Fig.5, S.Fig.7) we labelled the Control Diet as

“LFD” (Low Fat Diet). In all experiments and figures, the used Control Diet was identical. We unified the labelling of the Control Diet in all figures as “Control” and in the text of the revised manuscript. Respectfully, we do not agree that including the Liraglutide data is pointless. We included the Liraglutide in the context of the HFD as a direct comparison with the HFD + LiPR group to demonstrate that the two anti-obesity compounds exert similar effects on adult neurogenesis, especially on cell proliferation and tanocytes. Such comparison has not been done before in analyzing adult neurogenesis and is valuable for better understanding of functions of these anti-obesity compounds.

8. The final experiment shows that application of hPrRP31, a variation of LiPR, causes an immediate calcium increase in human induced pluripotent stem cell-derived hypothalamic nucleus. This finding is interesting in itself because it brings light about the function of the receptor/s. It would have been very useful to test what other receptors mentioned to bind LiPR is mediating the effect. In any case, the focus of the work are the neural stem/progenitor cells responsible for neurogenesis and the changes in their properties because of HFD and LiPR, therefore I would trade these experiments for a more thorough and detailed dissection of these effects.

RESPONSE: We thank the reviewer for recognizing the relevance of the experiments with the hiPSC-derived neurons. As described in the comments above, we conducted additional experiments to address the effect of LiPR on htNSCs and cell proliferation to more thoroughly as suggested by the reviewer.

Minor points:

A. Introduce "GLP-1RA"

RESPONSE: We thank the reviewer for identifying this omission. We introduced the term in the revised manuscript (line 53).

B. "HFD-induced inflammation and astrogliosis in the hypothalamus 45,46, whereas the long (4mo) protocol leads to DIO" Are these notions exclusive?

RESPONSE: This statement emphasized that HFD-induced inflammation and astrogliosis precede obesity. We prefer to leave the statement as it is.

C. LiPR displays no effects on astrocytes" "Displays" is not the correct term.

RESPONSE: We removed the term “display” from the revised manuscript.

****Referee cross-commenting****

I think we all referees agree for the most part. The main concern stated by all of us is the lack of a LiPR-alone group. The rest of the concerns are also related or complementary. In my opinion the mostly common view by the referees is reassuring.

Reviewer #2 (Significance (Required)):

The strengths of the work are its novelty in the field and the variety of techniques employed. The

work has the potential of unveiling mechanistic insight into the regulation of neural stem/progenitor cells and neurogenesis. The main audience of this work would be the community working on this field. The lack of experiments testing that the changes observed actually participate in food intake prevent the work from being of relevance for a broader audience (food intake, energy balance, obesity...). The limitations are the descriptive nature of the work and the lack of a consistent and systematic experimental design that would allow to extract solid conclusions upon to which build upon future research.

Reviewer 3

The work of Jørgensen et al describes the effect of a lipidized analogue of the prolactin releasing peptide (LiPR) on the mouse metabolism in response to high fat diet (HFD) and on hypothalamic and subgranular zone (SGZ) neurogenesis. They conclude that LiPR reduces body weight and improves metabolic parameters affected by HFD as well as it concomitantly stimulates neurogenesis in both niches the SGZ and the hypothalamus. The link between both effects is not demonstrated. The work is well conducted, the hypothesis is interesting and the experimental approach is adequate. The scope is wide and results are interesting, however a few aspects need to be further clarified. The manuscript is well written although the modification of some aspects would facilitate the reading such as the use of non described abbreviations for example.

RESPONSE: We thank the reviewer for the positive assessment of our manuscript and for recognizing its novelty and importance for the research in neurogenesis, endocrinology, and metabolism. We will strive to clarify and facilitate our conclusions to improve the manuscript.

1. One concern in this study is the experimental groups. Authors analyze three groups control, HFD and HFD treated with LiPR. Authors conclude that the effects of LiPR are diet independent. However, given the results obtained by the authors on the effect of LiPR, the main question that arises in here is whether LiPR would have an effect on control mice. It seems that a group is missing in the experimental design in which control mice are treated with LiPR during 7, 21 and the last two weeks of the 4 months. Author must include this information or at least argue the election of the experimental design.

RESPONSE: We thank the reviewer for this insight. We agree that including the Control Diet + LiPR in some of our analyses would improve the revised manuscript as also noted by Reviewer 2 (comment 1 and 2) and by Reviewer 3 (comment 1 and 2). In the original manuscript, we included the quantification of BrdU+ cells in the MBH for the Control Diet + LiPR in the 21-day group. To expand on these results, we added new quantifications from the 21-day animals and from newly generated Control Diet + LiPR mice for the 4-month group. New data from these analyses are presented in Figure 1B-C (body weight), Figure EV1A-G, K-N (quantification of tanycytes and proliferating cells), Figure EV5 (BrdU+ neurons, astrocytes, and cells for 21-day and 4-month time-points). In these new analyses, we observed that LiPR co-administered with the Control Diet increases the number of alpha-tanycytes (Fig EV1E), reduces cell proliferation (Fig EV1M-N) and increases the number of new BrdU+ neurons in the MBH (Fig EV5P-T). We incorporated this new data in Results on page 8 and 13 and in Discussion on page 21-22. Taken together, these new results suggest that LiPR exerts some of its neurogenic effects also in physiological conditions of Control Diet.

2. *Body weight is found reduced by LiPR as well as other metabolic parameters in mice treated with LiPR during the last two weeks of the 4 Mo HFD. However, no effects on hypothalamic or SGZ neurogenesis are not observed in this experimental group. How do authors explain this results?*

RESPONSE: The 4-month group contains animals that are over 6-month-old, which display very low levels of cell proliferation and differentiation in comparison with the 7 and 21-day groups that contain mice that are 2 and 2.5 months old, respectively. It is possible that these low levels of neurogenesis did not allow us to detect any pro-neurogenic effects of LiPR. Alternatively, the low neurogenesis in older animals precludes us from detecting the adverse effects of the HFD, which are rescued by LiPR in younger animals.

3. *In figure 1 I-K images are not clear and better resolution images would help.*

RESPONSE: We provided images with higher resolution for Figure 1I-K of the revised manuscript.

4. *Authors conclude that LiPR is increasing the number of NSC by reducing their activation. However, authors show an induced increase in htNSC only in mice fed HFD for 7 days and not in the 21 day fed mice or the 4 mo fed mice (fig 2 d-f). In addition, authors test for the number of cells expressing Ki67 (fig 2 L), however, the number of Ki67+ alpha tanicytes is not shown.*

RESPONSE: We thank the reviewer for this insight, which was also noted by Reviewer 1 (comment 5) and by Reviewer 2 (comment 4). In the revised manuscript, we show that LiPR rescues HFD-driven decrease in beta-tanicytes, which show neurogenic potential and higher cell quiescence than alpha-tanicytes (Haan *et al.*, 2013). This rescue occurs in all three time points (7-day, 21-day and 4-month). These new results support our original interpretation, which we state as following (on line 458): "The lower cell activation may be protective against adverse effects of the HFD, which is reflected in the rescue of the number of β -tanicytes by LiPR." In addition, we discuss the effects of LiPR on NSC activation on pages 17-18. As stated in the response to Reviewer 2 (comment 6), we tried to quantify the number of Ki67+ tanicytes. However, because they proliferate so rarely and infrequently, we did not manage to generate robust and reliable data of Ki67+Rax+GFAP+ tanicytes. In addition, we attempted to quantify Rax+Ki67+ cells in the Median Eminence (ME), which may represent predominantly neurogenic neural progenitors. However, because htNSCs proliferate so rarely and infrequently, we were able to identify only a single Rax+Ki67+ tanicyte in the HVZ of 20 mice from 4 treatment groups of the 21-day time point. Similarly, in the ME of these 20 mice, we were able to identify only 2 Rax+Ki67+ cells. These results suggest that quantifying proliferating htNSCs is not feasible as noted by previous studies (Robins *et al.*, 2013; Haan *et al.*, 2013). On the other hand, our results from *in vitro* time-lapse imaging of MBH-derived cells show that LiPR decreases their proliferation and activation. This suggests that LiPR can reduce proliferation of hypothalamic adult NSCs. We extensively discuss the effects of LiPR and HFD on proliferation and cell activation of htNSCs (page 18) and emphasize that changes in cell activation rather than proliferation explain the changes in number of tanicytes.

5. *On figure 2B it seems that is alpha 2 tanicytes that are missing in response to HFD.*

RESPONSE: Indeed, the panel in Figure 2B shows that the HFD reduces the number of alpha tanicytes, including the alpha 2 tanicytes. This representative image supports our quantification results in Figure 2D-G.

6. Are Fig 2 A-C images representative of mice fed HFD for 7 days?

RESPONSE: Yes, the representative images in panels of Fig 2A-C are from the 7-day group. However, the legend states that these images are from the 21-day group. This is an error that we corrected in the revised manuscript in the legend of Figure 2 (line 607). We apologize for this and thank the reviewer for double-checking.

7. By looking at figure 2B it seems like the proportion of alpha tanicytes is higher in HFD since no or very few tanicytes are observed and almost all of them are alpha tanicytes.

RESPONSE: Indeed, 7 days of HFD reduced the number of alpha 2 tanicytes, which occupy the ventral-lateral aspect of the 3rd ventricle. This reduction of alpha 2 tanicytes drives the lower proportion of GFAP+ alpha-tanicytes out of all GFAP+ tanicytes. We emphasized this in the text of the revised manuscript (line 437-439).

8. In fig 2 d-f, an increase in the number of GFAP+ alpha tanicytes and its proportion as well as labelled with vimentin is observed in control mice fed with normal diet for 7 days compared with mice fed normal diet for 21 days. How do authors explain this difference?

RESPONSE: There is no difference in the number of GFAP+ alpha tanicytes or proportion of GFAP+ alpha tanicytes between 7-day and 21-day Control Diet mice. We used the two-way, repeated measure ANOVA with the Bonferroni's pots-hoc test and did not observe any statistical difference between these 2 quantifications for the Control Diet mice at 7 and 21 days. There is a statistical difference between 7-day and 21-day Control Diet mice in the proportion of GFAP+Vimentin+ tanicytes. This could be due to expansion of the Vimentin+ tanicytes in relatively young adult mice. Given that this is not a major point, we prefer not expanding its discussion in the manuscript.

9. In fig 2 Why are the differences in RAX, KI67 and PCNA only present in mice fed HFD for 21 days?

RESPONSE: We thank the reviewer for this question, which reflects a similar comment 6 of Reviewer 2. To improve consistency of the presented data, we newly quantified Ki67+ cells for all time points (including the 7-day time point) for Fig 2R. In addition, we quantified Ki67+ cells in the Medial Eminence (see response to point 4 above) for Fig 2S. We chose to expand the cell analyses for the 21-day time point (for the new analyses of Rax+ cells, Fig 2L-M and for PCNA+ cells, Fig 2T) because at this point, we observe the most robust changes in tanicytes both *in vivo* and *in vitro*. Analyzing every single cell quantification for every time-point is so extensive that it is beyond the scope of this manuscript.

10. Authors test for adult hippocampal neurogenesis in the three groups. DO images in fig S2 correspond to the 21 day treatment group?

RESPONSE: Yes, the representative images in the Supplementary Figure 2 are from the 21-day group. This is stated in the figure legend.

11. On fig S2 C, it seems that in HFD fed mice treated with LiPR newly generated neuroblasts are more differentiated have authors looked at DCX+ cell morphology?

RESPONSE: We thank the reviewer for this observation. We have not analyzed the morphology of DCX+ cells or DCX+ neuroblasts in the SGZ. As the manuscript focuses on the hypothalamic and not hippocampal neurogenesis, we prefer not to analyze the morphology in the revised manuscript.

12. In this same figure, it seems like the number of DCX+ neuroblasts and the number of newly generated neurons is reduced in mice of the 21 d group compared to the 7 day group. Is this statistically significant?

RESPONSE: We used the two-way, repeated measure ANOVA with the Bonferroni's post-hoc test to analyze the DCX+ neuroblasts and neurons. We observed a statistically very significant effect of LiPR treatment on the number of DCX+ neuroblasts and neurons (page 10 of the original manuscript). However, the Bonferroni's test did not reveal any difference between 7-day and 21-day treatment groups.

13. There is a large reduction in the number of DCX+ cells from control 21 d treated mice to control 4 month treated mice. Is this statistically significant? How do authors explain this dramatic reduction?

RESPONSE: Yes, there is statistically significant reduction in the number of DCX+ cells and DCX+ neurons in the SGZ between the 21-day and 4-month group (the original S.Fig.2, now Fig EV2). This reduction is most likely a result of aging. The mice of the 21-day group were around 2.5 months of age when culled, whereas the 4-month group mice were over 6.5-month-old. The decline in SGZ neurogenesis with age is well documented. Because this decrease in DCX+ cells in the SGZ is an obvious consequence of the animals' age and because the hippocampal neurogenesis is not the primary focus of this manuscript, we prefer not to discuss this feature in the manuscript.

14. Authors do not show the effect of HFD on BrdU+ neurons in the Arcuate. However, all data need to be shown.

RESPONSE: We stated (on page 12 of the original manuscript) that in the Arcuate Nucleus of the 21-day group, there was "a statistically significant increase of BrdU+ neurons by HFD compared to Control (data not shown)". To satisfy reviewer's comment, we incorporated this data in the Fig EV50 and added the following text (lines 306-309) to the revised manuscript: "However, in the ArcN, the primary nutrient and hormone sensing neuronal nucleus of MBH (Betley et al., 2013), treatment significantly influenced number of BrdU+ neurons. 21d HFD increased their number, which was reversed by LiPR or Liraglutide (Fig EV50)." In addition, we adjusted the relevant discussion (lines 488-491).

Reviewer #3 (Significance (Required)):

In general the manuscript includes a great amount of work to demonstrate the effect of LiPR on neurogenesis (hippocampal and hypothalamic). The scope is wide, and the hypothesis is really interesting. Authors may need to solve some issues in order to completely demonstrate their claims and conclusions, but once the work is done, it will be very valuable to understand the effect of pharmacological agents used in the field of endocrinology to treat metabolic disorders such as type 2 diabetes di type 2 diabetes.

So far, no studies have been done in which the effect of this molecules have been described on SGZ and hypothalamic neurogenesis. Both the field of endocrinology and metabolism as well as the field of adult neurogenesis may benefit of a study of this type.

Dear Dr. Petrik,

Thank you for submitting your revised manuscript for consideration by EMBO reports. It has now been seen by the three original referees who were asked to re-evaluate your study, and we have received their reports (included below).

As you will see, referees #1 and #3 acknowledge that most of their concerns have been addressed. However, in the evaluation sheet that the referees return to us along with their reports, referee #1 has rated the technical quality of the study just "adequate" and the general interest "medium". Referee #2 is more critical mentioning that the conclusions of the study are not adequately supported by the rather descriptive data that fail to generate solid conclusions. They explain that this is due to the fact that their previously raised concerns were addressed only partially and insufficiently, referring mainly to the analysis of the LiPR-only group (that was included only in some experiments) and the incomplete characterization of the effects observed in tanycytes/htNSCs. Due to these important criticisms and limitations of the current version, the amount of experimental work likely to be required to address them, and the EMBO reports policy to allow a single round of major experimental revision, I am afraid that we do not think it would be productive to call for another revised version of your manuscript.

Given the potential interest of the findings, we would, however, have no objection to consider a new manuscript on the same topic if at some time in the near future you obtained data that would considerably strengthen the study and address all referees' concerns in full. If you were to send a new manuscript that would address the concerns put forward by the referees, it would be treated as a new submission rather than a revision, and it would be reviewed afresh, also with respect to any novel literature on the topic at the time of submission.

I am sorry to disappoint you on this occasion, but I nevertheless hope that you will find the referees' comments and suggestions helpful in your work. I would like to thank you once again for your interest in our journal and the opportunity to consider your manuscript.

Yours sincerely,

Ioannis Papaioannou, PhD
Editor
EMBO reports

Referee #1:

the authors have answered adequately to most of this reviewer's concerns.

Referee #2:

With a more systematic approach and more focused experimental design the manuscript would be of value because it addresses an interesting topic, the relation between hypothalamic neurogenesis and food intake, specially in the context of obesity. The manuscript fails to generate solid conclusions and only provides scattered descriptive data with weak relation to each other. This is due mostly, but not limited to, to two factors: The lack of a systematic inclusion of the LiPR-only group and the lack of thorough experiments that test the assumptions brought up regularly regarding the properties and changes in tanycytes/htNSCs.

LiPR is exerting actions also in the control non-HFD. But there is no overall effect on weight. These data strongly suggest that neurogenesis is not mediating the anorexigenic effects of LiPR if anything, this needs to be clarified.

The inclusion of the LiPR -alone group has been only in some experiments but it is lacking in others, resulting in weakened conclusions. The addition of this group should systematically added to all the experiments. The lack of effect of LiPR alone on weight in the 4-month paradigm does not justify the lack of this group in the rest of experiments because other clear effects of HFD and HFP+LiPR are found in the 7-day and 21-day paradigms.

A especially notorious case is the effect on alpha-tanycytes. LiPR clearly increases their number in the HFD group in the 7-day paradigm, but HFD decreases it. The response to my comment 4 is not satisfactory. Alterations in beta-tanycytes might explain some of the results regarding total tanycytes or proportions of tanycytes, but not the increase in absolute numbers in alpha-tanycytes. It does not explain either the clear lack of effect on the 21-day and 4-month paradigms. There is data on weight gain,

receptors expression, neurons, tanycytes, htNSCs, in vitro models, gene expression...but the experiments are too disconnected and no functional conclusion can be extracted.

The authors have done new experiments and modified the manuscript but in my opinion the effort falls short of producing solid work and many of the initial concerns remain.

These concerns are highlighted by my following concerns:

What are the conclusions of the experiments regarding receptor expression within the overall potential functional role of LiPR and neurogenesis?

In contrast to alpha-tanycytes, beta-tanycytes are affected in all the time-course paradigms in a similar manner, being decreased by HFD, an increased by LiPR within the context of HFD. The LiPR omission is again flagrant here. The authors even claim later: "These results suggests that some but not all effects of LiPR on tanycytes are independent of HFD". Yet no LiPR-only group is included.

The gliogenic versus the neurogenic capacity of the different populations of tanycytes is pointed out as relevant for the study but it is never addressed.

What are vimentin+ tanycytes? Alpha, beta...? Can they be truly considered to be htNSCs? The results are very similar to those obtained for alpha-tanycytes with only an effect on the 7-day paradigm and only in proportion but not in absolute numbers. How can this be explained?

"and the universal marker of tanycytes, the Retina and anterior neural fold homeobox transcription factor, Rax.." Then how is it possible that Rax+beta-tanycytes or Rax+tanycytes are decreased by HFD but not increased in HFD+LiPR as measured in the other experiments?

"This suggests that LiPR increases number of GFAP+ tanycytes but not all tanycytes positive by default for Rax". Was not Rax an universal marker of tanycytes? Once more, the lack of direct experimental testing generates confusing and lack of valid conclusions.

Results with Ki67 or PCNA show changes in the 21-day paradigm but not in the 7-day paradigm, in opposition to the data from tanycyte populations. How can these changes be reconciled?

Importantly, and this is the main issue, how can all the reported changes in all the different parameters analyzed in vivo and in vitro be linked to the effect on weight gain? The authors appear to aim to link PrRP receptor expression and neurogenesis, through effects on cell division and survival, to the anorexigenic effects of LiPR on tanycytes/htNSCs, but in reality, no actual link is shown. The discussion has been reworked but it still relies more on assumptions than on directly tested data.

In the 21d protocol (Fig 1B), the treatment had a significant effect on the body weight, however, without any statistically significant difference between any of the treatment groups.

I do not understand this sentence. Is there or not an effect?

Referee #3:

The work of Jørgensen et al describes the effect of a lipidized analogue of the prolactin releasing peptide (LiPR) on the mouse metabolism in response to high fat diet (HFD) and on hypothalamic and subgranular zone (SGZ) neurogenesis. They conclude that LiPR reduces body weight and improves metabolic parameters affected by HFD as well as it concomitantly stimulates neurogenesis in both niches the SGZ and the hypothalamus. The work is adequately conducted, the hypothesis is interesting and the experimental approach is adequate. The scope is wide and results are interesting, however a few aspects need to be further clarified. The manuscript is well written. Authors have taken into consideration all indications and comments received by reviewers to their previous version. Additionally, authors have addressed most concerns raised by this reviewer and therefore, I believe the manuscript is now ready to be published in EMBO reports.

Rev_Com_number: RC-2023-01900

New_manu_number: EMBOR-2023-57414V2

Corr_author: Petrik

Title: An analogue of the Prolactin Releasing Peptide reduces obesity and promotes adult neurogenesis

Manuscript EMBOR-2023-57414V2 | [RC-2023-01900] [D]**Rebuttal of Comments of Reviewer #2 to the Revised Manuscript**

Reviewer #2 made several false or misleading claims in the review of our revised manuscript. Based on the detailed analysis of comments of Reviewer #2, we are convinced that Reviewer #2 misled the editors of *EMBO Reports* into rejection of our revised manuscript, which was not fair or balanced. Reviewer #2 claims we did not conduct analyses requested by the reviewer when in fact we did. Also, Reviewer #2 claims our goal was to show that neurogenesis is required for the effects of anti-obesity compounds. This is not true. Our goal was to show for the first time that anti-obesity compounds affect adult neurogenesis. Below, we rebut the comments of Reviewer #2.

Reviewer #2: The manuscript fails to generate solid conclusions and only provides scattered descriptive data with weak relation to each other. This is due mostly, but not limited to, to two factors: The lack of a systematic inclusion of the LiPR-only group and the lack of thorough experiments that test the assumptions brought up regularly regarding the properties and changes in tanycytes/htNSCs.

RESPONSE: The claim that there is a “lack of a systematic inclusion of the LiPR-only group” is false. In the originally submitted manuscript, we analyzed LiPR-only effects not only *in vitro* but also *in vivo* by quantifying the number of BrdU+ cells and neurons in the MBH in the 21-day group (originally in S.Fig. 5O-Q, now in panels Fig EV5R-T) and by analyzing LiPR-only mice in the RNAseq experiment (Fig 5C). In the revised manuscript, to satisfy concern #1 of Reviewer #2, we systematically added analysis of LiPR-only treated mice as requested by Reviewer #2. We analyzed proliferating cells and both alpha- and beta-tanycytes for both 21-day and 4-month groups (EV1A-G, K-N) and BrdU-retaining neurons and cells for both 21-day and 4-month groups (Figure EV5P-T). In fact, we generated a new set of mice for the revision specifically to satisfy the concerns of Reviewer #2. We described the new analyses in the manuscript and in the point-to-point response to Reviewer #2. However, Reviewer #2 falsely claims we never addressed this key concern. In fact, Reviewer #2 contradict their own claims by the next comment where they acknowledge that “LiPR is exerting actions also in the control non-HFD”.

Reviewer #2: LiPR is exerting actions also in the control non-HFD. But there is no overall effect on weight. These data strongly suggest that neurogenesis is not mediating the anorexigenic effects of LiPR if anything, this needs to be clarified.

RESPONSE: The claim that “neurogenesis is not mediating the anorexigenic effects of LiPR” is not substantiated by our results and is misleading. The effects of LiPR on body weight are likely caused by neurogenesis-independent effects. We never claimed that LiPR acts solely through neurogenesis. The goal of our manuscript was not to prove that neurogenesis is required for actions of anti-obesity compounds but that anti-obesity compounds affect adult neurogenesis. To determine that neurogenesis is required for actions of anti-obesity compounds would require specifically ablating neurogenesis and test whether it diminishes anti-obesity effect of the compounds. While this is what we plan to do in the future, it is beyond the scope

of this manuscript as agreed by the managing editor before our revision. Nevertheless, LiPR clearly exhibits cellular effects in the context of both Control and HFD, which is not contradictory. Similarly, Insulin also acts in physiological and obesity conditions.

Reviewer #2: The inclusion of the LiPR -alone group has been only in some experiments but it is lacking in others, resulting in weakened conclusions. The addition of this group should systematically added to all the experiments. The lack of effect of LiPR alone on weight in the 4-month paradigm does not justify the lack of this group in the rest of experiments because other clear effects of HFD and HFD+LiPR are found in the 7-day and 21-day paradigms.

RESPONSE: The claim that we did not include the analysis of LiPR-only analysis substantially is misleading and/or false. Our revised manuscript contains cell analyses of LiPR-only mice for 2 time points (21-day and 4-month) for 5 different data sets: proliferating cells, alpha-tanycytes, beta-tanycytes and BrdU-retaining cells and BrdU-retaining neurons in the Medial Basal Hypothalamus. These cell analyses represent the major cell analyses in the manuscript for a short-term group (21-day) and the long term/obesity group (4-months).

Reviewer #2: A especially notorious case is the effect on alpha-tanycytes. LiPR clearly increases their number in the HFD group in the 7-day paradigm, but HFD decreases it. The response to my comment 4 is not satisfactory. Alterations in beta-tanycytes might explain some of the results regarding total tanycytes or proportions of tanycytes, but not the increase in absolute numbers in alpha-tanycytes. It does not explain either the clear lack of effect on the 21-day and 4-month paradigms. There is data on weight gain, receptors expression, neurons, tanycytes, htNSCs, in vitro models, gene expression...but the experiments are too disconnected and no functional conclusion can be extracted.

RESPONSE: These claims of Reviewer #2 are misleading and/or false. In the comment #4 for to our manuscript before the revision, Reviewer #2 stated that at the 7-day time point "*The number of GFAP+ α -tanycytes is not significantly changed by HFD therefore LiPR does not rescue, but rather increases the number of GFAP+ α -tanycytes*". In our revision, we acknowledged this and clarified our interpretation in the point-by-point response to Reviewer #2: "We thank the reviewer for this comment. We agree that we did not correctly interpret the original data on alpha-tanycytes. However, our new data show that LiPR rescues HFD-driven decrease in beta-tanycytes, which have neurogenic potential and higher cell quiescence than alpha-tanycytes (Haan *et al.*, 2013). These new results support our original statement, which we modified as following (on line 458): "The lower cell activation may be protective against adverse effects of the HFD, which is reflected in the rescue of the number of β -tanycytes by LiPR." In addition, we discuss the effects of LiPR on different tanycyte populations in the Discussion (page 18)." Our ANOVA analysis of alpha-tanycytes shows that treatment has a significant effect on their number (Figure 2E). Because alpha-tanycytes show different neurogenic potential and cell dynamics than beta-tanycytes, it is not surprising that LiPR affects one subpopulation but not the other. Importantly, the goal of our manuscript was not

conducting complex cell lineage-tracing experiments but to demonstrate effects of anti-obesity compounds on tanycytes in general.

MORE SPECIFIC COMMENTS

Reviewer #2: What are the conclusions of the experiments regarding receptor expression within the overall potential functional role of LiPR and neurogenesis?

RESPONSE: LiPR receptors are found both on neural stem cells and new hypothalamic neurons suggesting that LiPR can directly act on different stages of the neurogenic process. These conclusions are in the Discussion on page 17 of the revised manuscript.

Reviewer #2: In contrast to alpha-tanycytes, beta-tanycytes are affected in all the time-course paradigms in a similar manner, being decreased by HFD, and increased by LiPR within the context of HFD. The LiPR omission is again flagrant here. The authors even claim later: "These results suggest that some but not all effects of LiPR on tanycytes are independent of HFD". Yet no LiPR-only group is included.

RESPONSE: As above, Reviewer #2 states "The LiPR omission is again flagrant here". This claim is false. We analyzed both alpha-tanycytes and beta-tanycytes (and the proportion of Vimentin+GFAP+ tanycytes) in LiPR-only animals for 2 time-points (21-day and 4-month) in Figure EV 1E-G.

Reviewer #2: The gliogenic versus the neurogenic capacity of the different populations of tanycytes is pointed out as relevant for the study but it is never addressed.

RESPONSE: The goal of this study was not to dissect specific cell potency of different tanycyte subpopulations but to show for the first time that anti-obesity compounds promote adult neurogenesis.

Reviewer #2: What are vimentin+ tanycytes? Alpha, beta...? Can they be truly considered to be htNSCs? The results are very similar to those obtained for alpha-tanycytes with only an effect on the 7-day paradigm and only in proportion but not in absolute numbers. How can this be explained?

RESPONSE: As described in detail in Results, Discussion and Methods, Vimentin+ tanycytes in our analyses are alpha-tanycytes that express GFAP and Vimentin. As in the response above, the goal of this manuscript was not to dissect the nature of htNSCs but to observe effects of anti-obesity compounds on tanycytes and adult neurogenesis. This comment of Reviewer #2 does not diminish or contradict the general interpretation of our results.

Reviewer #2: "and the universal marker of tanycytes, the Retina and anterior neural fold homeobox transcription factor, Rax.." Then how is it possible that Rax+beta-

tanycytes or Rax+tanycytes are decreased by HFD but not increased in HFD+LiPR as measured in the other experiments? "This suggests that LiPR increases number of GFAP+ tanycytes but not all tanycytes positive by default for Rax". Was not Rax an universal marker of tanycytes? Once more, the lack of direct experimental testing generates confusing and lack of valid conclusions.

RESPONSE: Rax labels all tanycytes and some ependymocytes. It is not surprising that HFD but not LiPR exhibit different effects on different subpopulations of tanycytes.

Reviewer #2: Results with Ki67 or PCNA show changes in the 21-day paradigm but not in the 7-day paradigm, in opposition to the data from tanycyte populations. How can these changes be reconciled?

RESPONSE: Ki67+ and PCNA+ cells include not only cells in the neurogenic process but other proliferating cells in the hypothalamus including microglia, macrophages, oligodendrocyte precursors – as we acknowledge this in the results and the discussion of the revised manuscript. Therefore, cell dynamics of cell proliferation is likely different for all proliferating cells than only for tanycytes or their daughter cells. These results are not contradictory.

Reviewer #2: Importantly, and this is the main issue, how can all the reported changes in all the different parameters analyzed in vivo and in vitro be linked to the effect on weight gain? The authors appear to aim to link PrRP receptor expression and neurogenesis, through effects on cell division and survival, to the anorexigenic effects of LiPR on tanycytes/htNSCs, but in reality, no actual link is shown. The discussion has been reworked but it still relies more on assumptions than on directly tested data.

RESPONSE: The claim that "The authors appear to aim to link PrRP receptor expression and neurogenesis, through effects on cell division and survival, to the anorexigenic effects of LiPR on tanycytes/htNSCs" is wrong and misleading. We did not aim to show that LiPR acts (solely) through neurogenesis but to determine whether anti-obesity compounds affect adult neurogenesis.

Reviewer #2: In the 21d protocol (Fig 1B), the treatment had a significant effect on the body weight, however, without any statistically significant difference between any of the treatment groups. I do not understand this sentence. Is there or not an effect?

RESPONSE: As explained in detail in Results and Methods, our group data were analyzed by the Analysis of Variance (ANOVA), or its non-parametric alternatives, with post-hoc Bonferroni test comparing individual data sets. Two-way ANOVA used for data in Fig 1B (and also Fig 1A and 1C) showed that treatment (that is exposure to Control or HFD and/or administration of LiPR) has a significant effect on the body weight. Detailed P and F values are provided in the legend of Figure 1. The post-hoc Bonferroni tests revealed, however, that there was no significant difference between any of the treatment groups suggesting that, for example, HFD or HFD+LiPR did not

differ from Controls. We presented our statistical analyses clearly and according to the standard format. It is confusing and disappointing that Reviewer #2 was not able to follow it.

Dear Dr. Petrik,

Thank you for the letter sent to my colleague Ioannis in response to his editorial decision letter sent over the summer. I am writing as Ioannis has now moved on to an editorial role at our sister journal EMBO Journal. Where indicated, I will be sure to involve Esther in discussion of the manuscript, as requested by you.

We appreciate your analysis of the report of referee 2.

I should point out that while Ioannis rejected this version of the paper, he left open further consideration by this journal of a future revised version of the manuscript with a more developed dataset.

I should also point out that elements of ref 2's report - in particular on LiPR-only controls - was also echoed by referee 1. While I of course understand that ref 1 and 3 endorse publication of the manuscript, the issues that referee 2 raises are in my view mostly accurate and reasonable. We do not feel that there is compelling evidence that the referee has a political motive against the work or that s/he has shown gross incompetence that would warrant dismissing the report - in contrast the referee actually highlighted in both rounds of review the potential interest of the work after revision.

However, it would be very useful if you could please provide a point by point rebuttal - in particular, we try to understand why a full set of control experiments as requested by ref 1 (cf. point 1 of the original report), 2 and 3 (cf. point 1 of the original report) is not essential.

I want to emphasize that at the editorial level we appreciate the very significant interest of the claims made - we are just trying to assure that the claims are based on definitive data and therefore compelling.

best wishes,

Bernd Pulverer

~~~~~  
Bernd Pulverer, Ph.D.  
Chief Editor, EMBO Reports  
EMBO  
Meyerhofstrasse 1, D-69117 Heidelberg  
Tel: +4962218891501  
bernd.pulverer@embo.org  
~~~~~

Manuscript EMBOR-2023-57414V4 | [RC-2023-01900]**Rebuttal of Comments of Reviewer #2 to the Revised Manuscript (EMBOR-2023-57414V2) and of Comment 1 of Reviewer #1, #2 and #3 to the Original Manuscript (EMBOR-2023-57414V1).**

On 20.9.2023, Dr. Pulverer, Editor-in-Chief of the EMBO Reports, requested that we provide “a point-by-point rebuttal - in particular, we try to understand why a full set of control experiments as requested by ref 1 (cf. point 1 of the original report), 2 and 3 (cf. point 1 of the original report) is not essential.”

We think that a full set of these control experiments is essential, indeed. This is why we generated new animals and performed new analyses in our revised manuscript. These new data are included in Figure 1B-C (body weight), Figure EV1 and Figure EV5 and in the Results on page 8 and 13. While Reviewer #1 and #3 recognized that we provided these control experiments and recommended our manuscript for publication, Reviewer #2 claimed that we did not perform these experiments. We do not understand why these claims were made.

On 14.9.2023, I sent a point-by-point rebuttal of comments of Reviewer #2 to our revised manuscript to contacts@emboreports.org. This rebuttal is now included with our revised manuscript files under the new manuscript number EMBOR-2023-57414V4 as a PDF file “Rejection ReviewerX2 rebuttal September 2023”. In this rebuttal, I demonstrated that we performed these essential controls.

In this document, as requested by Dr. Pulverer, I extend this rebuttal to Reviewer #2 comments to our revised manuscript by responses to comment 1 of Reviewer #1, #2 and #3 that they made to our original manuscript. Here, I do not include the point-by-point response to all other comments of Reviewers because I did it when I submitted the revised manuscript.

Rebuttal of Comments regarding LiPR in the context of Control Diet

All 3 reviewers suggested in their review to our original manuscript (EMBOR-2023-57414V1) that we address the effects of the anti-obesity compound, LiPR, on adult neurogenesis in the context of the Control Diet:

Reviewer #1, comment 1: “In fact, this was examined (see Supplementary figures), but only for the cells in culture and, when performed with animals, was limited to 7 and 21 days, rather than 4 months, which would have been much more informative.”

Reviewer #2, comment 1: “A major issue is the lack of a LiPR-only group, which would much facilitate the interpretation of the results. The effect of LiPR alone is however tested, but only in comparison with the Control in one of the in vitro experiments (S.Fig. 3)”.

Reviewer #3, comment 1: “The main question that arises in here is whether LiPR would have an effect on control mice. It seems that a group is missing in the experimental design in which control mice are treated with LiPR during 7, 21 and the

last two weeks of the 4 months. Author must include this information or at least argue the election of the experimental design.”

RESPONSE: Because of this univocal consensus among reviewers, we generated new animals and analyses of LiPR + Control Diet groups. We systematically added analysis of LiPR-only treated mice for 2 time-points, 21-day, and 4-month. We analyzed different cell stages of the entire neurogenic process in these LiPR + Control Diet animals. We analyzed the impact of LiPR on the stem cells (tanycytes), proliferating cells and differentiated new neurons and astrocytes that originate from the stem cells. In addition, we analyzed the effects of LiPR on the number of two different subpopulations of tanycytes (alpha and beta tanycytes) and on the proportion of Vimentin-expressing tanycytes. We also analyzed proliferating cells both in the hypothalamic parenchyma and in the Median Eminence (ME).

Graphs from these new analyses are in Figure EV1C (the number of alpha-tanycytes), Figure EV1D (proportion of Vimentin+GFAP+ tanycytes), Figure EV1F (the number of beta tanycytes), Figure EV1M (the number of proliferating cells in the parenchyma) Figure EV1N (the number of proliferating cells in the ME), Figure EV5R (the number of new cells in the hypothalamic parenchyma), Figure EV5S (the number of new neurons), and Figure EV5T (the number of new astrocytes).

In addition, our revised manuscript contains analyses of the effects of LiPR that extend to cells *in vitro*. These include effects on tanycyte-derived neurospheres (Figure EV3E-I) and on primary tanycytes (Figure EV3J-L). Finally, our data include analyses of the effects of acute PrRP on naïve tanycytes (Figure EV4).

Taken together, these new results suggest that LiPR exerts some of its effects also in physiological conditions of Control Diet. Our results suggest that LiPR reduces cell proliferation in the 21-day time-point and increases number of new neurons in the 4-month time-point independently of diet. In our opinion, we satisfied the concerns of all 3 reviewers regarding the LiPR + Control Diet by a full set of control analyses.

We performed the *in vivo* analyses of LiPR + Control Diet for two time-points, 21-day and 4-months, because these complement results for two different phases of the exposure to the High Fat Diet (HFD). The 21-day time-point is for pre-obesity, whereas the 4-month time-point is for obesity. We did not include the 7-day time-point because it was redundant and because the 4-month time point and not the other time-points were explicitly requested by the reviewers. However, we want to emphasize that the manuscript focuses on the effects of an anti-obesity compound in the context of obesity. Determining the effects in the physiological conditions represents a different set of questions akin to studying the effects of insulin in non-diabetic subjects.

Rebuttal of Comments of Reviewer #2 to the Revised Manuscript (EMBOR-2023-57414V2) as sent to EMBO Reports on 14.9.2023

Reviewer #2: The manuscript fails to generate solid conclusions and only provides scattered descriptive data with weak relation to each other. This is due mostly, but not limited to, to two factors: The lack of a systematic inclusion of the LiPR-only group and the lack of thorough experiments that test the assumptions brought up regularly regarding the properties and changes in tanycytes/htNSCs.

RESPONSE: The claim that there is a “lack of a systematic inclusion of the LiPR-only group” is false. In the originally submitted manuscript, we analyzed LiPR-only effects not only *in vitro* but also *in vivo* by quantifying the number of BrdU+ cells and neurons in the MBH in the 21-day group (originally in S.Fig. 5O-Q, now in panels Fig EV5R-T) and by analyzing LiPR-only mice in the RNAseq experiment (Fig 5C). In the revised manuscript, to satisfy concern #1 of Reviewer #2, we systematically added analysis of LiPR-only treated mice as requested by Reviewer #2. We analyzed proliferating cells and both alpha- and beta-tanycytes for both 21-day and 4-month groups (EV1A-G, K-N) and BrdU-retaining neurons and cells for both 21-day and 4-month groups (Figure EV5P-T). In fact, we generated a new set of mice for the revision specifically to satisfy the concerns of Reviewer #2. We described the new analyses in the manuscript and in the point-to-point response to Reviewer #2. However, Reviewer #2 falsely claims we never addressed this key concern. In fact, Reviewer #2 contradict their own claims by the next comment where they acknowledge that “LiPR is exerting actions also in the control non-HFD”.

Reviewer #2: LiPR is exerting actions also in the control non-HFD. But there is no overall effect on weight. These data strongly suggest that neurogenesis is not mediating the anorexigenic effects of LiPR if anything, this needs to be clarified.

RESPONSE: The claim that “neurogenesis is not mediating the anorexigenic effects of LiPR” is not substantiated by our results and is misleading. The effects of LiPR on body weight are likely caused by neurogenesis-independent effects. We never claimed that LiPR acts solely through neurogenesis. The goal of our manuscript was not to prove that neurogenesis is required for actions of anti-obesity compounds but that anti-obesity compounds affect adult neurogenesis. To determine that neurogenesis is required for actions of anti-obesity compounds would require specifically ablating neurogenesis and test whether it diminishes anti-obesity effect of the compounds. While this is what we plan to do in the future, it is beyond the scope of this manuscript as agreed by the managing editor before our revision. Nevertheless, LiPR clearly exhibits cellular effects in the context of both Control and HFD, which is not contradictory. Similarly, Insulin also acts in physiological and obesity conditions.

Reviewer #2: The inclusion of the LiPR -alone group has been only in some experiments but it is lacking in others, resulting in weakened conclusions. The addition of this group should systematically added to all the experiments. The lack of effect of LiPR alone on weight in the 4-month paradigm does not justify the lack of this group in the rest of experiments because other clear effects of HFD and HFD+LiPR are found in the 7-day and 21-day paradigms.

RESPONSE: The claim that we did not include the analysis of LiPR-only analysis substantially is misleading and/or false. Our revised manuscript contains cell analyses of LiPR-only mice for 2 time points (21-day and 4-month) for 5 different data sets: proliferating cells, alpha-tanycytes, beta-tanycytes and BrdU-retaining cells and BrdU-retaining neurons in the Medial Basal Hypothalamus. These cell analyses represent the major cell analyses in the manuscript for a short-term group (21-day) and the long term/obesity group (4-months).

Reviewer #2: A especially notorious case is the effect on alpha-tanycytes. LiPR clearly increases their number in the HFD group in the 7-day paradigm, but HFD decreases it. The response to my comment 4 is not satisfactory. Alterations in beta-tanycytes might explain some of the results regarding total tanycytes or proportions of tanycytes, but not the increase in absolute numbers in alpha-tanycytes. It does not explain either the clear lack of effect on the 21-day and 4-month paradigms. There is data on weight gain, receptors expression, neurons, tanycytes, htNSCs, in vitro models, gene expression...but the experiments are too disconnected and no functional conclusion can be extracted.

RESPONSE: These claims of Reviewer #2 are misleading and/or false. In the comment #4 for to our manuscript before the revision, Reviewer #2 stated that at the 7-day time point “*The number of GFAP+ α -tanycytes is not significantly changed by HFD therefore LiPR does not rescue, but rather increases the number of GFAP+ α -tanycytes*”. In our revision, we acknowledged this and clarified our interpretation in the point-by-point response to Reviewer #2: “We thank the reviewer for this comment. We agree that we did not correctly interpret the original data on alpha-tanycytes. However, our new data show that LiPR rescues HFD-driven decrease in beta-tanycytes, which have neurogenic potential and higher cell quiescence than alpha-tanycytes (Haan *et al.*, 2013). These new results support our original statement, which we modified as following (on line 458): “The lower cell activation may be protective against adverse effects of the HFD, which is reflected in the rescue of the number of β -tanycytes by LiPR.” In addition, we discuss the effects of LiPR on different tanycyte populations in the Discussion (page 18).”

Our ANOVA analysis of alpha-tanycytes shows that treatment has a significant effect on their number (Figure 2E). Because alpha-tanycytes show different neurogenic potential and cell dynamics than beta-tanycytes, it is not surprising that LiPR affects one subpopulation but not the other. Importantly, the goal of our manuscript was not conducting complex cell lineage-tracing experiments but to demonstrate effects of anti-obesity compounds on tanycytes in general.

MORE SPECIFIC COMMENTS

Reviewer #2: What are the conclusions of the experiments regarding receptor expression within the overall potential functional role of LiPR and neurogenesis?

RESPONSE: LiPR receptors are found both on neural stem cells and new hypothalamic neurons suggesting that LiPR can directly act on different stages of the neurogenic process. These conclusions are in the Discussion on page 17 of the revised manuscript.

Reviewer #2: In contrast to alpha-tanycytes, beta-tanycytes are affected in all the time-course paradigms in a similar manner, being decreased by HFD, and increased by LiPR within the context of HFD. The LiPR omission is again flagrant here. The authors even claim later: "These results suggest that some but not all effects of LiPR on tanycytes are independent of HFD". Yet no LiPR-only group is included.

RESPONSE: As above, Reviewer #2 states "The LiPR omission is again flagrant here". This claim is false. We analyzed both alpha-tanycytes and beta-tanycytes (and the proportion of Vimentin+GFAP+ tanycytes) in LiPR-only animals for 2 time-points (21-day and 4-month) in Figure EV 1E-G.

Reviewer #2: The gliogenic versus the neurogenic capacity of the different populations of tanycytes is pointed out as relevant for the study but it is never addressed.

RESPONSE: The goal of this study was not to dissect specific cell potency of different tanycyte subpopulations but to show for the first time that anti-obesity compounds promote adult neurogenesis.

Reviewer #2: What are vimentin+ tanycytes? Alpha, beta...? Can they be truly considered to be htNSCs? The results are very similar to those obtained for alpha-tanycytes with only an effect on the 7-day paradigm and only in proportion but not in absolute numbers. How can this be explained?

RESPONSE: As described in detail in Results, Discussion and Methods, Vimentin+ tanycytes in our analyses are alpha-tanycytes that express GFAP and Vimentin. As in the response above, the goal of this manuscript was not to dissect the nature of htNSCs but to observe effects of anti-obesity compounds on tanycytes and adult neurogenesis. This comment of Reviewer #2 does not diminish or contradict the general interpretation of our results.

Reviewer #2: "and the universal marker of tanycytes, the Retina and anterior neural fold homeobox transcription factor, Rax.." Then how is it possible that Rax+beta-tanycytes or Rax+tanycytes are decreased by HFD but not increased in HFD+LiPR as measured in the other experiments? "This suggests that LiPR increases number of GFAP+ tanycytes but not all tanycytes positive by default for Rax". Was not Rax an universal marker of tanycytes? Once more, the lack of direct experimental testing generates confusing and lack of valid conclusions.

RESPONSE: Rax labels all tanycytes and some ependymocytes. It is not surprising that HFD but not LiPR exhibit different effects on different subpopulations of tanycytes.

Reviewer #2: Results with Ki67 or PCNA show changes in the 21-day paradigm but not in the 7-day paradigm, in opposition to the data from tanycyte populations. How

can these changes be reconciled?

RESPONSE: Ki67+ and PCNA+ cells include not only cells in the neurogenic process but other proliferating cells in the hypothalamus including microglia, macrophages, oligodendrocyte precursors – as we acknowledge this in the results and the discussion of the revised manuscript. Therefore, cell dynamics of cell proliferation is likely different for all proliferating cells than only for tanycytes or their daughter cells. These results are not contradictory.

Reviewer #2: Importantly, and this is the main issue, how can all the reported changes in all the different parameters analyzed in vivo and in vitro be linked to the effect on weight gain? The authors appear to aim to link PrRP receptor expression and neurogenesis, through effects on cell division and survival, to the anorexigenic effects of LiPR on tanycytes/htNSCs, but in reality, no actual link is shown. The discussion has been reworked but it still relies more on assumptions than on directly tested data.

RESPONSE: The claim that “The authors appear to aim to link PrRP receptor expression and neurogenesis, through effects on cell division and survival, to the anorexigenic effects of LiPR on tanycytes/htNSCs” is wrong and misleading. We did not aim to show that LiPR acts (solely) through neurogenesis but to determine whether anti-obesity compounds affect adult neurogenesis.

Reviewer #2: In the 21d protocol (Fig 1B), the treatment had a significant effect on the body weight, however, without any statistically significant difference between any of the treatment groups. I do not understand this sentence. Is there or not an effect?

RESPONSE: As explained in detail in Results and Methods, our group data were analyzed by the Analysis of Variance (ANOVA), or its non-parametric alternatives, with post-hoc Bonferroni test comparing individual data sets. Two-way ANOVA used for data in Fig 1B (and also Fig 1A and 1C) showed that treatment (that is exposure to Control or HFD and/or administration of LiPR) has a significant effect on the body weight. Detailed P and F values are provided in the legend of Figure 1. The post-hoc Bonferroni tests revealed, however, that there was no significant difference between any of the treatment groups suggesting that, for example, HFD or HFD+LiPR did not differ from Controls. We presented our statistical analyses clearly and according to the standard format. It is confusing and disappointing that Reviewer #2 was not able to follow it.

Dear Dr. Petrik

Thank you for the submission of your revised manuscript and detailed rebuttal.

I have re-evaluated the peer review and revision process from the beginning now, including assessment of every point raised by you in response to ref 2's final report in detail and appreciate your patience as intervening travel commitments delayed this important due diligence process.

I am pleased to say that I agree with the responses that you made and we will be pleased to publish the paper in EMBO Reports after minor additional revision as outlined below.

I would also like to add that all three referees raised extremely consistent points about the lack of systematic normal diet controls among other things. I appreciated that these controls were partially added in revision, which certainly represented considerable additional work on your part. Since this issue permeates most of the referee points and given the unique effects seen at this early time point, it is a little less clear to me why the controls for the 7d time point were not also added at the time of revision. I suggest to textually include a sentence on this in the final paper. Your argument is that 21d and 7d collectively represent early times - but then again these timepoints show partially differential affects as I understand it.

In fig 1B I admit that I am still confused that you report different conclusions based on two statistical approaches. I would politely request to revisit this and explain clearly in paper why both tests were applied and to justify the conclusion taken (by all means using a dedicated statistical section in the materials & methods). I think the confusion by ref 2 and myself will reflect a broader readership, thus warranting the adjustment.

Given the back and forth with ref 2, I also suggest to more explicitly include statements along the lines of your rebuttal 'We did not aim to show that LiPR acts (solely) through neurogenesis, but to determine whether anti-obesity compounds affect adult neurogenesis.'

Please review the claims of the paper to ensure they are definitive as far as the current data presented allows. In particular, to avoid claiming causality where correlations are reported and I recall here also ref 1's original point that 'the concept of LiPR/PrRP preventing the exhaustion of the hypothalamic stem cell pool was not clear, because it is not shown that this pool does actually get exhausted under normal or HFD conditions.'

I would also suggest to expand the discussion to include future issues to be addressed e.g. on food intake regulation.

Finally, please note these issues raised by my colleagues:

1) The manuscript sections should be in the following order: Title page - Abstract & Keywords - Introduction - Results - Discussion - Materials & Methods - Data Availability - Acknowledgments - Disclosure Statement & Competing Interests - References - Figure Legends - Expanded View Figure Legends.

2) Add headings 'References' and 'Expanded View Figure Legends'

3) Disclosure and Competing Interests Statement is missing

4) Data not shown: twice on p 7

5) Callouts for Fig 1Q, 3I, Fig EV4 are missing

6) Source Data: the checklist needs to be completed; the following appear to be missing (images and some numerical data): 1IJKMO, 2ABCHJK, 3ABCFG, 4ABCDJ, 5ABCDEGJ; the SD should be uploaded as one zip folder per one figure

7) Synopsis text and image are missing

Please contact our assistants if any of these requests are unclear.

We hope to process a final version incorporating the suggested minor revisions rapidly for publication.

Best wishes,

Bernd Pulverer

~~~~~  
Bernd Pulverer, Ph.D.  
Chief Editor, EMBO Reports  
EMBO

Meyerhofstrasse 1, D-69117 Heidelberg

Tel: +4962218891501

bernd.pulverer@embo.org

~~~~~

Rev_Com_number: RC-2023-01900

New_manu_number: EMBOR-2023-57414V4

Corr_author: Petrik

Title: An analogue of the Prolactin Releasing Peptide reduces obesity and promotes adult neurogenesis

2.11.2023

EMBOR-2023-57414V4

Manuscript EMBOR-2023-57414V4 | [RC-2023-01900]**Corresponding author(s):** David Petrik**Rebuttal of Minor Revision to the Revised Manuscript (EMBOR-2023-57414V4) as requested by Dr. Pulverer, Editor-in-Chief of the EMBO Reports on 22.10.2023.**

The responses are intercalated in the text of the email from Dr. Pulverer with point-by-point responses for each point raised. The newly adjusted text is highlighted by the blue font both in this rebuttal and in the submitted manuscript text. After addressing the points, we highlight changes in Figure 4 and Figure EV5.

From: contact@emboreports.org <contact@emboreports.org>**Sent:** Sunday, October 22, 2023 2:16 PM**To:** David Petrik <PetrikD@cardiff.ac.uk>**Subject:** Decision on Manuscript EMBOR-2023-57414V4 | [RC-2023-01900] [REV]

Dear Dr. Petrik

Thank you for the submission of your revised manuscript and detailed rebuttal. I have re-evaluated the peer review and revision process from the beginning now, including assessment of every point raised by you in response to ref 2's final report in detail and appreciate your patience as intervening travel commitments delayed this important due diligence process.

I am pleased to say that I agree with the responses that you made and we will be pleased to publish the paper in EMBO Reports after minor additional revision as outlined below.

1. I would also like to add that all three referees raised extremely consistent points about the lack of systematic normal diet controls among other things. I appreciated that these controls were partially added in revision, which certainly represented considerable additional work on your part. Since this issue permeates most of the referee points and given the unique effects seen at this early time point, it is a little less clear to me why the controls for the 7d time point were not also added at the time of revision. I suggest to textually include a sentence on this in the final paper. Your argument is that 21d and 7d collectively represent early times - but then again these timepoints show partially differential affects as I understand it.

RESPONSE: As suggested, we added following text to address the selection of two time-points for the Control Diet + LiPR data set (page 22, lines 551-553): “**For the Control diet + LiPR, we chose 21d and 4mo time points because they complement results for two different phases of the HFD exposure. The 21-day time-point represents pre-obesity, whereas the 4-month time-point is for obesity.**”

2. In fig 1B I admit that I am still confused that you report different conclusions based on two statistical approaches. I would politely request to revisit this and explain clearly in paper why both tests were applied and to justify the conclusion taken (by all means using a dedicated statistical section in the materials & methods). I think the confusion by ref 2 and myself will reflect a broader readership, thus warranting the adjustment.

RESPONSE: At the end of the method section, we expanded the segment on statistics pages 36-37), which was part of the manuscript text since the initial submission. We describe in greater details how we approach the statistics and we double-checked and properly annotated the used statistical methods in the figure legends. This is the text regarding ANOVA and post-hoc tests in the section, which we adjusted to make it more explicit as requested (page 37, lines 938-940):

“For multiple factor or group comparison, One-Way or Two-Way Analysis of Variance (ANOVA) was used with the Bonferroni’s or Tukey’s post-hoc test for the cross-comparison of individual data sets.”

In addition, we modified the text about data from Fig 1B to make it more explicit (page 6, lines 131-136):

In the 21d protocol (Fig 1B), the Two-Way ANOVA revealed that treatment, which includes either Control or HFD with or without LiPR, had a significant effect on the body weight. However, the multiple comparison Bonferroni post-hoc test did not find any statistically significant difference between any of the treatment groups, suggesting that HFD or LiPR alone cannot cause the variance found by ANOVA.

3. Given the back and forth with ref 2, I also suggest to more explicitly include statements along the lines of your rebuttal 'We did not aim to show that LiPR acts (solely) through neurogenesis, but to determine whether anti-obesity compounds affect adult neurogenesis.'

RESPONSE: In the discussion, we added following statement (page 16, lines 402-404): “These findings suggest that LiPR affects adult neurogenesis, but we cannot claim that its anorexigenic actions are conveyed through neurogenesis.”

4. Please review the claims of the paper to ensure they are definitive as far as the current data presented allows. In particular, to avoid claiming causality where correlations are reported and I recall here also ref 1's original point that 'the concept of LiPR/PrRP preventing the exhaustion of the hypothalamic stem cell pool was not clear, because it is not shown that this pool does actually get exhausted under normal or HFD conditions.'

RESPONSE: Thank you for highlighting this. In our first rebuttal on 12.7.2023, we acknowledged that we over-interpreted our data. This is what we wrote in response to Comment 4 of Reviewer 1:

“We thank the reviewers for their insights. We agree that our initial quantification of alpha-tanycytes only did not show that HFD exhausts tanycytes. We over-interpreted the data. In the revision, however, we show that LiPR rescues HFD-driven decrease in beta-tanycytes, which show neurogenic potential and higher cell quiescence than alpha-tanycytes (Haan *et al.*, 2013). These new results support our original statement, which we modified as following (line 458): “The lower cell activation may be protective against adverse effects of the HFD, which is reflected in the rescue of the number of β -tanycytes by LiPR.”

The adjusted statement is highlighted in red on page 18, lines 460-462.

5. I would also suggest to expand the discussion to include future issues to be addressed e.g. on food intake regulation.

RESPONSE: We added the following statement at the end of the discussion to address the issue (page 23, lines 582-587): “The future work should determine whether specific ablation of adult neurogenesis in the hypothalamus would diminish anorexigenic effects of anti-obesity compounds along the lines of previous causative evidence that anti-depressants require neurogenesis in the hippocampus for their action (Santarelli *et al.*, 2003).”

Finally, please note these issues raised by my colleagues:

1) The manuscript sections should be in the following order: Title page - Abstract & Keywords - Introduction - Results - Discussion - Materials & Methods - Data Availability - Acknowledgments - Disclosure Statement & Competing Interests - References - Figure Legends - Expanded View Figure Legends.

RESPONSE: We organized the manuscript according to the above order.

2) Add headings 'References' and 'Expanded View Figure Legends'

RESPONSE: The headings were added.

3) Disclosure and Competing Interests Statement is missing.

RESPONSE: The statement was added.

4) Data not shown: twice on p7.

RESPONSE: We are aware of this. We did not think these data are essential for the manuscript to be shown. We prefer to leave this as it is.

5) Callouts for Fig 1Q, 3I, Fig EV4 are missing.

RESPONSE: The callouts were added. Thank you for noticing their absence.

6) Source Data: the checklist needs to be completed; the following appear to be missing (images and some numerical data): 1IJKMO, 2ABCHJK, 3ABCFG, 4ABCDJ, 5ABCDEGJ; the SD should be uploaded as one zip folder per one figure.

RESPONSE: We apologize for not including all the source data. By our oversight, we did not realize that the source data include representative pictures. We corrected this and included the source data for all panels in the main figures. The files are now uploaded as zip folders.

7) Synopsis text and image are missing.

RESPONSE: We added the synopsis text, including the bullet points, and the synopsis image.

Changes in Figure 4 (Fig 4) and in Figure EV5 (Fig EV5).

Before we submitted this minor revision, we went through every data file relevant for this manuscript as a thorough final check of the data. We identified a mistake in Figure 4 and in Figure EV5 that we describe here to be completely transparent. These mistakes, however, do not substantially change any major conclusion of the manuscript.

In panels Fig 4G (BrdU+ neurons/mm³) and in panels Fig EV5L (BrdU+ cells/mm³) and Fig EV5N (BrdU+ astro/mm³) we used source data from pilot quantifications, which are not the correct data for these three panels. The lab member, who is a co-author in this manuscript and who conducted these analyses, left the laboratory 2 years ago and unclear labelling of her data files caused us to use the wrong data sets in Fig 4G, Fig EV5L and Fig EV5N for the original submission of the manuscript. We are extremely sorry about this and apologize for this mistake.

Here, we identified the correct data sets and made new graphs for all three panels. In panel Fig 4G, there is no change in the outcome of the data. There is no statistical difference in the density of BrdU+ neurons in the Arcuate Nucleus between any of the data sets in the new graph as there was no difference in the old graph. Therefore, there is no change in the figure legend or the manuscript text.

In Fig EV5L, there is no change in the outcome of the data. In Fig EV5N, however, we no longer see any difference between the HFD (red square symbols) and HFD + Liraglutide (green triangle symbols) in the density of BrdU+ astrocytes. However, given the fact that the effects of Liraglutide on newly generated astrocytes in the 21-day time-point is a very minor finding of the manuscript, we don't think this correction substantially changes any major conclusion.

In addition, because the data sets for Control in Fig EV5L and Fig EV5N (black circle symbols) were also used as Control in Fig EV5R and Fig EV5T, we had to adjust these two graphs as well. There is no change in the outcome of Fig EV5T (BrdU+ astrocytes/mm³). However, there is no longer a statistical difference between the Control and Control + LiPR in the density of BrdU+ cells for the 21-day time-point in the panel Fig EV5R, while the statistical difference is maintained for the 4-month time-point, which was the main message of this graph.

We reflected all these changes in the figure legend of Figure EV5 and acknowledged these changes in the text of the manuscript. Below, the red font is for the changes we made to address reviewer's comments in the revision submitted in July 2023 (EMBOR-2023-57414V2). The strike-through font annotates deleted text to reflect the changes described above. On the page 12 of the manuscript, we changed the claims to the following:

"In the 21d protocol (Fig EV5H-K), LiPR did not change the number of cells, neurons, or astrocytes positive for BrdU in the MBH parenchyma in the context of HFD (Fig EV5L-N). **However, in the Arc, the primary nutrient and hormone sensing neuronal nucleus of MBH (Betley *et al.*, 2013), treatment significantly influenced number of BrdU+ neurons. 21d HFD increased their number, which was reversed by LiPR or Liraglutide (Fig EV5O).** Interestingly, Liraglutide decreased the number of BrdU+ cells

compared to Control. ~~and decreased the number of BrdU+ astrocytes compared to HFD.~~ Taken together, these results suggest that Liraglutide in the context of HFD and in the MBH parenchyma and LiPR in the context of HFD (in the Arc) reduces the number of new BrdU+ cells in the MHB but only in the 21d and not 7d protocol.”

On page 13 of the manuscript, we did not change the text related to the new graphs in Fig EV5R and Fig EV5T because the original text from the revision in July 2023 highlights changes in the neurogenesis at the 4-month time-point, which was not affected by the latest changes in these panels. This underscores that these most recent changes do not substantially alter the major conclusions of the manuscript. This is the relevant text on page 13:

“LiPR in context of Control diet (Fig EV5P-T) significantly increased the number of BrdU+ neurons but decreased the number of astrocytes (Fig EV5S-T) suggesting it promotes adult neurogenesis in the MBH of older mice that are not experiencing DIO.”

We apologize for late changes in Figure 4 and Figure EV5. But we are confident that these minor changes do not substantially change the claims of our manuscript, considering that they result only in a single deletion of a half of a sentence on page 12.

Dr. David Petrik
Cardiff University
School of Biosciences
Museum Avenue
Cardiff, Wales CF10 3AX
United Kingdom

Dear Dr. Petrik,

I am very pleased to accept your manuscript for publication in EMBO reports. Thank you for your contribution to our journal.

Since we are at the end of year, we will try to ensure online publication this year, but please return proofs and any queries as soon as possible.

Yours sincerely,

Bernd Pulverer

~~~~~  
Bernd Pulverer, Ph.D.  
Chief Editor, EMBO Reports  
EMBO  
Meyerhofstrasse 1, D-69117 Heidelberg  
Tel: +4962218891501  
[bernd.pulverer@embo.org](mailto:bernd.pulverer@embo.org)  
~~~~~
